# End-to-end topographic networks as models of cortical map formation and human visual behaviour

Zejin Lu ®[1,2,7] ✉, Adrien Doerig ®[1,3,7], Victoria Bosch ®[1,7], Bas Krahmer[4], Daniel Kaiser ®[5,6,8], Radoslaw M. Cichy ®[2,8] & Tim C. Kietzmann ®[1,8] ✉

A prominent feature of the primate visual system is its topographic organization. For understanding its origins, its computational role and its behavioural implications, computational models are of central importance. Yet, vision is commonly modelled using convolutional neural networks, which are hard-wired to learn identical features across space and thus lack topography. Here we overcome this limitation by introducing all-topographic neural networks (All-TNNs). All-TNNs develop several features reminiscent of primate topography, including smooth orientation and category selectivity maps, and enhanced processing of regions with task-relevant information. In addition, All-TNNs operate on a low energy budget, suggesting a metabolic benefit of smooth topographic organization. To test our model against behaviour, we collected a dataset of human spatial biases in object recognition and found that All-TNNs significantly outperform control models. All-TNNs thereby offer a promising candidate for modelling primate visual topography and its role in downstream behaviour.

The spatial arrangement of neurons in the primate visual system is highly structured, following a systematic functional topography that is evident across all visual streams and hierarchical levels[1–3]. For example, early visual cortex (V1) is organized in repeating motifs of orientation selectivity that vary smoothly[4,5]. In higher-level visual cortex, spatial clusters of neurons respond preferentially to abstract stimulus categories, such as faces[6,7], bodies[8] and scenes[9,10], among other spatial organizational structures[11–14]. Human behaviour, too, exhibits spatial regularities. For example, objects are more easily recognized when displayed in their typical spatial position[15–17]. However, despite being a hallmark feature of primate vision, the origins of topographic

organization, its computational role and its relation to behaviour remain poorly understood.

To make progress on these questions, computational models take on a central role, as they allow researchers to express, instantiate and test computational hypotheses, including otherwise impossible experimental manipulations and direct (causal) access to their internal parts. In the space of computational models of vision, artificial neural networks (ANNs) are of particular interest. ANNs process naturalistic visual input while performing complex tasks and are able to bridge levels of explanation from single neurons to behaviour[18–20]. The most widely used ANNs in visual neuroscience are convolutional neural

[1]Machine Learning Group, Institute for Cognitive Science, Osnabrück University, Osnabrück, Germany. [2]Neural Dynamics of Visual Cognition Group, Department of Education and Psychology, Freie Universität Berlin, Berlin, Germany. [3]Cognitive Computational Neuroscience Lab, Department of Education and Psychology, Freie Universität Berlin, Berlin, Germany. [4]Donders Institute for Brain, Cognition and Behaviour, Department for Artificial Intelligence, Radboud University, Nijmegen, the Netherlands. [5]Neural Computation Group, Mathematical Institute, Justus-Liebig-Universität Gießen, Gießen, Germany. [6]Center for Mind Brain and Behavior, Philipps-Universität Marburg and Justus-Liebig-Universität Gießen, Marburg, Germany. [7]These authors contributed equally: Zejin Lu, Adrien Doerig, Victoria Bosch. [8]These authors jointly supervised this work: Daniel Kaiser, Radoslaw M. Cichy, Tim C. Kietzmann. ✉e-mail: zekinglu@gmail.com; tim.kietzmann@uni-osnabrueck.de

networks (CNNs), which have been successful at predicting visually evoked neural activities across multiple hierarchical levels[21–26] and at accounting for complex visual behaviour[27–30].

Yet, despite these important advances in modelling vision, CNNs are centred on an architectural inductive bias that limits them when it comes to modelling topography: weight sharing. Weight sharing, which refers to the application of identical feature detectors across space, is sensible for engineering purposes as it facilitates efficient learning and enables spatially invariant object recognition. At the same time, it presents a challenge in using CNNs as a modelling class in neuroscience. First, it is biologically implausible to assume that changes to the weights of one neuron are copied to other neurons all over the cortical sheet. Second, the fact that the same features are extracted across spatial locations makes it difficult to model the effects of topography on behaviour. As a first step in the direction of mirroring topography, recent work has started modifying CNNs—for example, by mapping CNN units to a virtual 2D grid[31,32], by using self-organizing maps based on CNN features[33,34] or by adding other topographic processes to late CNN layers[35]. While these models are able to reproduce topographic features of the brain, they remain reliant on weight sharing. Computational neuroscience thus remains in need of new ANN models that combine the ability of deep learning to perform complex tasks with the ability to operate on natural images while avoiding weight sharing to allow for fully topographic models.

Here we present such a model, which we call all-topographic neural networks (All-TNNs). All-TNNs are centred on three features. The first is locality: feature weights are learned individually at each location and not enforced to be exact duplicates of other unit weights. The second is arrangement along the cortical sheet: units in the model are arranged along an artificial 2D cortical sheet, which thus provides a sensible metric for the spatial distance between units. The third is smoothness: the primate brain exhibits a smooth decay in connectivity between neurons with increasing cortical distance[35,36]. As a result, topographies in the brain operate between the two extremes of spatially discontinuous models, where each unit detects a different feature, and spatially uniform models, where all units detect the same feature[13,37–39]. All-TNNs mirror this observation of intermediate-level smoothness.

We demonstrate that All-TNNs better reproduce important properties of neural topography in the primate visual system than CNNs and further control models, and they are able to flexibly and efficiently allocate resources depending on dataset statistics. Using an experimental dataset of human visual object recognition, we furthermore show that All-TNNs better align with spatial biases observed in human behaviour.

## Results

To study the emergence of feature topographies and their impact on behaviour, we developed fully topographic neural networks, called All-TNNs (Fig. 1 and Methods). Similar to CNNs, each unit has a local receptive field in the layer below. However, contrary to CNNs, All-TNNs have no weight sharing—that is, units at different spatial locations can learn different weights. In addition, units of each network layer are arranged along a 2D cortical sheet. Sets of neighbouring units on this 2D sheet share the same receptive field (that is, they receive connections from the same spatially limited area in the layer below). Finally, we use a smoothness loss that encourages neighbouring units to have similar weights. By tuning the strength of this loss as a hyperparameter $\alpha$, we can navigate the continuum from spatially discontinuous to spatially uniform feature selectivity in All-TNNs.

### All-TNNs classify images with smooth weight topography

To investigate the effect of the smoothness loss, we trained All-TNNs with different values of $\alpha$ ($\alpha \in [1, 10, 100]$). In all of our experiments, All-TNNs were contrasted with two main control models: CNNs and locally connected networks (LCNs; that is, All-TNNs with $\alpha = 0$). The controls have matching numbers of units, the same kernel sizes and

identical hyperparameter settings as All-TNNs (Supplementary Table 1). All models are trained on ecoset, a dataset of natural images with labels for 565 object categories selected to be representative of concrete categories that are of importance to humans[24] (see Methods for further training details). Early stopping was performed for all models to prevent overfitting (Supplementary Table 2). We trained five instances of each network type with different random initializations that we treat as experimental subjects[40].

In terms of classification performance, All-TNNs with $\alpha \in [1, 10, 100]$ achieve comparable accuracy to LCNs (LCN performance: 0.369; 95% confidence interval (CI), (0.366, 0.372); All-TNN ($\alpha = 1$) performance: 0.345; 95% CI, (0.343, 0.347); All-TNN ($\alpha = 10$) performance: 0.360; 95% CI, (0.355, 0.365); All-TNN ($\alpha = 100$) performance: 0.358; 95% CI, (0.351, 0.360)). CNNs outperform all other models in classification accuracy (CNN performance: 0.4323; 95% CI, (0.427, 0.437); two-sided permutation test, $n = 1 \times 10^4$, largest $P < 7.9 \times 10^{-3}$; Benjamini–Hochberg false discovery rate (BH-FDR) correction with $P < 0.05$; see Supplementary Fig. 1a for all statistical comparisons), in line with the well-known image-recognition abilities of this architecture. The smoothness loss in All-TNNs works as envisioned: All-TNNs develop spatially smooth weights (quantified by the inverse cosine distance between the weights of neighbouring units, averaged across all units and layers; All-TNN ($\alpha = 1$) smoothness: 1.374; 95% CI, (1.372, 1.376); All-TNN ($\alpha = 10$) smoothness: 17.179; 95% CI, (16.888, 17.469); All-TNN ($\alpha = 100$) smoothness: 364.327; 95% CI, (359.322, 369.333)), while CNNs and LCNs do not (quantified by mapping their activities onto the same 2D sheet as used for All-TNNs (Methods); LCN smoothness: 0.338; 95% CI, (0.338, 0.339); CNN smoothness: 0.335; 95% CI, (0.332, 0.337)). Furthermore, this effect is true of all network layers (Supplementary Fig. 2a), and increasing magnitudes of $\alpha$ lead to increasingly smoothly varying weights across units (see Supplementary Fig. 1b for statistical comparison).

To investigate whether structure emerges within the spatial extent of individual layers, we computed the inverse cosine distance between weights of neighbouring units at each spatial location of each layer (Supplementary Fig. 2b). This analysis reveals a strong centre–periphery distinction: while All-TNNs exhibit feature diversity in the centre of the model layer, the weights become smoother in the periphery of all model layers.

In summary, All-TNNs are able to learn smooth weights while at the same time successfully classifying a large dataset of natural object images. Furthermore, each layer shows a centre–periphery organization with more diverse weights in the centre of each layer.

### Topographic features of the ventral stream emerge in All-TNNs

Smooth orientation selectivity maps in V1 are a hallmark topographic feature of the primate visual system[1]. We thus tested whether the first layer of All-TNNs reproduces this feature upon training. To determine orientation selectivity for each unit in the layer, we followed the standard analysis procedure used in electrophysiology[4] by presenting sine-wave gratings of different angles and phases to the network and determining the angle for which each unit is most responsive (Methods). We found that the first All-TNN layer exhibits smooth orientation selectivities reminiscent of orientation selectivity maps in primate V1 (Fig. 2a, left; see Supplementary Fig. 3 for all model instances), including the emergence of pinwheel discontinuities (Supplementary Fig. 4). For larger $\alpha$ values, the orientation selectivity maps exhibit increasingly large cluster sizes (Fig. 2a, right; for statistical analysis, see Supplementary Fig. 5), show a stronger increase in cluster sizes with eccentricity (Supplementary Fig. 6) and are overall smoother (power spectrum analysis, Supplementary Fig. 7). As a comparison, we visualized the feature selectivities of the control models by mapping their activities on the same 2D sheet as used for All-TNNs (Methods). Because CNNs use weight sharing, which copies the same set of feature detectors at all spatial locations, we expected discrete repeating selectivity

**Fig. 1 | All-TNNs. a**, We use All-TNNs with six topographic layers of different dimensions and kernel sizes, followed by a fully connected category readout (565 ecoset categories) with softmax. In each layer, units are arranged retinotopically on a 2D sheet. Sets of neighbouring units on the sheet share the same receptive field in the layer below them. A smoothness loss encourages similar weights in neighbouring units on the sheet (as illustrated for one unit by blue arrows). Network training optimizes a composite objective, consisting of a classification cross-entropy loss and the smoothness loss, which is scaled by a tunable factor $\alpha$.

**b**, Classification accuracy of all All-TNNs and control models on the ecoset test set. The error bars show 95% CIs based on bootstrapping ($n = 1,000$). **c**, Spatial smoothness after training was quantified for all models as 1/(average cosine similarity between the weights of neighbouring units). Unit weights are significantly smoother in All-TNNs than in control models, and this effect increases with the $\alpha$ parameter that scales the smoothness loss (the 95% CIs are too small to see; the values are reported in the main text).

patterns instead of smoothly changing topography. This is indeed what we observed (Fig. 2a, left; Supplementary Figs. 3 and 5). Importantly, LCN control networks do not develop smooth selectivity maps either but exhibit salt-and-pepper maps instead. This suggests that the topographic organization observed in All-TNNs does not emerge purely from learning to categorize natural objects under the influence of autocorrelation in the input statistics. Notably, the topographic organization in All-TNNs is not present before training but emerges in the first epochs (Supplementary Fig. 8). Continued training largely fine-tunes selectivities to increase performance within this topographic scaffold, rather than bringing about further broad changes in the overall topographic structure.

In line with the centre–periphery distinction in weight smoothness described earlier, orientation selectivity in All-TNNs also exhibits a strong centre–periphery organization, with a higher variability of feature detectors in the layer centre and increasingly smooth feature selectivity towards the periphery. To quantify this observation, we computed the variability of selectivities at each location on the sheet by computing the local Shannon entropy of orientation

selectivities (entropy is low if all units in the sliding window have similar orientation selectivities; Fig. 2b, top; Supplementary Fig. 9 and Methods). All-TNNs exhibit a marked decline in entropy as eccentricity increases—that is, they exhibit less varied feature selectivity in the periphery (Fig. 2b, bottom). This effect is small for $\alpha = 1$ and strengthens with increasing $\alpha$. In contrast, entropy is consistently high for control CNNs and LCNs. This centre–periphery distinction in All-TNNs may seem surprising since All-TNNs do not have a central bias in their architecture or loss terms (note that the smoothness loss is applied in a toroidal fashion to avoid border effects; Supplementary Fig. 10). The location of this region with high feature variability must therefore depend on the location of task-relevant information in the network's input. To confirm this, we trained an All-TNN ($\alpha = 10$) on all ecoset images shifted by 30 pixels to the bottom right. Indeed, this results in a corresponding shift of the region with high feature variability (Fig. 2c and Supplementary Fig. 11). These results indicate that All-TNNs emphasize regions with more task-relevant information by developing more varied feature detectors in the respective locations.

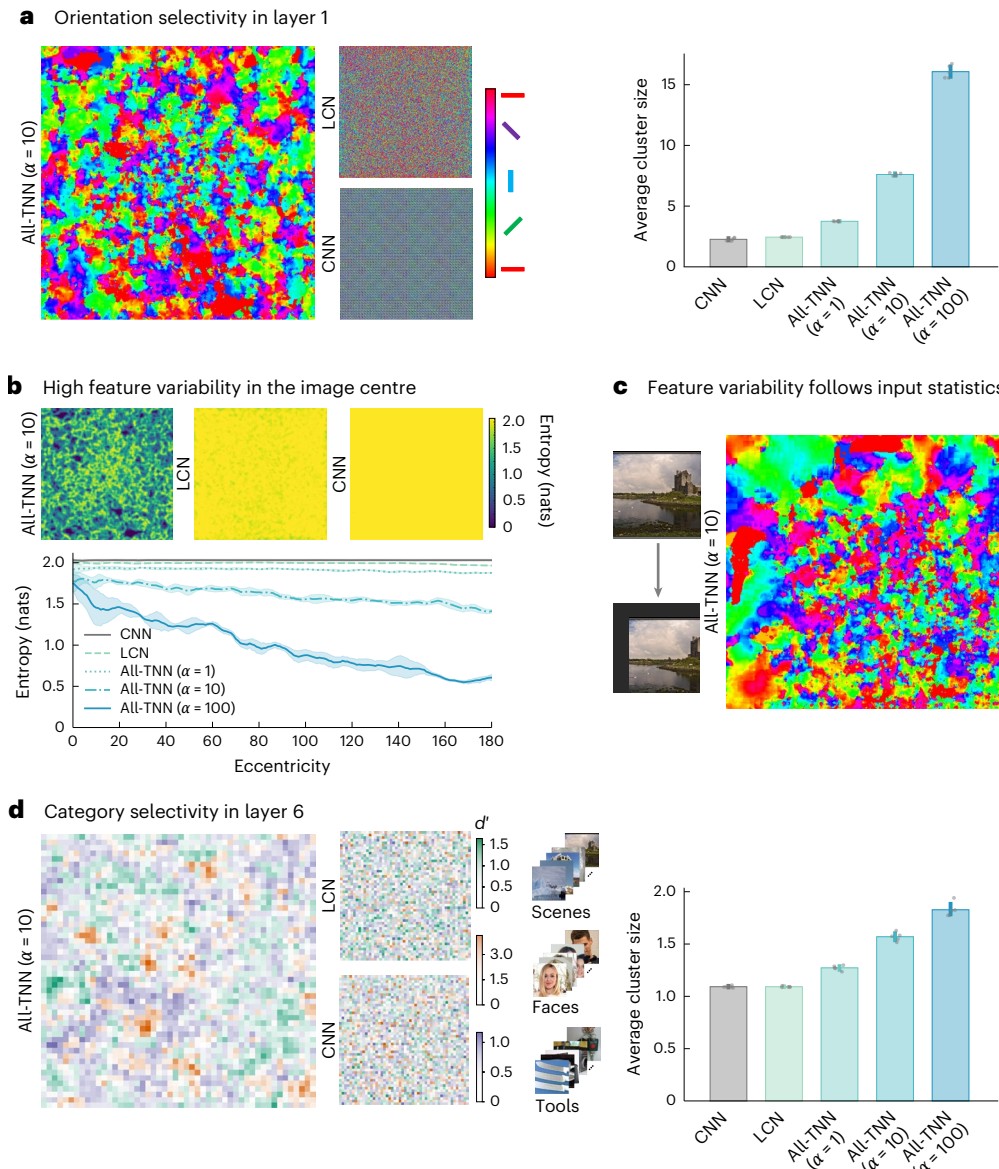

**Fig. 2 | All-TNNs mirror key features of the visual system's topography. a**, Left, visualization of orientation selectivity across the first layer of an All-TNN ($\alpha$ = 10), an LCN and a CNN (network instances with random seed 1; see Supplementary Fig. 3 for all seeds). Right, average cluster size of orientation selectivity for all models (the data are averaged across all seeds; the error bars show standard deviation across seeds; $n$ = 5). **b**, Top, entropy maps as a measure of local variability in feature selectivity for different models (seed 1; see Supplementary Fig. 9 for all seeds). Bottom, entropy decreases more with eccentricity as $\alpha$ increases in All-TNNs. Entropy remains relatively constant for CNNs and LCNs (the data are averaged across all seeds; the shaded regions indicate the 95% CIs).

**c**, Training on ecoset with all images shifted towards the bottom right leads to a corresponding shift of the region with high feature variability in orientation selectivity, indicating that feature variability is linked to task-relevant information in the network input. **d**, Left, analysis of the last network layer selectivity for high-level visual categories (measured using $d'$ for tools, scenes and faces; seed 1; see Supplementary Fig. 13 for all seeds; $n$ = 5). Right, average size of category-selective clusters (averaged over scenes, faces and tools) for All-TNNs and control models. All CNN and LCN layers are visualized using the same reshaping procedure as for the creation of All-TNN 2D sheets (Methods).

On the basis of this observation, we next tested whether units in regions with redundant selectivities can be lesioned without impacting task performance. To do so, we used the Shannon entropy maps described above as a guideline and lesioned 25% of units, in regions with either low entropy (that is, homogenous selectivities) or high entropy (that is, varied selectivities; Methods and Supplementary Fig. 12). CNNs have a strictly constant feature variability across space, hardwired by weight sharing (Fig. 2b), rendering them ill-suited for this experiment. For All-TNNs, we found for all values of $\alpha$ that the model performance strongly deteriorates when the high-entropy (heterogenous) regions are lesioned ($\alpha$ = 1 baseline performance, 0.345; 95% CI,

(0.343, 0.347); performance after high-entropy lesion, 0.211; 95% CI, (0.206, 0.216); dependent two-sample $t$-test, $P < 1 \times 10^{-3}$; $\alpha$ = 10 baseline performance, 0.359; 95% CI, (0.355, 0.364); performance after high-entropy lesion, 0.102; 95% CI, (0.089, 0.114); dependent two-sample $t$-test, $P < 1 \times 10^{-3}$; $\alpha$ = 100 baseline performance, 0.357; 95% CI, (0.356, 0.359); performance after high-entropy lesion, 0.101; 95% CI, (0.086, 0.117); $P < 1 \times 10^{-3}$; all $P$ values BH-FDR-corrected). Lesioning the low-entropy (homogeneous) regions led to a small but significant reduction in task performance ($\alpha$ = 1 performance after low-entropy lesion, 0.305; 95% CI, (0.301, 0.309); dependent two-sample $t$-test, $P < 1 \times 10^{-3}$; $\alpha$ = 10 performance after low-entropy lesion, 0.291; 95% CI,

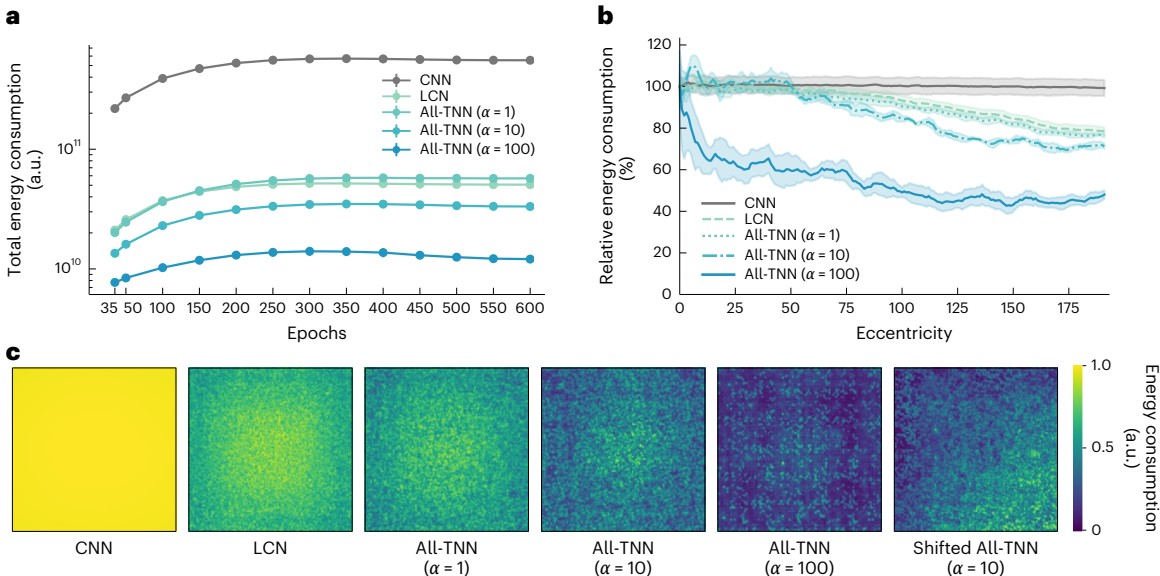

**Fig. 3 | Energy consumption in All-TNNs and control models. a,** Total energy consumption summed across all network units. Energy consumption increases with training for all models, but All-TNNs consistently consume less energy than other models. Higher $\alpha$ values increase this effect. **b,** Relative energy consumption at different eccentricities for various models (after early stopping (Methods); the data are averaged across all seeds, and the shaded regions indicate the 95% CIs; $n = 5$). All-TNNs show a steady decline of energy consumption with eccentricity. Higher $\alpha$ values increase this effect. **c,** Maps of energy consumption summed across layers for different trained models (early stopping, normalized by division by the maximum value). Note that the last layer is not included in this analysis, as it has no spatial information (that is, all units have the entire image in their receptive field).

(0.286, 0.295); dependent two-sample $t$-test, $P < 1 \times 10^{-3}$; $\alpha = 100$ performance after low-entropy lesion, 0.245; 95% CI, (0.235, 0.256); $P < 1 \times 10^{-3}$; all $P$ values BH-FDR-corrected). This reduction was significantly less than the effect of lesioning high-entropy regions ($P < 1 \times 10^{-3}$ for all models, BH-FDR-corrected). This finding was reproduced in the All-TNN trained on the images shifted to the bottom right (one network instance), showing that feature variability, and not layer periphery, is the relevant signal to lesion units (baseline performance, 0.25; performance after high-entropy lesion, 0.035; performance after low-entropy lesion, 0.221). In LCNs, entropy is almost constant across the layer. Lesioning relatively lower-entropy regions harms performance instead of improving it (baseline performance, 0.370; 95% CI, (0.368, 0.371); performance after high-entropy lesion, 0.321; 95% CI, (0.317, 0.325); dependent two-sample $t$-test, $P < 1 \times 10^{-3}$; performance after low-entropy lesion, 0.257; 95% CI, (0.250, 0.265); dependent two-sample $t$-test, $P < 1 \times 10^{-3}$). Hence, only All-TNNs allow the smoothness of the feature maps to act as reliable proxy for which units can safely be lesioned. Once lesioned, All-TNNs, with minimal loss in performance, use more units in locations with task-relevant information. This is reminiscent of cortical magnification, another key topographic feature of primate V1, which exhibits more neurons per degree of visual angle in regions processing foveal information.

Having investigated lower-level topographic features of All-TNNs, we moved on to higher-level representations and analysed whether the last layer reproduces another main topographic feature characterizing primate visual cortex: smooth clustering based on category-specific neural tuning in higher-level regions. To do so, we computed unit selectivities for faces, tools and places (computed using $d'$ for 500 stimuli per category; Methods), categories known to yield clusters of neural selectivity in primate higher-level visual areas[6,7,9,11,41,42]. In All-TNNs, too, we observed smooth category selectivity maps (Fig. 2d, left panel, and Supplementary Fig. 13; smoothness is quantified using Fourier analysis in Supplementary Fig. 7; see Methods) with clusters of units that are selective for faces, places and tools (Fig. 2d, right panel, and Supplementary Fig. 5; note that the locations of clusters are not linked to spatial information in the image, because units in the last layer have

the entire image in their receptive field). Similar to the emergence of orientation selectivity in the first layer of All-TNNs, this topographic organization into category-selective regions is smoother for larger values of $\alpha$, requires training and stabilizes early on (Supplementary Fig. 14). In contrast, LCN control models, despite identical training, exhibit salt-and-pepper selectivity maps. CNNs, when unfolded analogously to All-TNNs, exhibit no smoothness (Fig. 2d).

Taken together, our modelling results demonstrate that All-TNNs consistently reproduce key characteristics of primate neural topography in early and higher-level visual cortex, including V1-like smooth orientation maps, an emphasis on processing of information-rich parts of the visual input and smooth clustering based on category-specific neural tuning in the final model layer.

## All-TNNs allocate energy to task-relevant input regions

Despite a more primate-like topography, All-TNNs do not show increased classification performance compared with CNNs and LCNs. This observation raises the question of the (computational) benefits of topography. In primates, topographic organization is suggested to reduce energy costs[35,43,44]. To test whether this is the case in All-TNNs, we applied a measure of energy consumption from Ali et al.[45], which quantifies how much energy would be consumed by the network if it were implemented using biological neurons. This measure considers costs evoked by both 'synaptic transmission' (operationalized as the absolute value of pre-ReLU activations) and 'firing rates' (operationalized as the post-ReLU activity; Methods). We quantified this energy consumption for network activations in response to the test set of ecoset. Investigating the summed energy consumption of all network units at different stages of training, we found that CNNs use over one order of magnitude more energy than All-TNNs and LCNs (Fig. 3a; CNN energy consumption, $2.186 \times 10^{11}$; 95% CI, ($2.112 \times 10^{11}$, $2.260 \times 10^{11}$); at 35 epochs of training; two-sided permutation test, $n = 1 \times 10^{4}$, $P < 1 \times 10^{-3}$, BH-FDR-corrected). In addition, All-TNNs show a significant improvement in energy efficiency compared with LCNs. This improvement increases as $\alpha$ increases (LCN energy consumption, $2.112 \times 10^{10}$; 95% CI, ($2.099 \times 10^{10}$, $2.125 \times 10^{10}$); All-TNNs ($\alpha = 1$)

energy consumption, $2.012 \times 10^{10}$; 95% CI, $(2.007 \times 10^{10}, 2.017 \times 10^{10})$; All-TNNs ($\alpha = 10$) energy consumption, $1.351 \times 10^{10}$; 95% CI, $(1.340 \times 10^{10}, 1.362 \times 10^{10})$; All-TNNs ($\alpha = 100$) energy consumption, $7.716 \times 10^{9}$; 95% CI, $(7.637 \times 10^{9}, 7.795 \times 10^{9})$; at 35 epochs of training; two-sided permutation test, $n = 1 \times 10^{4}$, $P < 9 \times 10^{-2}$, BH-FDR-corrected).

To analyse the energy consumption of the models across their 2D sheets, we visualized the spatial distribution of energy expenditure relative to the centre unit (Fig. 3b) as well as summed across layers (except the final layer, because it has no spatial information due to pooling; Fig. 3c). These results indicate that All-TNNs expend more energy in the centre of the visual field, an effect that scales with $\alpha$. LCNs show this effect to a lesser extent, CNNs even less so (see Supplementary Fig. 15a for an additional visualization of the CNN energy expenditure in the original convolutional space). Moreover, when trained on shifted input data, All-TNNs exhibit an equally shifted energy expenditure (Fig. 3c and Supplementary Fig. 15b; see also Fig. 2c and Supplementary Fig. 11). In line with the aforementioned ability of All-TNNs to adapt to regions of the visual field that contain task-relevant information, this suggests that All-TNNs focus energy expenditure on those regions.

### All-TNNs better align with human spatial visual behaviour

Human visual behaviour exhibits spatial biases, such as improved classification performance for locations where objects are most often experienced[15,16,46,47]. The highly structured selectivity maps of All-TNNs open the possibility that they, too, may mirror such behavioural biases. We tested for this possibility by first empirically quantifying spatial biases in human visual object recognition and subsequently testing how far All-TNNs and control models reproduce these biases after training.

To quantify how well our models align with human behaviour, we collected data from 30 human participants, who were asked to classify 80 objects from 16 classes of the Common Objects in Context (COCO) dataset[48] (Fig. 4a). Stimuli were presented for 40 ms in a random location in a $5 \times 5$ grid, followed by a Mondrian mask[49] (Fig. 4b and Methods). Masking prevents ceiling effects in performance and has been proposed to limit recurrent processing in humans[49,50]. It is therefore ideal as a test bed for our current set of feedforward models. We quantified object recognition performance for each object class at each location, yielding spatial classification accuracy maps for each participant (Fig. 4c, Supplementary Fig. 16 and Methods). To test for category specificity, we computed an accuracy dissimilarity matrix (ADM), akin to representational similarity analysis[51] but based on the spatial accuracy distributions of the categories tested (ADMs were computed on the accuracy maps of each category averaged across participants; Fig. 4d and Methods; see Supplementary Fig. 17 for statistical comparisons of all category pairs). An analysis of the behavioural ADM using unique variance decomposition revealed significant effects of animacy (explained variance, 0.108; $P = 0.005$) and real-world size (explained variance, 0.039; $P = 0.036$). Hence, our data are consistent with previous research indicating that humans have category-specific spatial biases (Fig. 4d, Supplementary Fig. 18 and Methods).

To estimate the behavioural alignment between models and humans, we presented our ANN models (All-TNNs, LCNs and CNNs) with similar stimuli from the same categories as our human participants (Methods; note that all model types are in principle able to pick up spatial regularities in the training data; see Supplementary Fig. 19). In addition, we tested supervised and self-supervised versions of topographic deep ANNs (TDANNs), a recent CNN-based model extended to simulate topographic maps (Supplementary Fig. 20 and Methods; an in-depth analysis of TDANN topographic maps has been carried out previously[31]). For all models, we computed spatial classification accuracy maps, similarly to the human data (Methods and Supplementary Fig. 16). Note that the ANNs were tested on a selection of 16 categories × 500 objects from COCO images instead of the selection of 16 categories × 5 objects that humans saw (Methods), as this allowed

us to yield more reliable accuracy maps. The models were not shown Mondrian masks, as they are feedforward.

To quantify the extent to which the models can capture visual biases observed in our human participants, we first compared human accuracy maps (that is, variations in human object categorization performance across the visual field) with those of our models. For this, we computed the average category-wise Pearson correlation between the accuracy maps of individual model instances and the average human $\alpha$ accuracy maps (Fig. 4e, left; CIs were bootstrapped with replacement across participants, and the group-level noise ceiling was calculated using split-halved subject data with Spearman–Brown correction, $\rho = 0.869$; Methods). Correlations and statistics based on individual participants yield similar results and are provided together with the corresponding noise ceiling (leave-one-participant-out) in Supplementary Figs. 21 and 22b. All-TNNs with $\alpha = 10$ correlate with the human data at $\rho = 0.306$ (95% CI, (0.304, 0.308)), and CNNs correlate at $\rho = 0.096$ (95% CI, (0.095, 0.098)), a significant difference (two-sided permutation test, $n = 1 \times 10^{4}$, $P < 1 \times 10^{-3}$, BH-FDR-corrected; $n = 5$; Fig. 4d, right; see Supplementary Fig. 22a for all statistics). All-TNNs with $\alpha = 10$ also outperform LCNs ($\rho = 0.235$; 95% CI, (0.234, 0.237); two-sided permutation test, $n = 1 \times 10^{4}$, $P < 1 \times 10^{-3}$, BH-FDR-corrected; $n = 5$) and TDANNs (supervised TDANN: $\rho = 0.085$; 95% CI, (0.083, 0.086); self-supervised TDANN: $\rho = -0.077$; 95% CI, (−0.079, −0.075); two-sided permutation test, $n = 1 \times 10^{4}$, $P < 1 \times 10^{-3}$, BH-FDR-corrected; $n = 5$; see Supplementary Fig. 20a for statistics including TDANNs). There is a small albeit significant difference between All-TNNs with $\alpha = 10$ and $\alpha = 1$ ($\rho = 0.292$; 95% CI, (0.291, 0.294)), with $\alpha = 10$ outperforming $\alpha = 1$ (two-sided permutation test, $n = 1 \times 10^{4}$, $P < 1 \times 10^{-3}$, BH-FDR-corrected; $n = 5$). All-TNNs with $\alpha = 100$ lead to higher agreement with human data ($\rho = 0.333$; 95% CI, (0.331, 0.335)) than those with $\alpha = 10$ (two-sided permutation test, $n = 1 \times 10^{4}$, $P < 1 \times 10^{-3}$, BH-FDR-corrected; $n = 5$). Together, this pattern of results reveals that applying a spatial smoothness loss during training improves alignment with human behaviour. In fact, the best All-TNN ($\alpha = 100$) achieves a more than threefold increase in alignment with human behaviour over CNNs (Supplementary Table 3). An All-TNN trained on a shifted dataset performs significantly worse in terms of behaviour than regularly trained All-TNNs, showing that, when the topography is shaped by unnatural statistics, it does not match human spatial biases well (Supplementary Fig. 11f,g).

The better match of All-TNNs with human accuracy maps could be due to the fact that they better capture category-dependent human spatial biases or that they better capture category-independent human spatial biases (such as a centre bias). To differentiate these possibilities, we compared the ADMs of humans and models (Fig. 4d). Comparing two ADMs via Spearman correlations allowed us to focus on differences between accuracy maps rather than shared features between accuracy maps, such as a central bias. Indeed, shared features would show up as constants in the ADM (that is, all ADM cells are impacted identically by shared aspects), and such constant offsets do not affect Spearman correlations. As shown in Fig. 4f (right), ADM agreement between human data and All-TNNs with $\alpha = 10$ ($\rho = 0.185$; 95% CI, (0.170, 0.199)) is significantly higher than that for all other models, including CNNs ($\rho = 0.020$; 95% CI, (0.009, 0.031); two-sided permutation test, $n = 1 \times 10^{4}$, $P < 1 \times 10^{-3}$), LCNs ($\rho = 0.081$; 95% CI, (0.072, 0.091); two-sided permutation test, $n = 1 \times 10^{4}$, $P < 1 \times 10^{-3}$), All-TNNs with $\alpha = 1$ ($\rho = 0.070$; 95% CI, (0.057, 0.083); two-sided permutation test, $n = 1 \times 10^{4}$, $P < 1 \times 10^{-3}$), All-TNNs with $\alpha = 100$ ($\rho = 0.153$; 95% CI, (0.134, 0.173); two-sided permutation test, $n = 1 \times 10^{4}$, $P < 1.6 \times 10^{-2}$) and TDANNs (supervised TDANN $\rho = -0.002$; 95% CI, (−0.012, 0.008); self-supervised TDANN $\rho = 0.112$; 95% CI, (0.098, 0.125); two-sided permutation test, $n = 1 \times 10^{4}$, $P < 1 \times 10^{-3}$; Supplementary Fig. 20b). This suggests that the visual classification behaviour of All-TNNs with $\alpha = 10$ (and $\alpha = 100$ to a slightly lesser extent) best aligns with human behaviour in a category-specific manner (human noise ceiling estimated

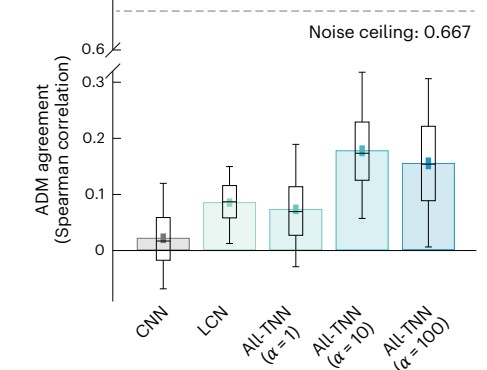

**Fig. 4 | All-TNNs better approximate spatial biases in human visual behaviour.**
**a**, To create stimuli for the behavioural experiment, objects from COCO were segmented from their background and placed on a 5 × 5 grid (16 object categories, 5 object exemplars). **b**, Each trial started with a variable fixation period (FP), followed by a brief stimulus presentation at one of 25 screen locations, after which a Mondrian mask was presented. Subsequently, a response screen showed the 16 target category labels. **c**, Human accuracy maps for all 16 categories, averaged across participants ($n = 30$; the maps are normalized by dividing by the maximum accuracy). **d**, Left, ADMs capture the dissimilarities between category-based accuracy maps (Pearson correlation distance). Object categories with similar spatial biases have low ADM dissimilarity. Middle, hierarchical clustering analysis on the averaged human ADM reveals structure in human spatial biases. Right, non-negative least-squares general linear model (GLM) analysis shows significant unique variance explained by animacy and real-world size, but not object 'spikiness' (the ratio between perimeter and surface area; Methods and Supplementary Fig. 18; asterisks note significant variance explained, $P < 0.05$). **e**, Left, model agreement with humans was computed

as the average Pearson correlation between human accuracy maps (averaged across participants, $n = 30$) and accuracy maps of each model ($n = 5$). Right, All-TNNs with increasing $\alpha$ exhibit significantly better alignment with average human accuracy maps, outperforming LCNs and CNNs. The error bars show 95% CIs. The group-level human noise ceiling was calculated using split-halves with Spearman–Brown correction. All comparisons are significant after BH-FDR correction (two-sided permutation test, $P < 1 \times 10^{-3}$; see all $P$ values in Supplementary Fig. 22a). **f**, Left, Spearman correlation between the average human ADM and the ADMs of the models ($n = 5$). Right, All-TNNs ($\alpha = 10$ and $\alpha = 100$) exhibit significantly better alignment with human ADMs than control networks. The error bars show 95% CIs. The group-level human noise ceiling was calculated using split-halves with Spearman–Brown correction. All comparisons are significant except between the LCN and All-TNNs ($\alpha = 1$) after BH-FDR correction (two-sided permutation test, all significant, $P < 1.6 \times 10^{-2}$; see all $P$ values in Supplementary Fig. 22a). In the box plots in **e** and **f**, the midline represents the median, the bounds indicate the 25th and 75th percentiles, and the whiskers extend to the 5th and 95th percentiles.

at $\rho = 0.667$), rather than merely reflecting a category-independent bias effect. Indeed, All-TNNs with $\alpha = 10$ exhibit a more than ninefold improvement in alignment with the human object-specific spatial biases compared with CNNs (an overview of all models is provided in Supplementary Table 4). To test for the robustness of our results, we reran all our behavioural analyses while varying the size of stimuli as inputs to our ANN models. We found that All-TNNs generally outperform control models across a range of sizes (Methods and Supplementary Fig. 23).

In summary, All-TNNs exhibit category-dependent spatial biases that are significantly better aligned with human behaviour than control models. In comparison, CNNs show weaker spatial biases, as expected given that they detect the same features at all spatial locations (the remaining spatial biases that CNNs exhibit can be explained by their fully connected readout layer that does not have weight sharing and can rely on different spatial locations present in the previous layer). LCNs are also able to latch on to category-specific biases, which is expected as they lack weight sharing. Still, both behavioural analyses performed (accuracy map agreement and category-specific accuracy effects via ADMs) indicate that All-TNNs ($\alpha = 10$) outperform LCNs. This suggests that the topographic features of All-TNNs play a role in increasing their behavioural alignment with human spatial visual biases.

### Self-supervised learning in All-TNNs

Recent work suggests that ANNs trained with self-supervised objectives can yield representations that align well with brain data[52,53]. To test for this in All-TNNs ($\alpha = 10$), we trained a set of networks using SimCLR[54], a self-supervised contrastive learning objective, to investigate how this influences the topography and behaviour of our model. After the self-supervised learning phase, we froze the network weights and trained a linear categorization readout for ecoset on their final layer (Methods), enabling us to quantify categorization performance and apply our behavioural analyses. Our SimCLR-trained All-TNNs achieve a slightly higher classification accuracy than category-trained All-TNNs ($\alpha = 10$; Fig. 5a, left; performance, 0.410; 95% CI, (0.408, 0.412); two-sided permutation test, $n = 1 \times 10^4$, $P < 1 \times 10^{-3}$, BH-FDR-corrected) and slightly increased smoothness between the weights of neighbouring units (Fig. 5a, right). They also develop largely similar topography to category-trained All-TNNs both for orientation selectivity (Fig. 5b) and category selectivity (Fig. 5c; see Supplementary Fig. 24 for all topography visualizations and smoothness quantification), including the increased feature variety in regions with relevant information (Fig. 5d). Interestingly, SimCLR-trained All-TNNs exhibit lower energy expenditure than their category-trained counterparts (Fig. 5e). However, SimCLR networks have to be trained with substantially larger data augmentation than their category-trained counterparts, complicating the interpretation of differences between the two training objectives.

In addition to in silico network analyses, we tested the SimCLR-trained All-TNNs ($\alpha = 10$) against the human behavioural data. SimCLR-trained All-TNNs are again better correlated with human accuracy maps ($\rho = 0.271$; 95% CI, (0.269, 0.273)) than the CNNs, LCNs and TDANNs described above (two-sided permutation test, $n = 1 \times 10^4$, $P < 1 \times 10^{-3}$ in all comparisons, BH-FDR-corrected; Supplementary Fig. 20a; see Supplementary Fig. 22c for statistics without TDANN), and they are slightly but significantly worse (two-sided permutation test, $n = 1 \times 10^4$, $P < 1 \times 10^{-3}$, BH-FDR-corrected) than category-trained All-TNNs (Fig. 5g). However, while the SimCLR self-supervised objective was sufficient to achieve similar topography and category-independent spatial biases as category-trained All-TNNs, the behavioural agreement of SimCLR-trained All-TNNs does not seem to be driven by category-specific spatial biases, as they do not show a good match to human ADMs ($\rho = -0.014$; 95% CI, (−0.029, 0.001); Fig. 5h and Supplementary Figs. 20b and 22c).

## Discussion

Here we introduce and test a new kind of topographic neural network model, All-TNN, that is trained end-to-end to perform complex visual behaviour on natural image inputs. All-TNN units have local receptive fields without weight sharing, are arranged on a 2D cortical sheet and develop spatially smooth feature selectivity through the use of a loss encouraging similar-weight kernels in neighbouring units. These features allow All-TNNs to develop genuine end-to-end topography while harnessing the benefits of deep learning.

Analysing the topography of All-TNNs using in silico electrophysiology, we found that, when trained on natural images, the models develop key features reminiscent of both lower- and higher-level areas of the primate visual cortex, including smooth orientation selectivity maps in the first layer and category-specific selectivity clusters in the last layer. CNNs do not show these characteristics, as expected due to weight sharing. LCNs, which differ from All-TNNs only by the absence of the smoothness loss, do not develop these topographies either. This lends support to the idea that topography in the visual cortex may result from the tendency of spatially neighbouring neurons to learn similar features, in line with experiments showing that neurons with similar response properties cluster together in primate visual cortex[55,56]. It also shows that the spatial statistics of natural images are themselves insufficient to yield smooth topography.

All-TNNs develop a centre–periphery spatial organization with enhanced processing in the centre of the visual field, demonstrated by more varied feature selectivity and increased energy expenditure. This observation was consistent across random seeds, model layers and various analysis types. By design, CNNs cannot exhibit such a distribution of resources. The spatial organization observed in All-TNNs was less pronounced in LCNs, suggesting that All-TNNs' architectural bias towards smoothness plays a role in shaping this focused processing. Under the influence of the smoothness loss, the network develops large peripheral regions with redundant selectivities. We found that we could lesion units in these predominantly peripheral regions without strongly impacting classification performance. Once lesioned, All-TNNs have more functional units per degree of visual angle in the central part of the visual field. This is reminiscent of cortical magnification, a key topographic feature of primate primary visual cortex, which exhibits more neurons per degree of visual angle in regions processing foveal information. Future work may test whether such spatially structured changes in unit density can be accomplished as part of the training procedure—for example, by introducing additional energy efficiency loss terms. In addition, exploring image transformations, akin to retinal sampling or retinopy-inspired transformations[57,58], will be of interest to study its effect on the topographic organization in All-TNNs. It is conceivable that applying spatially specific sampling to the input data will render the model selectivity to be spatially more homogenous, in line with experimental data from macaques, suggesting that ganglion cell density can account for the cortical magnification factor[59]. Importantly, the observed centre–periphery spatial organization in the networks is not driven by an architectural centre bias but rather by which locations in the input contain task-relevant information, as shown by the fact that shifting the training dataset correspondingly shifts the region with enhanced processing. The spatial organization in All-TNNs thus balances the need for fine-grained processing of relevant regions of the visual field while favouring smooth feature selectivities in less crucial regions. Relatedly, we interpret the emergence of category-selective clusters as satisfying a balance between the need for varied feature detectors and the tendency to have smooth selectivities, supporting the hypothesis that the high-level organization in inferior temporal cortex balances feature variability and homogeneity[12].

LCNs and All-TNNs with different $\alpha$ values achieve similar classification performance, raising questions about the computational benefits of the kind of topographic organization found in All-TNNs and primate brains. One possibility is that topography enables more

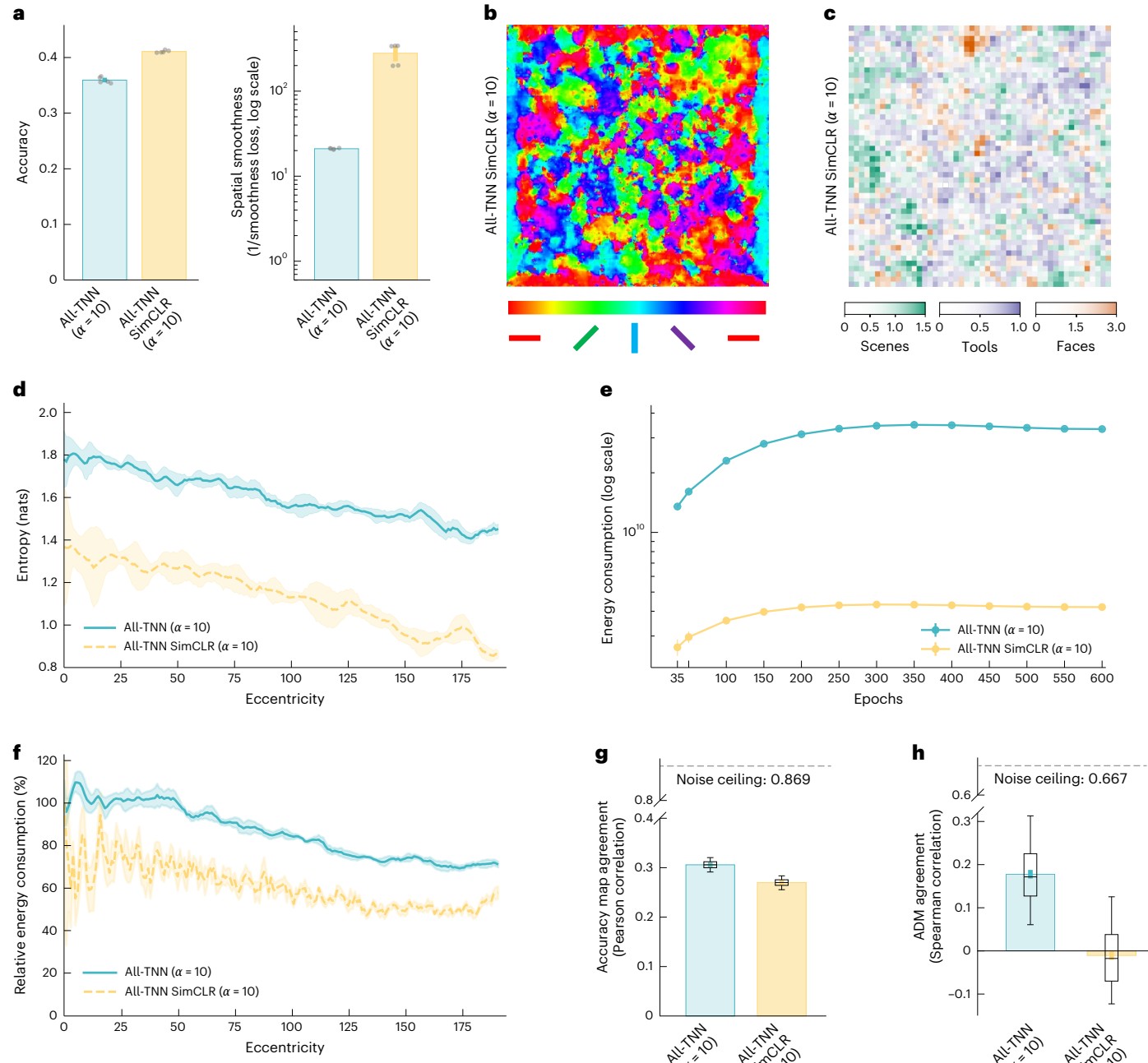

**Fig. 5 | Self-supervised All-TNNs.** We trained five All-TNN instances with different random seeds ($\alpha = 10$) using the SimCLR self-supervised training objective (Methods) and compared them with the supervised-trained All-TNNs ($\alpha = 10$). The error bars and shaded areas indicate the 95% CIs. Group-level human noise ceilings were calculated using split-halves with Spearman–Brown correction. **a**, Left, classification accuracy on the ecoset test set. Right, the spatial smoothness of the weights averaged over the network layers ($n = 5$). **b**, Orientation selectivity in the first layer of a SimCLR-trained All-TNN shows smooth and clustered orientation-selective units (see Supplementary Fig. 24 for all seeds and quantification of smoothness and clustering). **c**, Category selectivity analysis in the last layer of a SimCLR-trained All-TNN shows clustering of units selective for scenes, tools and faces (measured using $d'$; Supplementary Fig. 24). **d**, Orientation selectivity in the first layer of SimCLR-trained All-TNNs shows a centre region with higher feature density, similar to supervised All-TNNs. **e**, Energy consumption of SimCLR-trained All-TNNs is lower than that of supervised All-TNNs across training epochs ($n = 5$). **f**, Relative energy consumption in the first layer of SimCLR-trained All-TNNs decreases with increasing eccentricities, similar to supervised All-TNNs. **g**, SimCLR-trained All-TNNs show lower accuracy map agreement with human accuracy maps than supervised All-TNNs (see Supplementary Fig. 22c for statistical tests; 30 participants, five seeds for each model). The error bars show 95% CIs. **h**, ADMs of SimCLR-trained All-TNNs do not agree well with human ADMs (see Supplementary Fig. 22c for statistical tests; 30 participants, five seeds for each model). The error bars show 95% CIs. In the box plots in **g** and **h**, the midline represents the median, the bounds indicate the 25th and 75th percentiles, and the whiskers extend to the 5th and 95th percentiles.

energy-efficient processing[35,43,44]. Using a proxy for energy consumption in biological networks, we found CNNs to be the least energy efficient, probably because weight sharing does not allow spatially

specific feature selectivity. LCNs are the second worst, suggesting that a lack of smoothness is also detrimental to energy consumption. All-TNNs were the most energy efficient, focusing energy consumption

in areas with the most task-relevant information. This suggests that the smoothness loss used in All-TNNs may help limit energy consumption, consistent with the impact of metabolic constraints on topography considered elsewhere[35,43,44].

We found that the topographic organization of All-TNNs arises early during training and stabilizes quickly, while the network continues to increase its classification performance. This indicates that the topographic organization offers a stable structure while still allowing for enough flexibility for learning. This is similar to the primate visual cortex, where early maturation of important architectural structures is thought to provide scaffolding for functional selectivity[11,60–62]. This furthermore ties into an important debate about which aspects of the visual system's structure are genetically hardwired, and which require experience[60,63]. In our experiments, both input statistics and architectural constraints (smoothness) were needed for the formation of smooth topography, as it was neither present prior to training nor present in LCNs. One question for future research concerns whether further aspects, beyond or instead of external input, are sufficient to bring about this topographic organization. For example, could phenomena such as retinal waves also drive the emergence of a topographic scaffold[64]? Another avenue for future research may be to model the development of topographic organization in other modalities[65] using All-TNN-inspired approaches.

The fact that All-TNNs develop spatial biases that can be tied to human visual behaviour opens exciting new research directions. For example, a first hypothesis to test would be whether, if certain objects have more behavioural relevance than others during training, they take up more space in the topography, which in turn may account for biases in behaviour[66,67] and better discriminability for frequently seen stimuli[68]. All-TNNs could enable the study of further aspects of human visual behaviour, such as the interactive effects of scene background on object perception, or multi-object perceptual phenomena. As another example, the spatial topography of All-TNNs allows for targeted lesioning of their organized parts, such as the face-selective units in the later layers. This allows using All-TNNs to model brain lesions[69] with potential for clinical impact, or to model virtual lesioning methods such as transcranial magnetic stimulation[70], providing insights into the underlying mechanisms and effects of such experimental interventions.

In line with other work demonstrating the power of self-supervised learning for training models of primate vision[52,53], we found that training All-TNNs ($\alpha = 10$) using the self-supervised SimCLR training objective resulted in broadly similar topographies and matching to category-specific spatial biases in human behaviour as the supervised All-TNNs. This shows that the majority of our findings reflect general principles that do not depend on a specific training objective. However, SimCLR-trained All-TNNs did not perform well in matching category-specific spatial biases in human behaviour. This may be due to the more extensive image augmentations inherent to the SimCLR objective, which may disrupt object-specific spatial statistics. Alternatively, it may be due to the fact that there is no categorical readout during SimCLR training (as it is a self-supervised objective, and none of the image augmentations contain category information), and a diagnostic categorical readout is trained only after the SimCLR-trained weights are frozen (note that the data augmentation during fine-tuning is similar to the augmentations applied while training category-trained networks). Future research is needed to clarify circumstances in which self-supervised objectives can yield human-like spatial biases in behaviour in an All-TNN setting. For example, a focus on augmentations that do not affect positional information could be explored.

All-TNNs complement other recent approaches to modelling topography in the visual system[31–35,71–74], which have contributed to our understanding of cortical map formation. Existing models can be seen as forming two broad families. The first relies on simple features, sometimes hand-crafted, which form the basis for topographic organization—for example, using self-organizing maps[2,34,75,76], sparsity

optimization[77,78] or other self-supervised methods[79]. These models usually have no weight sharing, but, due to their simplicity, they are unable to perform complex tasks such as categorizing natural images, limiting their domain of applicability. The second approach is to augment CNNs—for example, by adding a spatial remapping to their units[31,32], building self-organizing maps based on unit activities[33], adding topographic layers after a CNN backbone[35] or creating multiple CNN streams[73,74]. By leveraging the computational advantages of CNNs, these models can perform complex tasks while still being mappable to topographic features. We strongly agree that CNNs can provide important insight into functional organization. However, the reliance on weight sharing remains biologically implausible. In addition, the fact that these models are ultimately based on CNNs means that they necessarily extract identical features across the visual field. In other words, CNN units mapped on a virtual 2D topographic sheet still remain CNN units, rendering it more challenging to study the impact of topography on behaviour. Accordingly, we found that TDANNs[31], a recent representative of this approach that aligns well with primate topographies, do not match human behaviour in our dataset as well as All-TNNs. However, given the importance of dataset statistics in our results, this comparison is limited by the fact that the available TDANNs have a different training dataset and architecture than our models. Further careful model contrasting[18] is needed to more properly compare the two model classes.

The current work has several limitations. First, the smoothness loss that we use to encourage neighbouring units to learn similar features may not capture the mechanism of smoothness in topographic organization in the brain. Future work using All-TNNs as a starting point can explore how smooth maps emerge naturally from model training, without imposing a secondary smoothness constraint. Possible avenues include using more biologically plausible constraints, such as wiring length optimization[34,35,80], energy constraints[45], cortical size[2,81] or recurrent connectivity patterns[82,83]. Second, the analyses in our work focus on topography in the early and late layers, since the tuning of intermediate regions in the brain is less clearly understood. Future research into visualizing and quantifying the features learned by All-TNNs in intermediate layers may help in studying representations in intermediate regions of visual cortex as well. Third, the current work focuses on tightly controlled comparisons between models with identical architecture, datasets and objectives[18]. A large-scale comparison of the different topographic deep neural network models in existence, controlling for architecture and dataset, will be important to determine the strengths, weaknesses and applications of each approach. Lastly, All-TNNs do not outperform LCNs in terms of categorization performance, despite the smoothness loss acting as a potential regularizer that could have prevented overfitting. This may be due to the (symmetric) smoothness loss limiting the ability of the network to change poor unit weights. Future work may address this issue—for example, by considering non-symmetric definitions of the smoothness loss to favour units with useful weights or weight updates. Finally, although All-TNNs outperform the CNN gold standard by a large margin, a substantial gap remains towards accounting for all variance in human spatial visual biases even if considering our best models. In response to this observation, we note that our comparisons of model and human spatial behavioural patterns entail zero free parameters (similar to representational similarity analysis). If parametric flexibility in the mapping were allowed (as seen in encoding models), the fit could be improved. However, we are convinced that the more conservative approach taken here is the stronger test for behavioural agreement. In terms of model improvements, multiple avenues of future work promise to further close the gap. First, all models are trained on static images of ecoset. While more ecologically relevant than common computer vision datasets (such as ImageNet), the input images still differ from the typical human visual diet. Moving towards more natural input statistics (including spatial, temporal and active aspects of vision) is therefore an

important aspect of future work. Moreover, explorations of different task objectives, changes in dataset size, model scale and the addition of recurrent connectivity motifs (such as lateral and top-down connectivity) should all be tested for their effects on the learned topographies and the resulting model's agreement with human spatial biases.

In conclusion, All-TNNs represent a promising new class of models that address questions that are beyond the scope of CNNs. They could serve as more accurate models of functional organization in the visual cortex and its behavioural consequences.

## Methods

### Neural network architectures and training

**All-TNNs.** All-TNNs are implemented on the basis of LCNs—that is, ANNs that are identical to CNNs except that no weight sharing is performed (that is, weight kernels are independent across space). In practice, this is implemented by subclassing tensorflow's LocallyConnected2D layer, which is arranged in a height × width × channels 3D structure.

To convert this 3D locally connected layer to our 2D All-TNN layer, we reshaped the height × width × channels layer to a height$\sqrt{\text{channels}}$ × width$\sqrt{\text{channels}}$ 2D sheet. This corresponds to 'unfolding' each channel to a square of units of size $\sqrt{\text{channels}} \times \sqrt{\text{channels}}$ that have the same receptive field but can learn varied feature selectivity. The relative spatial positions of these unfolded channels are preserved.

The central feature that differentiates All-TNNs from LCNs is that All-TNNs have an additional smoothness loss, which promotes similar selectivity for neighbouring units and is crucial for the emergence of topography (Figs. 1 and 2). In detail, this smoothness loss ($\mathcal{L}_{\text{S}}$) is computed as the average cosine distance between the weight kernels of neighbouring units in each layer. To avoid border effects that might introduce an artificial centre bias, the loss calculation uses periodic boundary conditions (that is, the layer is wrapped in a toroidal fashion). For example, the unit at the very bottom (or right) of the layer has its loss computed with the top-most (or left-most) unit (Supplementary Fig. 10). The total loss that the model aims to minimize is a composite of the categorization cross-entropy loss ($\mathcal{L}_{\text{CE}}$), equation (1), where $M$ is the number of classes, $p_c$ is the predicted probability of class $c$ and $y_c$ is the correctness of predicted class $c$) and the smoothness loss (summed over all layers; equation (2)). The smoothness loss is multiplied by a factor $\alpha$ that determines the additive weight of the smoothness loss:

$$\mathcal{L}_{\text{CE}} = -\sum_{c=1}^{M} \log(p_c)y_c \qquad (1)$$

$$\mathcal{L}_{\text{S}} = \sum_{l=1}^{L} \frac{1}{N_l} \sum_{n=1}^{N_l} \left( \frac{\text{cos\_dist}(w_{i,j}, w_{i+1,j}) + \text{cos\_dist}(w_{i,j}, w_{ij+1})}{2} \right) \qquad (2)$$

$$\mathcal{L}_{\text{total}} = \mathcal{L}_{\text{CE}} + \alpha\mathcal{L}_{\text{S}} \qquad (3)$$

where $w_{ij}$ is the flattened weight kernel of the unit at the position $i,j$ on the 2D sheet, $N_l$ denotes the total number of units in layer $l$, $\alpha$ is a hyperparameter that determines the magnitude of the smoothness loss and $L$ is the total number of layers in the network. The All-TNNs used here consist of six such layers. Layers 1, 3 and 5 are followed by 2 × 2 max pooling layers. See Supplementary Table 1 for the specific architectural details of all models. We trained five instances of each network with different random seeds, which are treated as experimental subjects.

Reliance on smoothness in the weights is different from previous approaches that have optimized for smoothness in the activity patterns of nearby units[31]. We here chose to focus on weights as this ensures that nearby units are selective to similar features, which also achieves correlated activity (Supplementary Fig. 25). The reverse is not true, however, as correlated activity does not necessitate similar feature selectivity. For example, it is conceivable that nearby units are

selective for different features that co-occur in the input statistics, and thus exhibit correlated activity. Note, however, that our smoothness loss could easily be adapted to optimize for correlated activity.

We compared three $\alpha$ values (with five seeds each) on a logarithmic scale (1, 10 and 100), as pilot studies showed that this provides a sensible range in terms of the emerging topographic features/smoothness. The choice to use three $\alpha$ values was driven by the large computational cost of training five All-TNN instances for each $\alpha$ value. In general, higher $\alpha$ leads to smoother topographies and similar categorization performance (but behavioural alignment with humans is best for intermediate $\alpha$; see 'Discussion'). Hence, there is no globally optimal $\alpha$, and we report results for all $\alpha$ values.

As shown in equation (2), the loss is computed between each unit and its 'east' and 'south' neighbours. We leave out the 'north' and 'west' neighbours because cosine distance is symmetric, so that if unit $(i,j)$ is similar to its eastern neighbour $(i,j+1)$, then unit $(i,j+1)$ is also similar to its western neighbour $(i,j)$. We do not include the 'southeast' diagonal $(i+1,j+1)$, because it is indirectly enforced to be similar to unit $(i,j)$. These unnecessary computations are omitted for computational efficiency. As a sanity check of these assumptions, we ran a control identical to our main All-TNNs with $\alpha = 10$, except that the loss is calculated across all eight neighbours (that is, isotropic and including diagonals; Supplementary Fig. 26). This network shows similar performance and topography as our main All-TNNs, albeit with increased smoothness in the selectivity maps (as expected since smoothness is enforced between distant units, so exactly replicating our main All-TNNs would require tuning $\alpha$ differently).

**Locally connected control model.** LCNs have layers in which each unit is connected only to a local subset of the input units, as in CNNs, but without weight sharing. This contrasts with fully connected networks, where each neuron is connected to all input neurons. All weights are separately learnable—that is, LCNs are identical to All-TNNs without a smoothness loss. As a control for the effect of the smoothness loss, we trained five LCNs with the same number of layers, number of units, hyperparameters and dataset as our All-TNNs but without smoothness loss ($\alpha = 0$; that is, trained only on the categorization cross-entropy loss).

**Convolutional control model.** We trained five convolutional control networks with the same number of layers, number of units, hyperparameters and dataset as our All-TNNs. The smoothness loss is not (and cannot be) enforced in this model due to the weight sharing. It is thus trained with only the categorization cross-entropy loss, in line with the current gold standard in the field.

**SimCLR-trained All-TNNs.** To test the importance of category-training to our results, we trained five instances of All-TNNs ($\alpha = 10$) using Sim-CLR (a self-supervised, contrastive objective) with the same number of layers, number of units, hyperparameters and dataset as our supervised All-TNNs. SimCLR training is based on maximizing the correlation between the representations of two augmentations of an image. The applied augmentations consist of cropping (8–100%), aspect ratio changes (3/4–4/3), colour jittering (80%), and changes in hue (20%), saturation (30–70%), brightness (20%) and contrast (80%). After the self-supervised training phase, we froze the backbone of the network, discarded the SimCLR projectors and fine-tuned a readout layer to predict object categories on the ecoset training set. In this fine-tuning phase, the same augmentations were applied to the input images, but with lower strength (cropping (80–100%), aspect ratio changes (3/4–4/3), colour jittering (5%), and changes in hue (20%), saturation (10%), brightness (10%) and contrast (10%)).

**TDANNs.** We included TDANNs in our behavioural analyses[31]. TDANNs are a recent extension to CNNs that are spatially embedded to model functional topography. We downloaded the weights of

the ImageNet-trained networks made available by the authors (five SimCLR-trained instances and one category-trained instance). We evaluated these models according to our procedures for the behavioural experiment, identically to our other models (see 'Behavioural experiment and analyses').

**Training dataset.** All-TNNs and all control models were trained using ecoset[24]. Ecoset consists of 1.5 million images from 565 ecologically relevant categories. Previous work showed that networks trained on ecoset better predict activities in human higher visual cortex than networks trained on ImageNet (ILSVRC-2012), making it a good choice for modelling the influence of natural image statistics on the emergence of cortical topography.

**Model training.** Each model was trained on images with 150 × 150 pixels. All models were trained with an early stopping criterion to prevent overfitting (that is, training was stopped when the validation loss started to increase; Supplementary Table 2). All results reported in this paper use the checkpoints for the epoch fixed by this early stopping criterion, except for Fig. 3a, which shows several epochs for each model. The models were trained with common image augmentations (brightness, 12.5%; saturation, 50%; contrast, 50%; cropping, 33–100%) and scaling, but excluding left–right flip, as this would influence spatial biases present in the image dataset. Each layer was subject to L2 weight regularization with a factor of $1 \times 10^{-6}$, followed by ReLU nonlinearity and layer normalization. We used an Adam optimizer with a learning rate of 0.05, $\epsilon = 0.1$ and a regularization ratio of $1 \times 10^{-6}$. Weights were initialized with Xavier initialization. All models are custom-made, implemented via Python v.3.10 (Python Software Foundation, https://www.python.org/) with tensorflow v.2.12 (https://www.tensorflow.org/), and trained using NVIDIA H100 GPUs.

**Shifted training dataset.** To study the effect of the spatial dataset statistics during training on topography in All-TNN, we applied a shifting augmentation to ecoset during training (Fig. 2c). We shifted the dataset to the right and down by 20% of the image size (30 pixels) and padded the now-empty upper left with zeros.

**Model analyses**
**LCN and CNN topography visualization.** The figures depicting feature maps for LCNs and CNNs were made using the same technique for 'unfolding' of the channels as was used for All-TNNs. This allows for comparable visualization of topography between models. While All-TNNs use this unfolded 2D map to compute the smoothness loss during training, this is not the case for CNNs and LCNs.

**Quantifying smoothness in trained models.** The model smoothness in All-TNNs is defined as 1/smoothness loss. To compare the global weight smoothness of All-TNNs with that of control models, we first determined the smoothness between units exactly like the calculation of smoothness loss for All-TNNs during training (see 'Neural network architectures and training') and then summed it over all units in all layers.

**Orientation selectivity.** To determine the orientation selectivity of units in the first layer of the models, we presented grating stimuli at eight angles (equally spaced between 0 and 180 degrees) to the models and recorded the elicited activity (post-ReLU activation function). The gratings had a spatial wavelength of three pixels, allowing for more than one cycle within each receptive field in the first layer of our networks. For each grating angle, and for each network unit, we presented various phases and picked the phase that maximized the unit's activation response to the grating, to find the best alignment between the stimulus and weight kernel. We combined the resulting eight activities (one per angle) vectorially projecting each angle onto

a circle and multiplying it by the corresponding activity, and taking a weighted sum of these vectors (a widespread method for measuring orientation selectivity in electrophysiology[84]). To visualize the orientation maps (as well as the entropy and category selectivity maps) for LCNs and CNNs, the kernels were flattened into a 2D sheet in an identical manner as for All-TNNs (see 'LCN and CNN topography visualization').

**Category selectivity.** To determine high-level object selectivity in the last layer of the models, we selected 500 images for each of three categories: faces, places and tools. The places and tools images were selected from the ten most common categories for the respective superclass in the ecoset test set (see below). The faces were taken from the VGG-Face dataset[85].

Places: 'House', 'City', 'Kitchen', 'Mountain', 'Road', 'River', 'Jail', 'Castle', 'Lake', 'Iceberg'
Tools: 'Phone', 'Gun', 'Book', 'Table', 'Clock', 'Camera', 'Cup', 'Key', 'Computer', 'Knife'
Faces: First ten identities taken from the VGG-Face dataset[85]

We computed the selectivity of each unit in the last layer of our networks to images of scenes, tools and faces using the $d'$ signal detection measure:

$$d' = \frac{|\mu_{in} - \mu_{out}|}{\text{mean}(\sigma_{in}, \sigma_{out})} \tag{4}$$

where $\mu_{in}$ and $\sigma_{in}$ are the mean and variance of the activations of the unit in response to stimuli of the category of interest, and $\mu_{out}$ and $\sigma_{out}$ are the mean and variance of the activations of the unit in response to stimuli from the other categories. The variances for both distributions are assumed to be in a similar range; therefore, we averaged over the variance of the two distributions in the denominator.

**Spatial feature variability and lesion study.** To quantify the diversity of selectivities at each location in the first layer of our models, we calculated Shannon entropy (using scipy.stats.entropy) in an $8 \times 8$ sliding window for each retinotopic position on the orientation selectivity maps. Units that did not respond to any of the grating stimuli were excluded from this analysis. Preferred orientations, computed as described above, were discretized into eight equally spaced orientation bins, which were used to compute the entropy (measured in nats). This yielded a map showing how varied unit responses are at each location (for example, the entropy is low if all units in the sliding window have similar preferred orientations).

To test whether the network can afford to lose units in regions with lower entropy (that is, more redundant coding), we performed an entropy-based lesion experiment. We used the entropy map described above, lesioned the 25% lowest-entropy units and measured the performance of the lesioned network on the ecoset test set. We also performed the same test, but lesioning the 25% highest-entropy units to test for the impact of losing units in areas with the most feature variability.

**Clustering and smoothness of the orientation and category selectivity maps.** To quantify the amount of clustering in the orientation selectivity maps, we computed the average radius needed to go from each unit to include units with all eight orientation selectivities. To determine cluster size for the category selectivity maps of the final layer, we assigned each unit to one of the three category bins (tools, places or faces) according to its highest category $d'$. We then followed the same procedure over this binned category selectivity map as with orientation selectivity. To statistically compare cluster sizes between model types, a two-sided independent $t$-test was performed with BH-FDR correction with $P < 0.05$.

As another way to quantify aspects of topography, we also performed a 2D discrete fast Fourier transform on the orientation

selectivity maps and $d'$ category selectivity maps. The intuition is that smooth orientation maps will show higher power in the lower frequency domain, while salt-and-pepper maps will lean towards the higher frequencies. We shifted the zero-frequency component to the centre of the spectrum and extracted the radial profile of the power spectrum, such that we could visualize the power in each spatial frequency bin. To summarize the faces, scenes and tools $d'$ maps in one figure, we took the average power spectrum of these three maps.

**Pinwheel analysis.** We quantified pinwheels in model orientation selectivity maps as done elsewhere[84] by detecting the units that had high variation in orientation selectivity in their direct Moore neighbourhood (the eight direct neighbour units). This is determined by calculating the 'winding number'—that is, accumulating the angular differences between the selectivities of pairs of neighbours. A high absolute winding number indicates high continuous variation in orientation selectivities and can thus be classified as pinwheel, while a low winding number indicates that the unit in question is in a region with low variation. Negative winding numbers are counterclockwise-turning pinwheels, while positive winding numbers indicate clockwise-turning pinwheels. This analysis is sensitive to hyperparameter tuning and noise, which can lead to multiple pinwheels appearing near each other; in these cases we considered only the unit with the highest winding number in the Moore neighbourhood as being the centre of the pinwheel.

**Energy efficiency.** We quantified a proxy for metabolic energy consumption in ANNs, based on a combination of the preactivation of units and their postactivation output as presented in previous work[45]. The presynaptic energy consumption ($E_{\text{pre}}$) is the energy required to transmit information at the synapses of biological neurons and is quantified here using the pre-ReLU activation of units. The post-synaptic energy consumption ($E_{\text{post}}$) is the energy associated with transmitting action potentials and is quantified here using the post-ReLU activation of units.

The overall energy consumption is given by the equations:

$$E = \frac{2}{3}E_{\text{pre}} + \frac{1}{3}E_{\text{post}} \tag{5}$$

$$E_{\text{pre}} = \sum_{l=1}^{L}\sum_{n=1}^{N_l} |W_n||X| \tag{6}$$

$$E_{\text{post}} = \sum_{l=1}^{L}\sum_{n=1}^{N_l} |A_n| \tag{7}$$

where $A$ is the post-ReLU activation of a unit, $W$ represents the unit weights and $X$ represents the inputs to this unit. $L$ represents the number of layers in the neural network, and $N$ denotes the number of units within a specific layer. The 2/3 versus 1/3 split is based on previous work[45].

For analysis and visualization of the spatial distribution of energy expenditure, all layers were interpolated to be the same size as the first layer. The relative energy consumption per eccentricity was calculated by setting the energy consumption of the unit at the image centre in the unfolded layers as the maximum and measuring how the average energy consumption changed as a percentage of this maximum with increasing eccentricity.

**Predefined spatial prior.** Objects in natural environments display category-specific spatial priors (positional frequency of occurrence in visual space). However, evaluating whether models can capture these spatial statistics is challenging due to the photographer's centre bias present in most image datasets. To test how well all models can capture

spatial priors, we constructed a toy training dataset with predefined spatial locations to evaluate the ability of models to latch on to spatial patterns in object locations. The object categories in this dataset were selected from our behavioural stimuli and placed on natural scene backgrounds randomly selected from the Place365 dataset[86]. As shown in Supplementary Fig. 19, we defined nine simplified, category-specific spatial priors for the nine categories on 5 × 5 maps. In these maps, one position has a 62.5% occurrence frequency, while the remaining 24 positions equally share 37.5% (1.5625% each).

**Object size variation.** To investigate the degree to which our results depend on object presentation size, we extended the behavioural analyses by testing models on the behavioural stimuli with object sizes scaled to 50%, 75%, 100%, 125%, 150%, 175% and 200%, allowing for a comparative study of different models' alignment under size effects (Supplementary Fig. 23). Moreover, we report classification accuracy on the ecoset test set while scaling the input to 50%, 75%, 100%, 125%, 150%, 175% and 200% of the original size. The scaled-down images are white-padded.

### Behavioural experiment and analyses

**Participants.** A total of 30 healthy adults (aged 21–30 years; mean, 25.47 years; s.d., 2.5 years; 17 female) participated and completed the visual classification task. All participants had normal or corrected-to-normal visual acuity. Prior to participation, ethical approval for the study was approved by the ethics committee of the Department of Education and Psychology of the Freie Universität Berlin to ensure compliance with ethical guidelines. All participants provided written informed consent and received monetary compensation or course credits for their participation.

**Experiment design.** The participants were asked to detect and classify the object presented on screen. The images on which the stimuli occurred were 215 × 215 pixels, calculated to equate to a five-degree visual angle and a 65-cm screen distance. Stimuli randomly occurred in one of the 25 positions on a 5 × 5 grid on the screen for 40 ms. The stimuli were then masked with a Mondrian mask, presented for 300 ms. Subsequently, the participants were presented with a response panel showing the 16 category names. They were given 2,150 ms to click on the category name that matched what they had perceived before. Feedback was given after each trial: the correct category name was displayed in a green font colour if the participants were correct and in red if they were incorrect. The experiment consisted of 2,000 trials in total: each object exemplar was shown one time in each location (that is, for each of the 16 categories, all 5 object exemplars were presented once in each of 5 × 5 locations). The stimulus order was randomized between participants. Each trial took around 2.5 s, and after each set of 200 trials, a two-minute pause was introduced. The total experiment took three hours.

**Stimuli for the human behavioural experiment.** The spatial classification behaviour of humans was evaluated using segmented objects from the COCO dataset[48]. This is a large-scale image dataset, gathered from everyday scenes containing common objects. COCO provides segmentation masks for each of a set of target objects in a given scene, rendering it highly useful for curating a dataset. The stimulus set used for behavioural testing consists of 16 categories that occur in both ecoset and COCO: 'airplane', 'bear', 'broccoli', 'bus', 'cat', 'elephant', 'giraffe', 'kite', 'laptop', 'motorcycle', 'pizza', 'refrigerator', 'scissors', 'toilet', 'train' and 'zebra'. Each category contains five exemplars, selected for having similar illumination. All images were resized to equal bounding box size (215 × 215 pixels), cropped onto grey backgrounds and visually checked to represent coherent objects. Mondrian masks used in the human experiment were generated by the generation script made available by M. Hebart[49,87].

**Data recording and processing.** The stimulus presentation in the behavioural experiment was controlled using the Psychtoolbox[88] in MATLAB. For each object category, participant and presentation location, we computed the average classification performance, yielding a set of category- and participant-specific spatial accuracy maps.

**Behavioural experiment for the models.** To collect behavioural responses from the models, we present the trained models with segmented COCO stimuli presented on a 5 × 5 grid as for our human experiment. In contrast to the human experiment, where we used only five stimuli per class due to time limitations, we screened all COCO stimuli to gather 500 valid stimuli per class to better estimate the performance of our ANNs. When creating this dataset, we found that certain segmented COCO object segmentations are extremely small, covering only a few pixels, and add noise to our results because ANNs cannot classify them. To mitigate this issue, we discarded objects that were smaller than 20 pixels in their largest dimension in the 640 × 480 COCO images. In addition, pilot experiments on our models showed that resizing COCO objects to fit in a single bounding box of the 5 × 5 grid led to very low accuracy at most locations. To mitigate this issue, we allowed stimuli to be larger by padding the 5 × 5 grid with blank space. This allowed segmented COCO objects to cover up to 3 × 3 grid space (64 × 64 pixels) without extending beyond the image, while retaining clear 5 × 5 grid locations. Segmented COCO objects were resized to 15/64th of their original size, and objects larger than 205 pixels in COCO were discarded, to ensure they did not extend beyond the limit of 3 × 3 grid space.

**Hierarchical clustering.** To highlight structure in the human visual biases measured in our behavioural experiment, we computed the ADM (see 'ADM agreement') between the category-averaged human accuracy maps using Pearson correlation distance. We performed agglomerative hierarchical clustering on this ADM using the UPGMA (unweighted pair group method with arithmetic mean) method, which we used to reorder the matrix on the basis of the found clusters and to visualize the resulting dendrogram.

**GLM fit.** To further characterize structure present in the human data, we applied a GLM using non-negative least squares to the ADM (see 'ADM agreement') constructed using the category-averaged human accuracy maps. We defined several predictors on the basis of the object categories present in the behavioural stimuli (Supplementary Fig. 18). These predictors consist of animacy, real-world object size and object 'spikiness', which are all factors thought to be represented in IT cortex[71,89]. We based object size on the size ranking reported by Konkle and Oliva[90] and computed the spikiness of all segmented objects of the human behavioural stimuli using the aspect ratio as described in Bao et al.[71], where $P$ is the object's perimeter and $A$ the object's surface area:

$$\text{Aspect ratio} = \frac{P^2}{4\pi A} \qquad (8)$$

We computed the unique variance explained by each model predictor by subtracting the total variance explained of the reduced GLM that excludes the predictor of interest from the total variance explained by the full model. To estimate the null distribution of variances for each predictor, we shuffled the rows and columns of the ADM and refit the GLM. We repeated this 1,000 times to determine whether the found explained variance was significant ($P < 0.05$).

**Accuracy map agreement.** To quantify the alignment of spatial biases in visual behaviour between humans and models, we directly measured the agreement between their accuracy maps for each category. We computed the Pearson correlation coefficient between the accuracy maps of humans (averaged across participants) and models for each of our 16 object categories. The mean correlation score was calculated across these 16 categorical accuracy maps and five seeds for each model. The error bars in the figures show the 95% CI of the mean correlation between models and bootstrapped average humans (bootstrap 30 participants from 30 participants with replacement, $n = 1 \times 10^3$). We calculated the group-level noise ceiling of the human data using split-halves and Spearman–Brown correction (see 'Noise ceiling calculation'). In addition to the accuracy map agreement method applied throughout the main text, we also performed this analysis by correlating all model accuracy maps with each individual human accuracy map (as opposed to using the average map across human participants; Supplementary Figs. 21 and 22b), for which we could calculate a participant-level noise ceiling (see 'Noise ceiling calculation'). Significance was tested using two-sided permutation tests, using BH-FDR correction with $P < 0.05$.

**ADM agreement.** We computed ADMs to quantify the dissimilarity between pairs of accuracy maps of different categories, in a similar spirit to the well-known representational dissimilarity matrices[51]. Objects yielding distinct accuracy maps have high dissimilarity in the ADM. ADMs were created on the basis of Pearson correlation distance. To quantify the agreement between humans and models, we first computed the human ADM on the basis of accuracy maps averaged across 30 participants with bootstrapping. In addition, for each model type, we computed five ADMs (one for each seed). Finally, we computed the average Spearman correlation across seeds between model ADMs and the human ADM. Significance was tested using two-sided permutation tests, using BH-FDR correction with $P < 0.05$. A higher ADM correlation indicates a higher alignment of the structure of class-wise positional dependencies while ignoring positional dependencies that are shared across maps.

**Noise ceiling calculation.** To estimate the reliability of the human data, we calculated the group-level noise ceilings for both the accuracy map and ADM analyses. We did so by calculating the correlation (Pearson correlation for the accuracy map analysis and Spearman correlation for the ADM analysis) between the two groups after split-halving the data of the 30 participants. To account for biases resulting from the split-half calculations, we performed a Spearman–Brown prophecy correction, in line with previous work[71]. We report these noise ceilings in the main behavioural results (Figs. 4 and 5).

To estimate the consistency in behaviour across participants (Supplementary Fig. 21), we computed an accuracy map or ADM noise ceiling by iteratively leaving out one human and seeing how well it correlated with the average accuracy map or ADM of the remaining 29 participants. This yielded 30 correlations, of which we report the mean (this is technically referred to as the lower bound of the noise ceiling).

## Reporting summary

Further information on research design is available in the Nature Portfolio Reporting Summary linked to this article.

## Data availability

The data required to reproduce our results are available via OSF at https://osf.io/6m3g4/ and via GitHub at https://github.com/Kietzmann-Lab/All-TNN/. Ecoset can be downloaded from https://huggingface.co/datasets/kietzmannlab/ecoset.

## Code availability

All analyses of human and model data were performed with custom Python software using the tensorflow, numpy and scikit-learn packages, among others. The code required to reproduce our results is available via GitHub at https://github.com/KietzmannLab/All-TNN/.

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

## Acknowledgements

We acknowledge support from the following grants: A.D. is supported by SNF grant no. 203018; T.C.K., V.B. and A.D. are supported by ERC STG grant no. 101039524 TIME; D.K. is supported by the Deutsche Forschungsgemeinschaft (SFB/TRR 135, project no. 222641018), an ERC Starting Grant (no. ERC-2022-STG 101076057) and 'The Adaptive Mind', funded by the Excellence Program of the Hessian Ministry of Higher Education, Science, Research and Art; Z.L. is supported by CSC grant no. 202106120015; and R.M.C. is supported by DFG grants no. CI241/1-1, no. CI241/3-1 and no. CI241/7- 1 and ERC STG grant no. 803370. Computing resources for this project are in part funded by the Deutsche Forschungsgemeinschaft (DFG, German Research Foundation), project no. 456666331. The funders had no role in study design, data collection and analysis, decision to publish or preparation of the manuscript.

## Author contributions

Z.L., A.D., V.B., B.K., D.K., R.M.C. and T.C.K. conceived and designed the experiments. Z.L., A.D. and V.B. performed the experiments. Z.L., A.D., V.B. and T.C.K. analysed the data. Z.L., A.D., V.B., B.K., D.K., R.M.C. and T.C.K. contributed materials or analysis tools. Z.L., A.D., V.B., D.K., R.M.C. and T.C.K. wrote the paper. T.C.K. oversaw all parts of the research.

## Competing interests

The authors declare no competing interests.

## Additional information

**Correspondence and requests for materials** should be addressed to Zejin Lu or Tim C. Kietzmann.

# Reporting Summary

## Statistics

For all statistical analyses, confirm that the following items are present in the figure legend, table legend, main text, or Methods section.

| n/a | Confirmed | |
|---|---|---|
| ☐ | ☒ | The exact sample size (*n*) for each experimental group/condition, given as a discrete number and unit of measurement |
| ☒ | ☐ | A statement on whether measurements were taken from distinct samples or whether the same sample was measured repeatedly |
| ☐ | ☒ | The statistical test(s) used AND whether they are one- or two-sided *Only common tests should be described solely by name; describe more complex techniques in the Methods section.* |
| ☐ | ☒ | A description of all covariates tested |
| ☐ | ☒ | A description of any assumptions or corrections, such as tests of normality and adjustment for multiple comparisons |
| ☐ | ☒ | A full description of the statistical parameters including central tendency (e.g. means) or other basic estimates (e.g. regression coefficient) AND variation (e.g. standard deviation) or associated estimates of uncertainty (e.g. confidence intervals) |
| ☐ | ☒ | For null hypothesis testing, the test statistic (e.g. *F*, *t*, *r*) with confidence intervals, effect sizes, degrees of freedom and *P* value noted *Give P values as exact values whenever suitable.* |
| ☒ | ☐ | For Bayesian analysis, information on the choice of priors and Markov chain Monte Carlo settings |
| ☒ | ☐ | For hierarchical and complex designs, identification of the appropriate level for tests and full reporting of outcomes |
| ☐ | ☒ | Estimates of effect sizes (e.g. Cohen's *d*, Pearson's *r*), indicating how they were calculated |

*Our web collection on statistics for biologists contains articles on many of the points above.*

## Software and code

Policy information about availability of computer code

| Data collection | Behavioural data collected with Matlab Psychtoolbox 3, ANNs trained with Python v.3.10 with Tensorflow v.2.12, and trained using NVIDIA H100 GPUs. |
|---|---|
| Data analysis | Python in-house code, Numpy (v.1.23.5) and Scikit-learn (v.1.4.0) packages. The code required to reproduce our results can be found on GitHub at https://github.com/KietzmannLab/All-TNN/. |

For manuscripts utilizing custom algorithms or software that are central to the research but not yet described in published literature, software must be made available to editors and reviewers. We strongly encourage code deposition in a community repository (e.g. GitHub). See the Nature Portfolio guidelines for submitting code & software for further information.

## Data

Policy information about availability of data

All manuscripts must include a data availability statement. This statement should provide the following information, where applicable:
- Accession codes, unique identifiers, or web links for publicly available datasets
- A description of any restrictions on data availability
- For clinical datasets or third party data, please ensure that the statement adheres to our policy

All analyses of human and model data were performed in custom Python software, making use of Tensorflow, Numpy, and Scikit-learn packages among others. The

## Research involving human participants, their data, or biological material

Policy information about studies with <u>human participants or human data</u>. See also policy information about <u>sex, gender (identity/presentation), and sexual orientation</u> and <u>race, ethnicity and racism</u>.

| | |
|---|---|
| Reporting on sex and gender | We report the sex of our participants in text (17 female out of 30 participants). |
| Reporting on race, ethnicity, or other socially relevant groupings | N/A |
| Population characteristics | Participant age 21-30 years, mean=25.47 years, SD=2.5 years. |
| Recruitment | Participants were recruited via departmental student email lists at FU Berlin. As a result, self-selection biases are possible. We expect our results to generalize to similar student populations. All participants have normal or corrected-to-normal vision. |
| Ethics oversight | Ethics committee of the Department of Education and Psychology of the FU Berlin. |

Note that full information on the approval of the study protocol must also be provided in the manuscript.

# Field-specific reporting

Please select the one below that is the best fit for your research. If you are not sure, read the appropriate sections before making your selection.

☒ Life sciences   ☐ Behavioural & social sciences   ☐ Ecological, evolutionary & environmental sciences

For a reference copy of the document with all sections, see <u>nature.com/documents/nr-reporting-summary-flat.pdf</u>

# Life sciences study design

All studies must disclose on these points even when the disclosure is negative.

| | |
|---|---|
| Sample size | Our sample size (n=30) was selected based on running previous experimental paradigms, see e.g. Kaiser, Daniel, and Radoslaw M. Cichy. "Typical visual-field locations facilitate access to awareness for everyday objects." Cognition 180 (2018): 118–122. Given the extensive data collected per participant (2000 trials), our sample size is adequate to ensure reliable and generalizable findings. |
| Data exclusions | No participants or models were excluded from the analyses. |
| Replication | We trained multiple seeds per ANN model type (n=5 per model type) to ensure that the modelling results are not the result of random sampling error. All training models yielded consistent results. |
| Randomization | Stimulus sequence in the behavioural paradigm was randomised. |
| Blinding | Not applicable: stimulus presentation and scoring are fully computer-automated, responses are objective (correct/incorrect), and analysis is executed via a scripted pipeline. Experimenters have no influence on trial outcomes. |

# Reporting for specific materials, systems and methods

We require information from authors about some types of materials, experimental systems and methods used in many studies. Here, indicate whether each material, system or method listed is relevant to your study. If you are not sure if a list item applies to your research, read the appropriate section before selecting a response.

## Materials & experimental systems

| n/a | Involved in the study |
|---|---|
| ☒ | ☐ Antibodies |
| ☒ | ☐ Eukaryotic cell lines |
| ☒ | ☐ Palaeontology and archaeology |
| ☒ | ☐ Animals and other organisms |
| ☒ | ☐ Clinical data |
| ☒ | ☐ Dual use research of concern |
| ☒ | ☐ Plants |

## Methods

| n/a | Involved in the study |
|---|---|
| ☒ | ☐ ChIP-seq |
| ☒ | ☐ Flow cytometry |
| ☒ | ☐ MRI-based neuroimaging |

