## [Peer Review File · Nature Human Behaviour]

End-to-end topographic networks as models of cortical map formation and human visual behaviour

Corresponding Author: Professor Tim Kietzmann

A version of this paper was originally rejected for publication by Nature Human Behaviour, however that decision was reconsidered after appeal by the authors.

Version 0:

Decision Letter:

12th October 2023

Dear Professor Kietzmann,

Thank you once again for your manuscript, entitled "End-to-end topographic networks as models of cortical map formation and human visual behaviour: moving beyond convolutions", and for your patience during the peer review process.

Your Article has now been evaluated by 3 referees. You will see from their comments copied below that, although they find your work of potential interest, they have raised quite substantial concerns. In light of these comments, we cannot accept the manuscript for publication in the current form, but would be interested in considering a revised version if you are willing and able to fully address reviewer and editorial concerns.

We hope you will find the referees' comments useful as you decide how to proceed. If you wish to submit a substantially revised manuscript, please bear in mind that we will be reluctant to approach the referees again in the absence of major revisions. We are committed to providing a fair and constructive peer-review process. Do not hesitate to contact us if there are specific requests from the reviewers that you believe are technically impossible or unlikely to yield a meaningful outcome.

In our view, the most important issue which must be addressed is the point raised by Reviewer #1 (and also Reviewer #2) regarding the parallels between features of the all-TNN, and the human visual system. The existence of such parallels is central to the scientific impact of this work, but the reviewers are not fully convinced by the evidence currently presented. We therefore request that you a) conduct further analyses to investigate the existing parallels (e.g. orientation maps in early layers; category maps in later layers; increased complexity in central parts of the visual field) and b) extend your study to consider additional parallels with other features of the human brain. We believe that to satisfy the reviewers, b) will be important as well as a).

If you wish to submit a suitably revised manuscript, we would hope to receive it within 4 months. I would be grateful if you could contact us as soon as possible if you foresee difficulties with meeting this target resubmission date.

- Include a "Response to the editors and reviewers" document detailing, point-by-point, how you addressed each editor and referee comment. If no action was taken to address a point, you must provide a compelling argument. When formatting this document, please respond to each reviewer comment individually, including the full text of the reviewer comment verbatim followed by your response to the individual point. This response will be used by the editors to evaluate your revision and sent back to the reviewers along with the revised manuscript.
- Highlight all changes made to your manuscript or provide us with a version that tracks changes.

Link Redacted

Thank you for the opportunity to review your work. Please do not hesitate to contact me if you have any questions or would like to discuss the required revisions further.

Sincerely,

██████████

████████████████████

████████████████████

Nature Human Behaviour

REVIEWER COMMENTS:

Reviewer #1:

Remarks to the Author:

The authors introduce All-TNN model architectures, which remove the weight-sharing convolutional assumption of traditional CNN architectures, creating all topographic layers. They trained these models on a joint loss function with object classification and a spatial tuning similarity loss. They argue to show emergent smooth orientation tuning and clustered category-selectivity. And, they compare object recognition capacity across the visual field, with a new human behavioral experiment and parallel analysis on their models. They claim these models make an important step towards understanding the spatial organization of the visual brain.

Overall, I found the work to be well-written, and I think all-tnns are important and approach topography in a different and powerful way, compared to TDANNs and SOMs. They clearly enable the authors to ask new questions about the position-dependence of learned tuning and category information, and they even go on to do initial tests of these ideas in this paper. I'm strongly in support of these directions and the novelty of the approach and its contributions.

I have several comments on the methods, results, and claims which I think can clarify, qualify, and bolster the empirical and theoretical claims. These involve fixing up some terminological confusions, explaining your commitments about the alpha and spatial loss calculations, and optionally doing some controlled rearing variations over alphas/spatial-loss neighborhoods or deeper explorations of accuracy-smoothness tradeoffs, clarifying its inferential value for to foveal sampling and cortical magnification in the biological system, quantifying the topography results and/or qualifying the claims a little, and reporting more of the noise ceiling of the human data so we can get a clearer understanding of the magnitude/reliability of position-dependent recognition effects and item-level variation. (Also, pre-emptively signing: review by Talia Konkle)

1- Questions about the All-TNN model implementation

A- Alpha hyperparameters

Figure 1 is very helpful, and in general your methods are very clear. To understand the results, I needed more details about the hyperparameter alpha values on the smoothness of the layers, how they were explored, selected, and how that impacts the outcomes. For example, the methods state: "we used a spatial loss of $\alpha=10$ in all layers except the final, which for which we used $\alpha = 10$, due to increased smoothness observed in the higher visual cortex". A couple typos on there, and also the same alpha value, so, hard to follow. *How were these alpha values set and how do they matter?* There is room to do a controlled rearing experiment here, with a suite of models varying in alpha that you could characterize. It also sounds like you needed to increase the alpha of late stages to make it 'blobbier'? But maybe a supplemental figure just showing the range, from low and salt-and-pepper, to extremely high (e.g. very blobby), would be valuable. As it is now, it is just a little confusing what your commitment is to that alpha value, across layers, and what your search procedure for setting those was, especially as I assume your main topographic results depend on this parameter setting.

B- Spatial similarity loss

The way you implemented tuning spatial similarity loss, as far as I can tell, is that for each unit you compute the kernel weights correlation with the unit next to it to the right, and one below it, then you sum this. Then, you sum this across all units. If I got it correctly, this neighborhood function seems a bit asymmetric and small to me (just the next neighbors, and only two of them rather than, say, all units within some distance on the cortical sheet, including diagonal units). I know this is a huge space and you had to pick something, but I admit this choice of just two adjacent units surprised me a bit. I suspect that variations in the way you operationalize this L_s , e.g. larger neighborhood, would have big effects on the kinds of results you'd get. So, an explanation of why this choice (was it on purpose, and you explored others and this was good for reasons, or was this just a start) would help me as a reader, both in the methods/results and in general discussion. These implementation (along with the Alpha's) really seem like the key decisions you had to make and discussing them, and your commitments to them, would strengthen this work and clarify how future work could build off it.

C- Object classification results and smoothness

I found that I really needed a results section, after you describe the model, but before you talk about layer 1 orientation maps, that reports the results of the model training and smoothness objectives! (across all $N=30+$ models you have). What is the average object classification accuracy within a model type (and was there a lot of variance across the 10 seeds?). Next, you can even quantify the emergent smoothness in the LCNs (!!) and ask: what is the L_s of these models? (helpful in the context of the all-tnn models that are optimized for this directly). This just reports that your models trained well and did what you expected (and provides context for the next comment).

D – Accuracy-smoothness tradeoff:

You assert that the spatial similarity loss “forces the network to trade off between i) learning varied feature selectivity... and ii) preserving similar features selectivity between neighboring units.” However this claim is never shown or quantified. You could directly test for accuracy-smoothness trade offs: if you varied alpha, does the level of smoothness (Ls) really trade off with accuracy? (For a richer exploration, you could even let the Alpha’s-by-layer be learnable, and see what values are learned!). These analyses/variations are important, because they get at the theoretically important question of whether topographic tuning smoothness can be a useful inductive bias to help promote classification accuracy in the huge space of possible solutions. Or, alternatively, if in this way of operationalizing the cortical formation as a joint loss function of categorization and smoothness, these constraints directly trade off.

I recognize there’s a range of possible action items in the comments above. At a minimum, I would suggest reporting the classification accuracy of all the models and their spatial smoothness loss, either by varying alpha directly, or at least by describing your process for hyperparameter search over alphas. And, either test whether these trade off and by how much, or remove that assertion.

E- “Hypercolumn” – I believe you are using “hypercolumn” incorrectly throughout the manuscript, especially as you introduce this in the results section. A hypercolumn is a vertical column through the laminar structure of the cortex, within a single cortical area. These neurons along the depth axis of the cortex tend to have the same RF and tuning profiles. However, you’re using it more like a way to refer to the hierarchical construction of receptive fields, where the increasing receptive field of a unit in layer 2 is linked directly to the connections from layer 1. This is not a hypercolumn structure. This is, maybe, a ‘hierarchical spatial receptive field motif’ to try to give it a name. Definitely remove “hypercolumn” and use a different term for this.

2. Topographic features of the ventral stream

A - Lovely orientation results. The main thing that lacked for me is quantification. You talk about orientation ‘smoothness’ but, e.g. didn’t necessarily quantify pinwheels or degree of emergent smoothness, compared to LCNs and CNNs. One idea: what’s the average radius you need to go from a unit to include units with all 8 preferences? This will also allow you to quantify that for All-TNNs the radius will vary as a function of center-to-edge, but won’t for LCN vary in an eccentricity dependent way for LCNs.

B – Cortical magnification claims are incorrect.

The way you linked these results to the ‘foveal bias’ confused me—you claim that “cortical magnification emerges”. But, I think there is a deep mismatch here. To expand on this: your first model layer map has units which each see a FIXED size of the visual field. There is no oversampling at the center of the image through a fovea. And, the definition of cortical magnification is that a disproportionately large area of the visual cortex is devoted to processing information from the central region of the visual field (the fovea) compared to the peripheral regions. Thus, your architecture does not have cortical magnification. So, claims to this end need to be removed. What your model DOES show is that, given an architecturally EQUAL sampling of the visual input with even sized receptive fields across the ‘visual field’, that more tuning variation emerges for processing central image statistics than for peripheral image statistics. This outcome implies a more normative possible claim, which is that if one wanted to have similar ‘smoothness’ of orientation tuning across the cortical sheet, one would need to over-represent the fovea and under-represent the periphery, and that would presumably still capture the image statistics well for the task. So, I think it’s fair to claim that your map of orientation gestures at an implication like this, but does not show emergent cortical magnification.

C– Category-selectivity results.

Unfortunately, when I look at Figure 2, layer 6 visualization, I don’t easily see a ‘face-selective area’ or a ‘scene-selective area’. E.g. these do not look like what you get when you make face-selective contrasts and can define face and scene region of OTC. I feel like this should be acknowledged. These maps are in the right direction (e.g. relative to LCNs), but claims of this map forming “category-selective areas” feel to me to be unwarranted by my inspection of the results. Either a case should be made, ideally in a quantified way, that these are ‘areas’ in some sense, or this claim needs to be significantly weakened. Certainly, for example, the category-tuning clustering is more clumpy than in the LCN, again would be nice if this were quantified.

3. Behavioral Results

A – control models.

The LCN models absolutely need to be included in these experiments (Fig 4 and Fig 5). They are critical comparisons! All-TNNs should have the capacity to succeed, while LCNs should fail, based on your accounts. If the LCNs succeed, we need to know and understand those results! Similarly, the CCNs also succeed to some degree above chance it seems, at picking up on these spatial bias. Why? How? Do they? That seems worth discussing.

B - Data reliability.

As a first step of the data analysis, I would like to see the noise ceiling of the human data (before you verify these maps correspond to the objects occurrence statistics) For example, in Figure 3b, you show several templates for 16 objects, but I do not have a good sense from the results how much this variation across items is stable or noise. Can you do an analysis that quantifies this? some ideas: reporting the split-half reliability of the pattern? Maybe showing that a pattern is more similar to itself than other items on average across split halves? Maybe a linear mixed effects model statistic where you have to model accuracy with position as a predictor and a random effect of subject, and you can include item as a predictor as well, and item*position, and show that including item*position interactions are significant? Further, situating the results in Figure 4a and 5b with respect to the data reliability would help contextualize the results.

Edit: I just saw in the methods that you do in fact compute the noise ceiling and you are already showing reliability adjusted correlations! Definitely report the noise ceiling first please, and state clearly in the results you’re showing noise adjusted correlations. I frankly would prefer to see the raw correlation values with the noise ceiling of the data indicated on the same axes. If

the data are noisy, make the case for why the effect is important even if it's noisy, and/or qualify the results and point to important directions for future work. Don't hide the noise.

C - Shuffle control.

For figure 4, in addition to adding the LCNs, it would be useful to add a spatially shuffled prior as a control analysis, to get a sense of where the CNN is falling with respect to the shuffled control.

4. Discussion/Abstract

Based on the above comments, I recommend: qualifying the claims about the emergent organization of category-selective regions, clarifying the logic of the cortical magnification argument, and removing the claims about a trade-off between classification and smoothness unless this is directly shown with new analyses.

Minor/Optional/Potential small edits for increased clarity

1. Terminology: LCNs: "local" is confusing to me here, because I usually think of local like local WITHIN A MAP/AREA, local connectivity is near-neighbor (in the cortex) connectivity. I think maybe your LCNs could be called 'pure-hierarchy' models or some alternative?
2. Intro: "locality" desideratum: I'd say 'units need to have local receptive field *feature weights* that are learned individually and not enforced to be exact duplicates of other RFs'. This is to help clarify the difference between feature tuning (flexibly learned) and spatial receptive fields (fixed by the architecture)
3. Methods: Just checking: the cosine difference between the weight kernels, here this would be calculated over say, a flattened 27x27 set of learned weights if that is the size of the kernel? (rather than, say, over the activation profiles of a test set of images, which is more like what is done in the empirical literature noting smoothness, because we don't have access to the connectome weights). Clarifying this in the methods might be valuable.
4. Figure 2. How are you plotting CNN preference maps, as in 2a. Are you just spatializing the features in a systematic grid? Maybe say this in the caption?
5. Figure 2. You might clarify how you calculated entropy in a sentence in the main text, this term caught me off guard while reading through.
6. Figure 4A: it took me a while to parse this diagram. I think it's because the 16 model object maps are arranged vertically and the 16 human object maps horizontally in a matrix, but you're only doing the corresponding correlations (and not making and rdm? Unless I really missed something). Figure 4b, in truth I struggled with this analysis and figure.
7. Results order: I got confused by the presentation of behavior-to-object-occurrence statistics rather than behavior-to-model accuracy. Either these could be switched, OR, a little signposting ahead of time might help, because the results header really leans into the behavior-to-model accuracy as the key result.
8. ADMs results section and stereotypical unit results section. In truth I found these sections a little hard to follow. Maybe separating out logic of the analysis and methods details in different paragraphs could help?

Reviewer #2:

Remarks to the Author:

The authors propose the All-topographic Neural Network (All-TNN), an artificial neural network designed to better model the topographic organization of representations in the visual stream, as compared to CNNs and Locally Connected Networks (LCNs). The All-TNN relies on architectural design choices (notably, neurons are locally connected and arranged spatially in a 2D organization) and regularization (spatial similarity loss) in effort to satisfy three criteria, (1) neurons of the network have local receptive fields, (2) neurons are arranged on a cortical sheet, and (3) neural activity is topographically organized in ways similar to those observed in the human visual stream. In-silico experiments were run with All-TNNs and control ANNs (locally connected and convolutional neural networks) to evaluate emergent network properties following training on ecoset (specifically, cortical magnification, smooth orientation selectivity maps, and category selective regions). Human and artificial network visual behavioral bias comparisons are also evaluated to further link the topographic organization of All-TNNs.

Strengths:

1. The authors leverage generalizable constraints (e.g., local convolutions, similarity loss) which, in theory, could be adapted to more general ANN architectures to promote emergent brain-like topographic organization in the ANN. Common with the theme of much brain-like ANN research, these techniques can yield models (like the All-TNN) that suggest or provide evidence for/against theories of processing in the brain.
2. It is quite interesting that topographic organization arises at multiple levels in the All-TNN. Early on, the authors observed topographic similarities in the form of cortical magnification and orientation maps. Deeper in the network, categorical organization is emergent.
3. Emergent neural organization in All-TNNs was very different from that observed in the evaluated control models.
4. Multiple instantiations of each network were trained (with different random seeds), permitting more robust evaluation.
5. Authors ran multiple in-silico experiments and analyzed All-TANNs from multiple perspectives to demonstrate emergent topographic organization
6. Authors take the extra step to collect data from human subjects and compare the behavioral visual biases of All-TNNs and CNNs with humans.

Weaknesses/Questions:

1. The topographic neural networks proposed by Lee et al (2020) [Editors note - the referee has clarified that this refers to Lee, H., Margalit, E., Jozwik, K. M., Cohen, M. A., Kanwisher, N., Yamins, D. L., & DiCarlo, J. J. (2020). Topographic deep artificial neural networks reproduce the hallmarks of the primate inferior temporal cortex face processing network. *bioRxiv*, 2020-07], which similarly assumed 2D organization of neurons and demonstrates topographic organization, should be included in the evaluating and referenced appropriately as related work. Differences to Lee et al should be pointed out to clarify the novelty of the work.
2. The author's hypothesize that the center-periphery organization (e.g., cortical magnification) of the All-TNNs is likely emergent from the greater presence of object-relevant information in the center of the image. I am not fully convinced of this, especially since similar properties may emerge in LCNs if this were the case. Could this instead be an artifact of the spatial similarity loss? This loss function seems to suggest that central weight filters are implicitly regularized by all neighboring filters (above, below, left, and right), whereas border filters near the edge are not regularized by neighbors in all directions. The authors' claim may be supported better by demonstrating that this center-periphery organization emerges when trained on an augmented dataset that shifts the main image content outside of the image center or by training on a dataset where objects are not commonly centered in the image.
3. In the lesion studies (which aim to support the greater importance of foveal processing in All-TNNs), LCNs and CNNs do not have high-entropy and low-entropy regions. Meanwhile, the All-TNN high-entropy lesion region contains the image center, which has more object relevant information, and is therefore more likely to affect classification performance. If the authors were to lesion the same image portions and feed them to the CNN and LCNs, I would imagine more similar observations of post-lesion classification accuracy would be observed across the three models. A more fair comparison may show the accuracy of LCNs and CNNs when the same high-entropy region from the All-TNNs is used as the lesion area.
4. In the behavioral bias analysis, positional occurrence frequency and uncertainty are computed from COCO images and it is assumed that stereotypical object locations could be computed from these images. Images taken by humans, however, naturally have a bias towards centering an object of interest near the center of the image and may therefore not actually contain representative positional occurrence frequencies as would be observed through the eyes of a human.
5. This may simply be my misunderstanding, but further clarity on how 3D image representations (height, width, and channel) are "unfolded" into 2D neural sheets would be appreciated. The authors state that 3D representations of the form height x width x channel are unfolded into 2D representations of the form $(\sqrt{\text{channels}} \times \text{height}) \times (\sqrt{\text{channels}} \times \text{width})$. How are channel dimensions incorporated into height and width dimensions while preserving spatial information in the image representations?
6. The authors should provide architectural information about the CNN. They mention that the CNN had a similar number of parameters, but should also include information about filter sizes, channels, etc.
7. The authors should consider showing object classification accuracy of each evaluated model on ecoset. Although it is not a motivating aspect of this paper, it could help the audience better understand the applicability of these networks to more general computer vision tasks.

Reviewer #3:

Remarks to the Author:

In contrast to architectures based on convolution kernels, the authors propose an architecture that exploits the full richness of possible connections on a topographical and local space. The paper shows that this network achieves relatively good classification performance, while demonstrating the emergence of several properties that can be observed in the visual system of primates, and in human behavior in particular. In particular, the first layer exhibits a retinotopic-like organization, while the last categorization layer displays a category-selective patch-based organization.

The paper is well presented, and the results are original, justifying their publication in *Nature Human Behaviour*. Note that while I'm more of a specialist on the theoretical side, I consider myself less of an expert on the behavioral side. Although I find the analogy between the results of the model, the subjects and the priors found in the images very relevant in this study, I can only judge superficially the significance of the experimental results, specifically on the variability of the responses obtained. I recommend a minor but significant revision to increase the paper's impact.

First of all, a main point: through its supervised nature and being dedicated to categorization, this model stands out from similar models of self-organized learning in topographical maps. However, the emergent properties, particularly of the first layer, are similar to some of these models (such as <http://yann.lecun.com/exdb/publis/pdf/koray-cvpr-09.pdf>, doi:10.1016/S0042-6989(01)00114-6 or doi:10.1371/journal.pcbi.1010270). It would therefore be crucial to differentiate what makes this supervised model different from these unsupervised models of topographical self-organization. Indeed, a crucial difference lies in the definition of a cost function for the retinotopic organization observed in primates. Self-organizing models propose, for example, a metabolic-type optimization (e.g. of wiring length) or in relation to the efficiency of local information coding (e.g. sparseness). Your study could thus propose an alternative with a cost function linked to connection locality and categorization performance, as shown in your perturbation study. This point could be put forwards to discuss the origin and function of cortical topography.

I then have several secondary points:

- The analysis of the first layer is detailed, but is less precise for the following layers. Do you observe a similar organization? The origin of the variability of the topographies observed remains an enigma (cf doi: 10.1016/j.neuron.2021.09.053 for example) and your study could help to explain it.

- Concerning the parallel between the model's response and observed human behavior, the regressions are significant, but the behavioral responses are highly variable. It seems possible that other variables are involved, such as the effect of the intrinsic size of the visual objects in the images, e.g. that a bus is a priori larger than an animal, as is for instance demonstrated in Figure 3 (e.g. airplane, bus, train vs pizza, laptop or scissors).

- On the other hand, have you observed whether the network develops size invariance? This is a property inherent to the dataset

used, and should emerge as the network is trained.

- In order to make a controlled comparison of the performance of the convolutional network versus the proposed network, it would be useful to quantify the number of free parameters in each of the networks being compared.

- The definition of localization loss seems to derive from a heuristic (as the average of cosine similarity over horizontal and vertical directions) and its use needs to be justified. In particular, is this loss isotropic and does not introduce biases on the cardinals? Will other measures (e.g. the sum of the positive parts of cosine similarities) give similar results?

- The development of topographical structure is linked to the development of spontaneous waves of activity in the form of retinal waves, which help to structure topographical relationships in the network. Do you think you could observe similar waves of activity in a continuously activated network?

Minor:

- Figure 1, the RFs in layer #5 are not to scale,

- page 6: when giving the accuracy after low entropy lesion, it is indeed a "relative change in accuracy" (I guessed it from seeing it is over 100%). This is indeed explained later in the paper.

- I couldn't find a quotation in your paper describing the Mondrian mask synthesis method.

Version 1:

Decision Letter:

Dear Professor Kietzmann,

Thank you for submitting your revised manuscript entitled, "End-to-end topographic networks as models of cortical map formation and human visual behaviour".

After careful consideration, I regret that we do not feel it would be fruitful to send this manuscript back out to peer review at Nature Human Behaviour. There are two main reasons for this:

Firstly, we note that the publication of Margalit et al. in Neuron(1) which we feel reduces the scientific novelty of the current work. Nature Human Behaviour policy is that we do not consider novelty to be compromised by preprints, or by papers published during the agreed revision period, but we do take into account papers published outside this period. (1) was published on the 10th May 2024. This was outside of the agreed revision period for the current manuscript which, by my count, was last extended by one month in my email of 5th March 2024 meaning that the agreed period ended on 5th April.

Secondly, as stated in our previous decision letter, we felt that "the parallels between features of the all-TNN, and the human visual system... is central to the scientific impact of this work" and "We therefore request that you... b) extend your study to consider additional parallels with other features of the human brain". In this revision, we feel that overall the evidence for clear parallels between the all-TNN and the brain has been refined, but not extended.

I understand that this will be disappointing news - please let me know if you have any questions or concerns about this decision.

Although we cannot offer to publish your manuscript, my colleagues at Communications Psychology have agreed to send your work back to review if you were to transfer the manuscript to them. To transfer your manuscript please use our Link Redacted manuscript transfer portal. You will not have to re-supply manuscript metadata and files, unless you wish to make modifications. For more information, please see our http://www.nature.com/authors/author_resources/transfer_manuscripts.html?WT.mc_id=EMI_NPG_1511_AUTHORTRANSF&WT.ec_id=AUTHOR manuscript transfer FAQ page.

I am sorry that we cannot respond more positively on this occasion, and hope that the negative outcome in this instance will not deter you from submitting future work to Nature Human Behaviour.

Sincerely,

[Redacted]

[Redacted]

[Redacted]

Nature Human Behaviour

1) Margalit, E., Lee, H., Finzi, D., DiCarlo, J. J., Grill-Spector, K., & Yamins, D. L. (2024). A unifying framework for functional organization in early and higher ventral visual cortex. Neuron. 10th May 2024

Version 2:

Decision Letter:

5th December 2024

Dear Professor Kietzmann,

Thank you once again for your revised manuscript, entitled "End-to-end topographic networks as models of cortical map formation and human visual behaviour," and for your patience during the re-review process.

Your manuscript has now been evaluated again by the same reviewers who evaluated your original manuscript. All reviewer feedback is included at the end of this letter. Although the reviewers found your manuscript to have improved during revision, Reviewer #1 raised some important outstanding concerns.

You kindly indicated how you would respond to these concerns in an email exchange, and we agree that your proposed response would be promising.

Therefore, we remain very interested in the possibility of publishing your study in Nature Human Behaviour, but would like to consider your response to these outstanding concerns in the form of a further revised manuscript before we make a final decision on publication.

Finally, your revised manuscript must comply fully with our editorial policies and formatting requirements. Failure to do so will result in your manuscript being returned to you, which will delay its consideration. If you have any questions about any of our policies or formatting, please don't hesitate to contact me.

In sum, we invite you to revise your manuscript taking into account all reviewer and editor comments. We are committed to providing a fair and constructive peer-review process. Do not hesitate to contact us if there are specific requests from the reviewers that you believe are technically impossible or unlikely to yield a meaningful outcome.

We hope to receive your revised manuscript within 4-8 weeks. I would be grateful if you could contact us as soon as possible if you foresee difficulties with meeting this target resubmission date.

- Include a "Response to the editors and reviewers" document detailing, point-by-point, how you addressed each editor and referee comment. If no action was taken to address a point, you must provide a compelling argument. This response will be used by the editors and reviewers to evaluate your revision.
- Highlight all changes made to your manuscript or provide us with a version that tracks changes.

Link Redacted

We look forward to seeing the revised manuscript and thank you for the opportunity to review your work. Please do not hesitate to contact me if you have any questions or would like to discuss these revisions further.

Sincerely,

[Redacted]

[Redacted]

[Redacted]

Nature Human Behaviour

REVIEWER COMMENTS:

Reviewer #1 (Remarks to the Author):

Thanks to the authors for their extensive revisions and thoughtful care to my comments. I re-read the entire manuscript and again find the direction of all-TNNs and addressing the weight-sharing issue to be an important direction for the field. The revised manuscript is much clearer, and I appreciate the alpha sweep inclusion, and the steps taken revising the manuscript, following interpreting those results. The authors are also more careful in many of the claims.

I have only one remaining point. To me one of the strong points of the work is the potential to test for position-dependent categorization effects. The first step of that is understanding how stable those effects are in humans. Supplementary figure 21 shows the lower bound noise ceilings which are around .3 and .4. This is modest but definitely workable. But, if I'm reading these results right, the model correlations with these human data are substantially lower than the reliability in the human data. To the point where, perhaps the first take home claim might be that "none of the models are able capturing much of the stable human effects". And the second claim, along the manipulation of interest is that, looking at relative performance among the models, the all-TNNs slightly edge out the other models.

This matters, because the stated claim at the end of this section is: "In summary, All-TNNs capture category-dependent spatial biases that match those of human behaviour". And, if I'm reading these results right, I just don't think that is an accurate statement! I think All-TNNs have a slight edge compared to other models, when trained with this supervised objective. I certainly wish they nailed it, but at least you can still say that this is a new evaluation test of topographic models, that leave room for other improvements to close the gap.

Also, if you want to put a group-level human noise ceiling on your graphs in Figure 4E/F, you could estimate the group ceiling by split-halving your subjects, iterating over all split halves, and using the spearman-brown prophecy correction formula to account for the half-data problem.

Assuming my interpretations are correct here, I think it is important to acknowledge the fit of the models relative to what they could be giving the reliability of the data. If there's a gap, there's a gap. State it as a future puzzle. To me that doesn't undermine the contributions of this work, even though it's a bummer that it's not higher. But there is value in non weight-sharing approaches, regardless of if they are yet to capture the full rich spatialized human behavioral measures.

Reviewer #2 (Remarks to the Author):

The authors have addressed all of my concerns.

Reviewer #3 (Remarks to the Author):

First of all, I would like to thank the authors for the complete revision, the revised paper and the quality of the detailed responses to the reviewers.

I am fully satisfied with the responses to my comments, and the changes made to the paper have greatly improved its quality. I would recommend minor revisions. In particular, I would be very grateful for responses to the following points, and their inclusion in the paper if you consider this would strengthen the paper even further:

- A key component is the introduction of smoothness on weights in the loss. This is standard practice to work on weights for regularizing models (weight decay, ...) and I understand your choice. However, could you achieve the same result by using a smoothness on activities? This would be more biologically realistic as neurons tend to differentiate activities rather than synaptic weights.

- I appreciated both reviewer #1's comment that there is no retinotopy (cortical magnification) in the input and your detailed response. It seems that your model optimizes efficiency by learning the non-uniform mapping, but that this non-linear transformation could also be introduced a priori using a mapping of the input. Such CNN-based architectures with retinotopic mapping do exist (see doi:10.48550/arXiv.2402.15480), and I would guess that the entropy map computed in Figure 2-B would become more uniform. Following your statements in the conclusion, could you say that an optimal mapping of the input would be one where you include this non-uniform mapping in your model and that the entropy becomes uniform? (Intuitively, this would optimize resource usage)

- It would be useful to provide more details on the SimCLR method, in particular the set of augmentations used. Indeed, Figure 5-H shows the lack of similarity of the SimCLR-trained network, and this may be due to the fact that the augmentations used to guide learning in this scheme are based only on geometric transformations. If an augmentation based on localization and categorization were introduced, the similarity might increase.

Other minor points:

- in Figure 1, error bars are too small to be seen or plotted. Indicate them as values in the text or define a larger CI.
- page 7: " performance.To " > " performance. To "
- Figure 4 (and following): you do not define precisely "spikiness" in your text
- page 19: Eq(2) uses wrong indices
- page 20 "computationally efficiency." > "computational efficiency."

Version 3:

Decision Letter:

Our ref: NATHUMBEHAV-23082732C

24th February 2025

Dear Dr. Kietzmann,

Thank you for submitting your revised manuscript "End-to-end topographic networks as models of cortical map formation and human visual behaviour" (NATHUMBEHAV-23082732C).

We sent the manuscript to Reviewer #1, but unfortunately they did not reply to our invitations to re-review, despite many requests. We therefore sent the manuscript to Reviewer #2 and #3 and asked them to comment on your responses to Reviewer #1; they were happy with your responses. I apologize for the delay in this final round of review.

We will therefore be happy in principle to publish it in Nature Human Behaviour, pending minor revisions to comply with our editorial and formatting guidelines.

We are now performing detailed checks on your paper and will send you a checklist detailing our editorial and formatting requirements within two weeks. Please do not upload the final materials and make any revisions until you receive this additional information from us.

Sincerely,

[Redacted]

[Redacted]

[Redacted]

Nature Human Behaviour

Version 4:

Decision Letter:

Dear Professor Kietzmann,

We are pleased to inform you that your Article "End-to-end topographic networks as models of cortical map formation and human visual behaviour", has now been accepted for publication in Nature Human Behaviour.

We welcome the submission of potential cover material (including a short caption of around 40 words) related to your manuscript; suggestions should be sent to Nature Human Behaviour as electronic files (the image should be 300 dpi at 210 x 297 mm in either TIFF or JPEG format). Please note that such pictures should be selected more for their aesthetic appeal than for their scientific content, and that colour images work better than black and white or grayscale images. Please do not try to design a cover with the

Nature Human Behaviour logo etc., and please do not submit composites of images related to your work. I am sure you will understand that we cannot make any promise as to whether any of your suggestions might be selected for the cover of the journal.

With best regards,

██████

██████████████

██████████

Nature Human Behaviour

P.S. Click on the following link if you would like to recommend Nature Human Behaviour to your librarian
<http://www.nature.com/subscriptions/recommend.html#forms>

** Visit the Springer Nature Editorial and Publishing website at http://editorial-jobs.springernature.com?utm_source=ejp_NHumB_email&utm_medium=ejp_NHumB_email&utm_campaign=ejp_NHumB for more information about our career opportunities. If you have any questions please click [here](mailto:editorial.publishing.jobs@springernature.com).

Open Access This Peer Review File is licensed under a Creative Commons Attribution 4.0 International License, which permits use, sharing, adaptation, distribution and reproduction in any medium or format, as long as you give appropriate credit to the original author(s) and the source, provide a link to the Creative Commons license, and indicate if changes were made. In cases where reviewers are anonymous, credit should be given to 'Anonymous Referee' and the source. The images or other third party material in this Peer Review File are included in the article's Creative Commons license, unless indicated otherwise in a credit line to the material. If material is not included in the article's Creative Commons license and your intended use is not permitted by statutory regulation or exceeds the permitted use, you will need to obtain permission directly from the copyright holder.

Dear editor, dear reviewers,

Thank you for the opportunity to provide a new and strengthened version of our manuscript. We would like to express our gratitude for the many constructive and helpful comments. With this revision, we address all concerns raised by the reviewers, as detailed below in our point-by-point reply. The main improvements included with this revision are:

Major improvement 1: A large set of control models trained and tested

The reviewers highlighted several aspects for which additional control models could improve our manuscript. We conducted all of these controls, and more. They include

- (a) All-TNNs with various α values to explore the effect of the smoothness loss magnitude,
- (b) adding LCNs to the behavioural analyses (previously they were only included in the topographical maps section),
- (c) an All-TNN trained using a self-supervised objective (SimCLR) to test for the effects of training without supervision on the emergence of topography and behaviour,
- (d) TDANNs (a CNN modified to study neural topography by Margalit et al., 2023/2024) to test if this alternative approach also matches human spatial biases in behaviour,
- (e) an All-TNN trained using a spatially shifted version of ecoset to show that the topographic maps are driven by the location of task-relevant information in the training dataset, and
- (f) an All-TNN for which the smoothness loss is computed across all 8 neighbouring units to test if our approach of computing the loss for the 'east' and 'south' neighbours only is appropriate.

All of these new controls support and strengthen the paper's main claims, while further allowing us to specify which model aspects are important to mirror primate topography and behaviour.

Major improvement 2: Smoothness, performance, and energy efficiency

The reviewers asked us to expand our analyses of the impact of our smoothness loss on performance. Is there a trade-off between smoothness and performance? Could smoothness be seen as a regulariser that can improve performance? We now report the test performance of all our networks, and find that CNNs, LCN, and All-TNNs with various α values all reach comparable performance (with a slight advantage for CNNs). This raises the question of the computational advantages of topographic organisation in the brain. In previous work, topographic organisation has been linked to reduced energy costs of primate visual processing. In a new results section, we test for this possibility in our networks. Using a proxy of energy consumption in biological networks, we find that All-TNNs are more energy efficient than control models, and that regions with smoother selectivities consume less energy. This suggests that topography may be beneficial to reduce energy consumption, rather than to improve classification performance.

Furthermore, following reviewer and editor comments, we now include a more thorough investigation of the parallels between our models and ventral stream organisation, including a quantification of spatial feature smoothness, radial cluster sizes for low-level orientation selectivity and high-level category selectivity, estimates of spatial variability in feature selectivity with increasing eccentricity, and the developmental trajectory of topographic emergence across model training.

Major improvement 3: Reworked and improved behavioural results section

The reviewers asked for several changes of our section on spatial biases in object recognition behaviour. They pointed to a lack of clarity in certain subsections, raised limitations in the reliability of certain aspects of our results (especially the old Figure 4), and suggested additional control models. In response, we have thoroughly restructured and improved this section. We removed analyses that were not directly linked to our main claims (old Figures 4 & 7), and replaced them with new analyses that strengthen and deepen our central results about the match of All-TNNs to spatial biases in human object recognition. This includes

- (a) New hierarchical clustering and GLM analyses to show structure in our measured human spatial biases in object recognition,
- (b) a direct demonstration that our models can learn object-specific spatial biases,
- (c) the additional control models mentioned earlier (All-TNNs with various alphas, SimCLR-trained All-TNNs, LCNs, and TDANNs), and
- (d) analyses with different stimulus sizes, reproducing our main results.

While the main conclusions from our previous submission (the benefit of All-TNNs over the gold standard of CNNs) remain the same, this substantially reworked section offers a more complete and solidified picture of the alignment of our various models with human behaviour.

Further improvements

Aside from these main changes, to further support our conclusions, we added various quantitative measurements of topography in our networks, and generally improved our analyses, plots, and narrative. The reviewer comments have also helped us to better contextualise our work in the literature, and to provide an expanded Methods section and reworked figures that clarify our approach.

Detailed responses to each reviewer comment are provided below, following this format:

Text items in blue boxes are reviewer comments.

Response: Black text items are our replies.

Actions:

- We provide page and line numbers for the main relevant changes made to the manuscript.

Reviewer #1

1- Questions about the All-TNN model implementation

1A - Alpha hyperparameters

Figure 1 is very helpful, and in general your methods are very clear. To understand the results, I needed more details about the hyperparameter alpha values on the smoothness of the layers, how they were explored, selected, and how that impacts the outcomes. For example, the methods state: “we used a spatial loss of $\alpha=10$ in all layers except the final, for which we used $\alpha = 10$, due to increased smoothness observed in the higher visual cortex”. A couple typos on there, and also the same alpha value, so, hard to follow. *How were these alpha values set and how do they matter?* There is room to do a controlled rearing experiment here, with a suite of models varying in alpha that you could characterize. It also sounds like you needed to increase the alpha of late stages to make it ‘blobbier’? But maybe a supplemental figure just showing the range, from low and salt-and-pepper, to extremely high (e.g. very blobby), would be valuable. As it is now, it is just a little confusing what your commitment is to that alpha value, across layers, and what your search procedure for setting those was, especially as I assume your main topographic results depend on this parameter setting.

Response: Thank you for pointing out the need for more detailed explanations regarding the spatial similarity loss weight (i.e., the α hyperparameter). We did not mean to imply that there is a globally optimal value of α that we commit to strongly. Following your suggestion, we now include various α values ($\alpha \in [1, 10, 100]$) across all of our experiments. We justify the choice of these values in the expanded subsection on All-TNN architecture in the Methods. As detailed in response to your following comments, we find that α can be high without deteriorating classification performance, while impacting most aspects of topography and behaviour. We now discuss these effects throughout the Results and Discussion.

Finally, in the prior version of our work we multiplied the level of regularisation in the final layer by 10, motivated by the fact that higher visual cortex shows more smooth regions. However, we decided to change this to equal regularisation of all layers in the current version for simplicity and clarity.

Actions:

- All our results now include All-TNNs trained with $\alpha \in [1, 10, 100]$, to characterise how topography and behaviour change when α changes. These results are shown in all relevant main Figures, as well as all relevant Supplementary Figures, and are extensively discussed in the main text.
- Our networks are now trained with identical α in each layer.
- *p.19 l.567ff:* We have expanded the *All-Topographic Neural Networks* subsection in the Methods to include better explanation and justifications of our modelling choices.
- The Results and Discussion sections now explicitly address the effect of α .

1B - Spatial similarity loss

The way you implemented tuning spatial similarity loss, as far as I can tell, is that for each unit you compute the kernel weights correlation with the unit next to it to the right, and one below it, then you sum this. Then, you sum this across all units. If I got it correctly, this neighborhood function seems a bit asymmetric and small to me (just the next neighbors, and only two of them rather than, say, all units within some distance on the cortical sheet, including diagonal units). I know this is a huge space and you had to pick something, but I admit this choice of just two adjacent units surprised me a bit. I suspect that variations in the way you operationalize this L_s , e.g. larger neighborhood, would have big effects on the kinds of results you'd get. So, an explanation of why this choice (was it on purpose, and you explored others and this was good for reasons, or was this just a start) would help me as a reader, both in the methods/results and in general discussion. These implementation (along with the Alpha's) really seem like the key decisions you had to make and discussing them, and your commitments to them, would strengthen this work and clarify how future work could build off it.

Response: We agree that the choice of the neighbourhood in which the loss is computed was not sufficiently clearly motivated in the previous version. Our loss is computed by averaging the cosine distance between the kernels of all (south and east) neighbouring units in each layer. Then, the total loss is summed across layers. The choice of only using the south and east neighbours is based on the observation that if we took a symmetrical neighbourhood (for example the von Neumann neighbourhood, i.e., north, east, south and west neighbours), all distances between units would be included twice, since the cosine distance between units A and B is symmetric. This results in costly calculations that have no effect, given that the final loss is averaged across the units. We now motivate our analysis choices in the methods.

This, however, leaves open the possibility that including diagonal units may have an impact. Our reasoning was that the similarity between diagonally adjacent units is achieved indirectly: if unit A is similar to its eastern neighbour B, which is similar to its southern neighbour C, then A and C are (indirectly) driven to be similar. We chose this option because it has the advantage of being computationally more efficient. To check the validity of these choices, following your suggestions, we added a control that includes all 8 neighbours (i.e. Moore neighbourhood, which is isotropic and includes diagonals). This control network achieves comparable categorisation performance, and selectivity maps, confirming the adequacy of our assumptions (Supp. Fig. 25).

Note that we use periodic boundary conditions (i.e., the layer is wrapped in a toroidal fashion), meaning that the unit at the very bottom (resp. right) of the layer has its loss computed with the top-most (resp. top-left) unit of the layer. This is done to ensure that all units have the same number of neighbours, thus avoiding border effects. We clarified these aspects in the Methods, and added a Supp. Fig. 10 to show the boundary conditions.

Actions:

- *p.19 1.578ff*: We have better described how our loss is computed, and justify the choice of only using 'east' and 'south' neighbours.

- Supp. Fig. 25: We have included one model instance trained with a Moore neighbourhood (i.e., all 8 neighbours).
- Supp. Fig. 10: We have added an illustrative figure of how the periodic boundary conditions of the smoothness loss ensure that there are no boundary effects.

1C - Object classification results and smoothness

I found that I really needed a results section, after you describe the model, but before you talk about layer 1 orientation maps, that reports the results of the model training and smoothness objectives! (across all $N=30+$ models you have). What is the average object classification accuracy within a model type (and was there a lot of variance across the 10 seeds?). Next, you can even quantify the emergent smoothness in the LCNs (!!) and ask: what is the L_s of these models? (helpful in the context of the all-tnn models that are optimized for this directly). This just reports that your models trained well and did what you expected (and provides context for the next comment).

Response: Thank you for flagging the need for more in-depth reporting on network training results. As you suggest, we added a new results subsection that includes the classification performance (Fig. 1B), and spatial smoothness (quantified as $1/\text{smoothness loss}$) for all models (Fig. 1C), including LCNs, All-TNNs with various alphas, and CNNs. We find that All-TNNs achieve similar performance to LCNs, while developing spatially smoother weights (for all α values tested). This shows that All-TNNs successfully learn a large-scale image recognition task, while satisfying the smoothness constraint.

Actions:

- *p.4 l.101ff*: We have added a new section reporting on classification performance and smoothness for all models.
- *Fig. 1B & C*: We have added a new figure visualising classification accuracy and emergent smoothness for all models.

1D - Accuracy-smoothness tradeoff

You assert that the spatial similarity loss “forces the network to trade off between i) learning varied feature selectivity... and ii) preserving similar features selectivity between neighboring units.” However this claim is never shown or quantified. You could directly test for accuracy-smoothness trade offs: if you varied alpha, does the level of smoothness (L_s) really trade off with accuracy? (For a richer exploration, you could even let the Alpha’s-by-layer be learnable, and see what values are learned!). These analyses/variations are important, because they get at the theoretically important question of whether topographic tuning smoothness can be a useful inductive bias to help promote classification accuracy in the huge space of possible solutions. Or, alternatively, if in this way of operationalizing the

cortical formation as a joint loss function of categorization and smoothness, these constraints directly trade off.

Response: As covered in above responses, we analysed All-TNNs with varying magnitudes of the α hyperparameter. We find that increasing α does not substantially change classification performance, but does lead to smoother weights (see comment 1C) and smoother selectivity maps (see comment 2A). Hence, we agree that “tradeoff” is not the right term, as it suggests that smoother networks should perform worse. We have removed this term.

We think that the network, having to minimise both losses simultaneously, allocates more computational resources to task-relevant areas of the inputs at the expense of less relevant regions, in line with our analysis showing higher feature variability in the centre of the image (Fig. 2B). In other words, the network achieves a different balance between the two losses in different regions of the layer. We added new results to support this hypothesis, in which we trained an All-TNN on a spatially shifted version of ecoset. This leads to a corresponding shift in the spatial organisation of the network (see Fig. 2C and Supp. Fig. 11).

Since we have no evidence for an “optimal” α where performance is maximised, our smoothness loss cannot be seen as an inductive bias to promote classification performance. This raises the question of what the benefits of smooth topographical organisation are. We added a new section using a proxy of biological energy consumption, showing that, if All-TNNs were to run on biological wetware, they would be more energy efficient than CNN and LCN controls (see Fig. 3). In the discussion, we elaborate on how this may be linked to the aforementioned ability of All-TNNs to allocate computational resources to task-relevant regions.

Concerning the learnable smoothness loss: we agree that this is an interesting option that we intend to explore in the future together with other computational modes to obtain smoothness. In sum, our results suggest that the balancing of our two losses does not yield one optimal α value with optimal performance/smoothness. Therefore, we opted to show results for various α values throughout the manuscript (see our response to your comment 1A).

Actions:

- *p.4 l.101ff:* We have added a new section reporting on classification performance and smoothness for all models.
- *Fig. 1B & C:* We have added a new figure visualising classification accuracy and emergent smoothness for all models.
- *Fig. 2C & Supp. Fig. 11:* We have added a new analysis showing that shifting the input dataset leads to a corresponding shift in topographical organisation.
- *p.8 l.241ff; Fig. 3; Supp. Fig. 15:* We have added a new results section showing that All-TNNs are more energy efficient than controls.
- We have removed the term “trade-off”, and included a more nuanced discussion of the effects of the model balancing the dual loss throughout the manuscript.

I recognize there's a range of possible action items in the comments above. At a minimum, I would suggest reporting the classification accuracy of all the models and their spatial smoothness loss, either by varying alpha directly, or at least by describing your process for hyperparameter search over alphas. And, either test whether these trade off and by how much, or remove that assertion.

Response: Thank you for the comments, which have helped us clarify our approach in the paper, as described above.

1E- "Hypercolumn"

I believe you are using "hypercolumn" incorrectly throughout the manuscript, especially as you introduce this in the results section. A hypercolumn is a vertical column through the laminar structure of the cortex, within a single cortical area. These neurons along the depth axis of the cortex tend to have the same RF and tuning profiles. However, you're using it more like a way to refer to the hierarchical construction of receptive fields, where the increasing receptive field of a unit in layer 2 is linked directly to the connections from layer 1. This is not a hypercolumn structure. This is, maybe, a 'hierarchical spatial receptive field motif' to try to give it a name. Definitely remove "hypercolumn" and use a different term for this.

Response:

We agree that this term can be confusing and decided to remove it from the manuscript. In the previous version, by using the term "hypercolumn", we referred to the classic *ice-cube model* in which units that share a spatial receptive field in V1 are arranged according to their feature selectivity. Hypercolumns (which consist of multiple columns each) thereby have units selective to all orientations at a given location of the visual field. To quote Hubel (1982): a hypercolumn is a 1-mm block of V1 containing "all the machinery necessary to look after everything the visual cortex is responsible for, in a certain small part of the visual world". Analogously, in All-TNNs, each layer contains groups of adjacent units that share the same spatial receptive field location while being selective for varied features. These adjacent units sharing a RF are what we called a "hypercolumn" (note that the connectivity across network layers defines the spatial RF of a given model unit). In theory, each of these All-TNN "hypercolumns" could contain units tuned to all orientations, but the outcome of learning may not always satisfy this constraint, which may be confusing for readers.

Hubel, D. H. (1982). Exploration of the primary visual cortex, 1955–78. *Nature*.

Actions:

- We have removed reference to hypercolumns in the manuscript.

2. Topographic features of the ventral stream

2A - Lovely orientation results.

The main thing that lacked for me is quantification. You talk about orientation ‘smoothness’ but, e.g. didn’t necessarily quantify pinwheels or degree of emergent smoothness, compared to LCNs and CNNs. One idea: what’s the average radius you need to go from a unit to include units with all 8 preferences? This will also allow you to quantify that for All-TNNs the radius will vary as a function of center-to-edge, but won’t for LCN vary in an eccentricity dependent way for LCNs.

Response: We agree that our results can be strengthened by better quantification. To this end, we have implemented several analyses. First, as described in response to your comments 1A and C, we have quantified the emergent smoothness between the weights of neighbouring units. Furthermore, to quantify the smoothness of our orientation- and category-selectivity maps, we apply two methods: the analysis you suggested to determine cluster sizes, as well as a 2D Fast Fourier Transform (FFT). In the analysis quantifying cluster sizes, we find that the average cluster size of the orientation selectivity and category selectivity maps becomes larger with increasing α , while it is smaller in both CNNs and LCNs (Fig. 2A and 2D, Supp. Fig. 5). Plotting this cluster size against eccentricity, we find that, indeed, cluster size increases with eccentricity (Supp. Fig. 6).

In the Fourier analysis, we retrieve the power spectrum distribution of frequencies present in the selectivity map. A smooth map will display more power in low spatial frequencies than a salt-and-pepper map. We find that power in low spatial frequencies increases with increasing values of α , indicating a direct relation between the spatial similarity loss and amount of smoothness (Supp. Fig. 7).

Finally, to show that the All-TNNs are not only smooth, but also have discrete point discontinuities in orientation selectivity (akin to pinwheels), we applied the pinwheel analysis approach by Kaschube et al. (2008) (see Supp. Fig. 4). This analysis shows that, for All-TNNs, the number of pinwheels decreases with α magnitude. LCN ($\alpha = 0$) and CNN controls have an unreasonably high number of pinwheels, due to their salt-and-pepper orientation maps. Note that the common way of retrieving pinwheel density is by normalising pinwheel count by mms of cortex. However, since the exact number is dependent on a contingent decision of matching visual cortex distances to model layer size, we decide to refrain from this. Furthermore, we find that this analysis is very sensitive to noise, causing small clusters of units classified as pinwheels to appear. One way to resolve this is to artificially smoothen the map before performing the analysis; we refrain from doing this, too, and instead select the unit with the highest winding number in the Moore neighbourhood.

Kaschube, M., Schnabel, M., & Wolf, F. (2008). Self-organization and the selection of pinwheel density in visual cortical development. *New journal of physics*, 10(1), 015009.

Actions:

- *Fig. 2A and 2D, Supp. Fig. 5:* We have implemented the cluster size analysis you suggested.
- *Supp. Fig. 6:* We have quantified the variation of cluster size of orientation selective clusters per eccentricity.

- *Supp. Fig. 7*: We have quantified smoothness using power spectrum analysis of selectivity maps.
- *Supp. Fig. 4*: We have quantified the pinwheel discontinuities in our networks.
- *p.22 l.721ff*; We have described the methods for these new analyses in the Methods section.

2B - Cortical magnification claims are incorrect.

The way you linked these results to the ‘foveal bias’ confused me—you claim that “cortical magnification emerges”. But, I think there is a deep mismatch here. To expand on this: your first model layer map has units which each see a FIXED size of the visual field. There is no oversampling at the center of the image through a fovea. And, the definition of cortical magnification is that a disproportionately large area of the visual cortex is devoted to processing information from the central region of the visual field (the fovea) compared to the peripheral regions. Thus, your architecture does not have cortical magnification. So, claims to this end need to be removed. What your model DOES show is that, given an architecturally EQUAL sampling of the visual input with even sized receptive fields across the ‘visual field’, that more tuning variation emerges for processing central image statistics than for peripheral image statistics. This outcome implies a more normative possible claim, which is that if one wanted to have similar ‘smoothness’ of orientation tuning across the cortical sheet, one would need to over-represent the fovea and under-represent the periphery, and that would presumably still capture the image statistics well for the task. So, I think it’s fair to claim that your map of orientation gestures at an implication like this, but does not show emergent cortical magnification.

Response: We agree with your characterisation of the effects observed; our vanilla architecture does not have structural cortical magnification, as it has the same number of units per degree of visual angle at all eccentricities. Our claim, and this is now clarified in the text, is that All-TNNs are able to allocate more feature variety to relevant portions of the visual field at the expense of the spatial smoothing loss. We implemented several changes and a new analysis to better support this claim.

First, we now include results showing that shifting the training dataset such that there is more classification-relevant information in the bottom right quadrant, leads to a corresponding shift in the location where feature variability is increased (Fig. 2C, Supp. Fig. 11). This confirms that the increased feature variability is linked to processing task-relevant image statistics.

Second, and perhaps more importantly, given that All-TNNs have more varied feature detectors in regions with more task-relevant information, we tested whether units in regions with redundant selectivities can be lesioned without impacting task performance. Unlike LCNs and CNNs, All-TNNs allow for the smoothness of the feature maps to act as reliable proxy for which units can be lesioned. Once lesioned, All-TNNs, with minimal loss in performance, use more units in locations with task-relevant information (Supp. Fig. 12). The resulting model is reminiscent of cortical magnification in the primate visual cortex, where more neurons per degree of visual angle in regions process foveal information. We clarified the interpretation of this result in the results and discussion.

Actions:

- *Fig. 2C & Supp. Fig. 11:* We have included new results showing that the region with increased feature variability is driven by the location task-relevant information.
- *p.7 l.190ff:* We have clarified the presentation of our lesion study in the results.
- *p.15 l.457ff:* We have clarified and rephrased the interpretation of our results related to cortical magnification in the discussion.

2C - Category-selectivity results.

Unfortunately, when I look at Figure 2, layer 6 visualization, I don't easily see a 'face-selective area' or a 'scene-selective area'. E.g. these do not look like what you get when you make face-selective contrasts and can define face and scene region of OTC. I feel like this should be acknowledged. These maps are in the right direction (e.g. relative to LCNs), but claims of this map forming "category-selective areas" feel to me to be unwarranted by my inspection of the results. Either a case should be made, ideally in a quantified way, that these are 'areas' in some sense, or this claim needs to be significantly weakened. Certainly, for example, the category-tuning clustering is more clumpy than in the LCN, again would be nice if this were quantified.

Response: We agree that a quantification of the category clustering will improve our results, and that our wording hinted at an overly strong interpretation of our findings in terms of brain areas. We now address both of these concerns.

We do not claim that these category-based clusters are directly comparable to the regions found in visual cortex in terms of relative size, shape or distribution, etc. Instead, we claim that both All-TNNs and visual cortex display tuning-specific clustering. We have quantified this clustering in the final layer using the aforementioned methods (power spectrum analysis and cluster size quantification, see Fig. 2D, Supp. Fig. 5B; Supp. Fig. 7B). This improves on our previous analyses by quantitatively demonstrating that category-selective units are more clustered in All-TNN than in control models, and that higher α leads to more clustering.

In the textual revision of the manuscript, we made sure to avoid claims of a direct match to face/tool/scene areas in the brain.

Actions:

- We have verified that we do not refer to direct correspondence with category-selective regions in the primate visual cortex. We only claim that the existence of category-selectivity clusters is reminiscent of the brain.
- *Fig. 2D:* We have quantified category-selective cluster sizes throughout models.
- *Supp. Fig. 5B:* We have quantified the differences of category selective cluster sizes between models.
- *Supp. Fig. 7B:* We have quantified smoothness using power spectrum analysis of category selectivity maps.

3. Behavioral Results

3A - control models.

The LCN models absolutely need to be included in these experiments (Fig. 4 and Fig. 5). They are critical comparisons! All-TNNs should have the capacity to succeed, while LCNs should fail, based on your accounts. If the LCNs succeed, we need to know and understand those results! Similarly, the CCNs also succeed to some degree above chance it seems, at picking up on these spatial bias. Why? How? Do they? That seems worth discussing.

Response:

Before diving into the addition of LCNs and other control models, we would like to point out that we have decided to remove the old Figure 4, which was based on the spatial occurrence statistics of our 16 object categories in COCO. We made this decision for several reasons. First, our models are not trained on COCO, and neither are humans. Therefore, we realised that these COCO based statistics were at best indirectly linked to our results. Second, after implementing the shuffling you suggested in comment 3C, we found that it made little difference. This shows that category-specific differences in COCO spatial occurrence statistics were not relevant for humans and ecoset-trained networks. Please note that this does not invalidate our subsequent results, because they are not based on COCO occurrence statistics. Nevertheless, it makes the old Figure 4 largely irrelevant, and potentially confusing, which motivated us to remove it.

We agree that the LCN models are relevant to the behavioural comparisons, thank you for pointing this out. We have thus included the LCN in all of our main experiments. The LCN results match our hypotheses: they can match human behaviour better than CNNs, but not as well as All-TNNs.

In addition, in response to reviewer 2, we have also included TDANNs (Margalit et al., 2023/2024) as new control models for the behavioural experiments (Supp. Fig. 20), as well as an All-TNN trained with the SimCLR self-supervised learning objective (Fig. 5; Supp. Fig. 24). Together with our exploration of various α values, these new control models offer a more comprehensive picture of the abilities and shortcomings of the different model architectures, hyperparameters, and training objectives.

As you mention, CNNs do have spatial biases to some extent. This is linked to the fact that the last layer of CNNs, despite not having a spatial dimension itself, is free to "listen" to a spatially distinct set of units in the layer below, and hence can represent spatial biases in behaviour, too. Similarly, the fully connected readout can pick up on spatial biases, too. We expect these features to provide the CNNs with limited spatial priors. LCNs and All-TNNs, in contrast, are free to learn different features at different spatial locations throughout the network, which may explain the stronger spatial biases they exhibit. We discuss this in the relevant part of the Results section.

Margalit, E. et al. A Unifying Principle for the Functional Organization of Visual Cortex. 2023.05.18.541361 Preprint at <https://doi.org/10.1101/2023.05.18.541361> (2023).

Margalit, E., et al. A unifying framework for functional organization in early and higher ventral visual cortex. *Neuron* (2024).

Actions:

- *Old figure 4:* we have removed the analyses that involved the positional occurrence statistics of MS COCO.
- *New figure 4:* we have included the LCN in all experiments, including the behavioural experiment.
- *Supp. Fig. 20:* We have added two more control models; namely TDANN trained on object categorisation (one instance) and TDANN trained using SimCLR (5 instances).
- *Fig. 5; Supp. Fig. 24:* We have added a figure that covers all analysis performed on All-TNN trained with SimCLR.
- *p.13 l.383ff:* We now discuss the limited spatial bias in CNNs.

3B - Data reliability. [see edit]

As a first step of the data analysis, I would like to see the noise ceiling of the human data (before you verify these maps correspond to the objects occurrence statistics) For example, in Figure 3b, you show several templates for 16 objects, but I do not have a good sense from the results how much this variation across items is stable or noise. Can you do an analysis that quantifies this? some ideas: reporting the split-half reliability of the pattern? Maybe showing that a pattern is more similar to itself than other items on average across split halves? Maybe a linear mixed effects model statistic where you have to model accuracy with position as a predictor and a random effect of subject, and you can include item as a predictor as well, and item*position, and show that including item*position interactions are significant? Further, situating the results in Figure 4a and 5b with respect to the data reliability would help contextualize the results.

Edit: I just saw in the methods that you do in fact compute the noise ceiling and you are already showing reliability adjusted correlations! Definitely report the noise ceiling first please, and state clearly in the results you're showing noise adjusted correlations. I frankly would prefer to see the raw correlation values with the noise ceiling of the data indicated on the same axes. If the data are noisy, make the case for why the effect is important even if it's noisy, and/or qualify the results and point to important directions for future work. Don't hide the noise.

Response: We performed multiple analyses to estimate the data quality and variation across individuals. First, we now provide two different analysis approaches: once testing the individual model seeds against the average human behaviour per category (Fig. 4E, 4F), and once testing every model seed against every individual separately together with an explicit report of the noise ceiling (as requested; see Supp. Fig. 21). Both analysis approaches confirm the main results, and both are referenced in the main paper.

We perform multiple additional analyses to further test the reliability of our behavioural data. First, we performed a hierarchical clustering analysis and a GLM analysis on the ADM computed from the average human category accuracy maps. We use several predictors inspired by large-scale organisation principles in IT, including real-world-size, animacy, and “spikiness” (main text and Fig. 4D). These show that, indeed, real-world size and animacy explain significant unique variance in the behavioural data, while spikiness does not. Second, we directly test object-based differences in behaviour by statistically comparing the accuracy maps of all pairs of categories, as now reported in the Supplemental Materials. Together, these results indicate that humans do express reliable positional effects on accuracy depending on the object category seen.

Actions:

- *p.25 l.838ff*: We have extended the Methods describing the analysis approach in detail.
- Fig. 4E, 4F: We report on the analysis comparing individual models to average human behaviour.
- Supp. Fig. 21; Supp. Fig. 22B: We have added a figure showing the analysis comparing individual models to individual human behaviour.
- Fig. 4D; Supp. Fig. 18: New hierarchical clustering and GLM analyses showing structure in the human ADM.
- Supp. Fig. 17: We have added a statistical comparison of averaged object accuracy maps to show there are significant object-specific differences in human behaviour.

3C - Shuffle control

For figure 4, in addition to adding the LCNs, it would be useful to add a spatially shuffled prior as a control analysis, to get a sense of where the CNN is falling with respect to the shuffled control.

Response: As described in our response to the previous comment 3A, we have removed the analyses present in Figure 4.

Actions:

- *Old figure 4*: we have removed the analyses that involved the positional occurrence statistics of MS COCO.

4. Discussion/Abstract

Based on the above comments, I recommend: qualifying the claims about the emergent organization of category-selective regions, clarifying the logic of the cortical magnification argument, and removing

the claims about a trade-off between classification and smoothness unless this is directly shown with new analyses.

Response: Thank you for the many great suggestions, all of which we have addressed, as detailed above.

5 Minor/Optional/Potential small edits for increased clarity

5A. Terminology: LCNs: “local” is confusing to me here, because I usually think of local like local WITHIN A MAP/AREA, local connectivity is near-neighbor (in the cortex) connectivity. I think maybe your LCNs could be called ‘pure-hierarchy’ models or some alternative?

Response: We use the term “locally connected network” because it is standard usage in machine learning (for example, our LCNs are implemented using tensorflow’s “LocallyConnected2D” layer). To prevent confusion with lateral connectivity we have included a more detailed description of what differentiates LCNs from CNNs and All-TNNs throughout the paper, for example by pointing out that a LCN has technically the same architecture as a CNN but without weight sharing, and that it is equivalent to an All-TNN but with $\alpha=0$. Furthermore, we have added subsections in the Methods section that describe each model and their differences in more detail.

Actions:

- *p.19 l.567ff:* We have added a section describing each model and their differences more clearly in the Methods.
- We have clarified the differences and similarities between LCN and other models better throughout the paper.

5B. Intro: “locality” desideratum: I’d say ‘units need to have local receptive field *feature weights* that are learned individually and not enforced to be exact duplicates of other RFs’. This is to help clarify the difference between feature tuning (flexibly learned) and spatial receptive fields (fixed by the architecture)

Response: We agree, and have modified this point in the paper following your suggestion. We also rephrased the term “desiderata” to prevent misunderstanding that they provide a complete list of requirements.

Actions:

- *p.1 l.61ff:* We have updated the relevant sentence according to your suggestion.

5C. Methods: Just checking: the cosine difference between the weight kernels, here this would be calculated over say, a flattened 27x27 set of learned weights if that is the size of the kernel? (rather than, say, over the activation profiles of a test set of images, which is more like what is done in the empirical literature noting smoothness, because we don't have access to the connectome weights). Clarifying this in the methods might be valuable.

Response: You are correct, we compute the loss based on the weights rather than on the activity patterns. For each unit, the $kernel_size \times kernel_size$ weight matrix is flattened to a $kernel_size^2$ vector, and cosine distances are computed between such vectors. As you mention, we do not have access to the full connectome of the human brain, but neural network modelling gives us the opportunity to investigate the effect of constraining neural connectivity during learning. We implemented several changes in the methods to clarify how the loss is computed.

Actions:

- *p.19 l.578ff:* We have updated the Methods section to be more clear about the calculation of the smoothness loss.

5D. Figure 2. How are you plotting CNN preference maps, as in 2a. Are you just spatializing the features in a systematic grid? Maybe say this in the caption?

Response: We plot the CNN and LCN maps with the same procedure that is used to create the 2D "cortical sheet" of All-TNNs. This procedure simply involves reshaping the 3D layer to a 2D $\sqrt{channels} * height \times \sqrt{channels} * width$ layer. This means that each channel is reshaped to a 2D $\sqrt{channels} * \sqrt{channels}$ square of units. Upon your feedback, we have clarified this in the methods, and included a new subsection ('LCN and CNN topography visualisation') that explains how we create 2D maps from the 3D layers of CNN and LCN. We now clarify this in relevant parts of the text, such as the caption of figure 2.

Actions:

- *Fig. 2:* We have updated the caption to clarify how CNN and LCN maps are visualised.
- *p.19 l.573ff:* We have expanded the Methods section on the construction of the 2D sheet in All-TNN.
- *p.21 l.668ff:* We have added a new Methods subsection ('LCN and CNN topography visualisation') that clarifies this procedure and where it is used.

5E. Figure 2. You might clarify how you calculated entropy in a sentence in the main text, this term caught me off guard while reading through.

Response: We have clarified this in the text by referencing the type of entropy (Shannon entropy) in the results and extending the relevant Methods subsection.

Actions:

- *p.7 l.176ff:* We have added mention of Shannon entropy and clarified how to interpret it in our context.
- *p.22 l.710:* We have specified the exact function we use in the Methods section.

5F. Figure 4A: it took me a while to parse this diagram. I think it's because the 16 model object maps are arranged vertically and the 16 human object maps horizontally in a matrix, but you're only doing the corresponding correlations (and not making and rdm? Unless I really missed something). Figure 4b, in truth I struggled with this analysis and figure.

Response: We thank you for pointing out that the current arrangement of the first panel of plot 4A is confusing. As mentioned in response to your previous comment 3B, we have removed the old Fig. 4 completely. Previously, we used the same confusing arrangement for the old figure 5A, and have adapted that now to be clearer (new Fig. 4D in the current manuscript).

Actions:

- We have removed the analysis on behaviour-to-object-occurrence-statistics that was previously presented in the old Figure 4.
- The diagram in the new Figure 4D had a similar layout as the one referenced in this comment. We improved it following your comment.

5G. Results order: I got confused by the presentation of behavior-to-object-occurrence statistics rather than behavior-to-model accuracy. Either these could be switched, OR, a little signposting ahead of time might help, because the results header really leans into the behavior-to-model accuracy as the key result.

Response: We have removed the old Figure 4 (i.e., the behaviour-to-object-occurrence-statistics, reasons described in detail in response to your comment 2B). The storyline is hopefully clearer now.

Actions:

- We have removed the analysis of behaviour-to-object-occurrence-statistics, streamlining the flow of our behavioural results.

5H. ADMs results section and stereotypical unit results section. In truth I found these sections a little hard to follow. Maybe separating out logic of the analysis and methods details in different paragraphs could help?

Response: Thank you for pointing out that these sections were not clear. To address this issue, we have firstly decided to remove the section on stereotypical unit activation (old Fig. 7). The idea was that All-TNNs may develop, for example, a “bus subnetwork” during training, and this subnetwork is driven by buses presented at expected locations (dataset statistics). We attempted to locate this subnetwork in the last layer by using the average activation map for buses in the ecoset test set. We thought that, when the bus is presented at unexpected locations, this subnetwork is not engaged, leading to poor performance, thus tying topography and behaviour together. However, when testing LCNs with this analysis we found that they showed the same pattern as All-TNNs, due to their ability to also learn different weights at different locations. Still, All-TNNs better match human visual behaviour. This made us realise that this analysis could not explain the true reason why All-TNNs better match human spatial biases compared to control models, and hence decided to remove this analysis altogether.

Second, removing the old Figures 4 and 7 has allowed us to significantly streamline our storyline, while better motivating, explaining and expanding our core behavioural analyses. Specifically, we added analyses showing that there is object-specific structure in biases in human behaviour (Fig. 4D; Supp. Fig. 17; Supp. Fig. 18), validating the relevance of our behavioural dataset. Then, we proceed in two steps. a) we show that All-TNNs best capture human object independent spatial biases by directly correlating the model and human accuracy maps (Fig. 4E), and b) we show that All-TNNs with $\alpha = 10$ best capture human object dependent spatial biases by correlating Accuracy Dissimilarity Matrices (Fig. 4F). These core results are further strengthened by the inclusion of new control analyses. This includes showing that TDANN control models are outperformed by All-TNNs with $\alpha = 10$ in matching human behaviour (see our response to your comment 3A; Supp. Fig. 20), showing that the All-TNN with $\alpha = 10$ trained using a shifted dataset (see our response to your comment 1D) fails to match human behaviour likely due to misaligned topography (Supp. Fig. 11), and a reproduction of our results using different object sizes (Supp. Fig. 23). Finally, we now conduct our behavioural analyses using SimCLR-trained All-TNNs (Fig. 5).

Actions:

- *p.10 l.282ff*: We have improved the narrative and explanation of the entire behavioural results section.
- We have removed the section on stereotypical unit activation (old Figure 7)
- *Fig. 4, Fig. 5, Supp. Fig. 11, Supp. Fig. 17, Supp. Fig. 18, Supp. Fig. 20, Supp. Fig. 23*: We have added several new analyses and controls to validate our behavioural data and to support our main claim that All-TNNs better match spatial biases in human vision

Reviewer #2

1. The topographic neural networks proposed by Lee et al (2020) [Editors note - the referee has clarified that this refers to Lee, H., Margalit, E., Jozwik, K. M., Cohen, M. A., Kanwisher, N., Yamins, D. L., & DiCarlo, J. J. (2020). Topographic deep artificial neural networks reproduce the hallmarks of the primate inferior temporal cortex face processing network. bioRxiv, 2020-07], which similarly assumed 2D organization of neurons and demonstrates topographic organization, should be included in the evaluating and referenced appropriately as related work. Differences to Lee et al should be pointed out to clarify the novelty of the work.

Response: We agree that the work on TDANN, developed in parallel to ours, is relevant. Following your suggestion, we have included the most recent version of the model (as presented by Margalit et al., 2023/2024, and previously Lee et al., 2020), trained both with supervised and self-supervised objectives, as control models for our behavioural analyses (Supp. Fig. 20). We find that these models mirror human spatial biases in visual behaviour less well than All-TNNs. Please note that given the importance of dataset statistics in our results, the comparison is limited by the fact that the publicly available TDANNs are trained on Imagenet, instead of the ecologically-motivated ecoset that our models are trained on. While All-TNNs outperform TDANN in their published form, we therefore decided to include the comparison as a supplement, and mention this limitation in the discussion.

In addition to the inclusion of TDANNs in the results, we discuss the overall class of topographical models both in the Introduction and Discussion sections. We argue that the central benefit of All-TNNs is that they offer powerful topographic models capable of performing complex tasks, while entirely avoiding weight-sharing. We thereby differentiate our model from TDANN, which, under the hood, is a spatially embedded CNN with weight sharing and thus cannot directly model spatial variance in topography and behaviour. While we do think that CNNs can provide important insights into functional organisation, the assumption of identical features across the whole visual space is biologically unrealistic. As we now mention in the discussion, a large-scale and well-controlled comparison of diverse topographic networks will be important in future work.

Lastly, we reference Lee et al., as you suggested. While we initially cited the most recent publication on TDANN by Margalit et al., the early and influential efforts by Lee et al. in the domain of modelling cortical topography are now also included to properly contextualise our contributions.

Margalit, E. et al. A Unifying Principle for the Functional Organization of Visual Cortex. 2023.05.18.541361 Preprint at <https://doi.org/10.1101/2023.05.18.541361> (2023).

Margalit, E., et al. A unifying framework for functional organization in early and higher ventral visual cortex. *Neuron* (2024).

Lee, H., et al. (2020). Topographic deep artificial neural networks reproduce the hallmarks of the primate inferior temporal cortex face processing network. Preprint at bioRxiv, 2020-07.

Actions:

- *p.10 l.306ff; p.20 l.636ff; Supp. Fig. 20*: we have included TDANN in our evaluation of correspondence to human behaviour.
- *p.1 l.45ff*: We have more extensively discussed the differences to TDANN and other topographic CNNs in the introduction.
- *p.17 l.523ff*: We have contrasted our approach with several other approaches from the literature on topographic models in the discussion.
- We now include Lee et al. (2020) in our references, in addition to Margalit et al. (2023 and 2024).

2. The author's hypothesize that the center-periphery organization (e.g., cortical magnification) of the All-TNNs is likely emergent from the greater presence of object-relevant information in the center of the image. I am not fully convinced of this, especially since similar properties may emerge in LCNs if this were the case. Could this instead be an artifact of the spatial similarity loss? This loss function seems to suggest that central weight filters are implicitly regularized by all neighboring filters (above, below, left, and right), whereas border filters near the edge are not regularized by neighbors in all directions. The authors' claim may be supported better by demonstrating that this center-periphery organization emerges when trained on an augmented dataset that shifts the main image content outside of the image center or by training on a dataset where objects are not commonly centered in the image.

Response: Thank you for pointing out that we were not clear enough in explaining how we rule out the possibility that the centre-periphery organisation may be caused by border effects. As we now clarified in the results and methods, we apply the loss calculation in a toroidal fashion. This means that, a unit at the very bottom (resp. left) of a layer has its loss computed with the unit at the top (resp. right) of the layer. This means that all units are regularised to the same extent, avoiding border effects as an artefact of the smoothness loss. We added a new Supp. Fig. 10 to make this point clear, and show that the toroidal border conditions work as expected.

Further, thank you for the great analysis suggestion. To show that the increased feature variety in the foveal region is caused by unit weights adapting to the spatial statistics of the dataset, we have trained a control All-TNN on a shifted version of ecocet, where all images are displaced to the bottom right (see Fig. 3C, Supp. Fig. 11). In this model, the region with high feature variety is correspondingly shifted to the bottom right as well.

Finally, please note that we only use the "bottom" and "right" neighbours, and not all four direct neighbours. This is done to avoid unnecessary computations, since the loss is symmetric (i.e., the cosine distance between A and B is the same as the cosine distance between B and A). We have added a detailed discussion of this choice in the methods. In addition, we have added a new control where the loss is computed for all eight neighbours to fully rule out any deleterious effect of our neighbourhood choice (Supp. Fig. 25). As expected, this control yields similar results as our main model.

Actions:

- *p.7 l.183ff; p.19 l.581ff*: We now more explicitly mention that the smoothness loss is applied in a toroidal fashion in the Methods section and throughout the relevant sections on the topographical organisation of All-TNNs.
- *Supp. Fig. 10*: We have added a figure visualising that the features of All-TNN are spatially continuous.
- *p.7 l.185ff; Fig. 3C; Supp. Fig. 11*: We have trained an All-TNN on a shifted dataset and show that its feature organisation correspondingly shifts.
- *p.19 l.578ff*: We have extended the methods section where we describe the neighbourhood in which we compute the loss.
- *Supp. Fig. 25*: We have trained a new control model in which the smoothness loss is computed over all 8 neighbours.

3. In the lesion studies (which aim to support the greater importance of foveal processing in All-TNNs), LCNs and CNNs do not have high-entropy and low-entropy regions. Meanwhile, the All-TNN high-entropy lesion region contains the image center, which has more object relevant information, and is therefore more likely to affect classification performance. If the authors were to lesion the same image portions and feed them to the CNN and LCNs, I would imagine more similar observations of post-lesion classification accuracy would be observed across the three models. A more fair comparison may show the accuracy of LCNs and CNNs when the same high-entropy region from the All-TNNs is used as the lesion area.

Response: We agree that the framing of our lesioning study was unclear (please note that we have moved this figure to Supp. Fig. 12). Our claim is that All-TNNs exhibit focused allocation of resources to information-rich regions, including increased feature variety. This is supported by showing that All-TNNs are the only model for which the diversity of feature selectivities can be used as a proxy for which units can be lesioned without strongly deteriorating performance. The same findings hold for the All-TNN trained on a shifted dataset (mentioned in response to your previous comment 2), which develops a corresponding shift in the region with higher feature variability. Again, this shifted feature diversity can be used as a proxy for lesioning. In contrast, CNNs, by definition, fail to express this feature due to weight sharing (we hence removed them from the analysis). LCNs do not develop this characteristic during training, as reported. Together our results support the conclusion that feature variability is driven by dataset statistics in All-TNNs but not in control models, and is a good proxy for which units can be lesioned. Other models may exhibit similar drop in performance if one used the All-TNN maps to guide the lesioning, but the point made is that the All-TNNs themselves can discover those informative regions, whether the control models can or do not.

The view that All-TNNs exhibit focused allocation of resources in information-rich regions is furthermore strengthened by two new results. First, as we now quantify rigorously, All-TNNs have more varied weights in information-rich regions in their layers than control models (Supp. Fig. 2). Second, we added a new section where we quantify energy consumption throughout the network and at different locations within the model layers (Fig. 3; Supp. Fig. 15). We find that All-TNNs

consume more energy in the region with higher feature variability. This focus of energy consumption also shifts for the All-TNN trained with the centre of the images shifted.

Actions:

- *p.7 l.190ff; Supp. Fig. 12:* We have clarified the interpretation of the lesioning results both in the results and in the discussion.
- *Supp. Fig. 12:* We replicated the lesioning results in the All-TNN trained on a shifted version of ecoset.
- *Supp. Fig. 2:* We have included a quantification of weight smoothness throughout layers of all main networks, which also shows a focus on regions with task-relevant information.
- *p.8 l.241ff; Fig. 3; Supp. Fig. 15:* We have added a new analysis of the energy consumption throughout layers of all networks, again showing an increased allocation of resources in regions with relevant information.

4. In the behavioral bias analysis, positional occurrence frequency and uncertainty are computed from COCO images and it is assumed that stereotypical object locations could be computed from these images. Images taken by humans, however, naturally have a bias towards centering an object of interest near the center of the image and may therefore not actually contain representative positional occurrence frequencies as would be observed through the eyes of a human.

Response: We agree with your concern that the statistics of COCO images may not reflect the human visual diet (and our models are trained on ecoset and not COCO). As a result, the COCO based statistics were, at best, indirectly linked to our results. We have hence decided to remove the old Figure 4, which was based on the spatial occurrence statistics for our 16 object categories from COCO.

The rest of our behavioural analyses is not subject to this concern, as it is based on our experimental setup with objects presented on a 5x5 grid for both humans and models. With regard to the experimental human data, we added new hierarchical clustering and GLM analyses to show structured spatial biases in human behaviour (Fig. 4D), in line with previous work, and validating the value of the data for our purposes.

Actions:

- We have removed the analysis linking behaviour to COCO occurrence statistics that was previously presented in Figure 4.
- Fig. 4D: New hierarchical clustering and GLM analyses showing structure in human spatial biases.

5. This may simply be my misunderstanding, but further clarity on how 3D image representations (height, width, and channel) are “unfolded” into 2D neural sheets would be appreciated. The authors state that 3D representations of the form height x width x channel are unfolded into 2D representations of the form $(\sqrt{\text{channels}} \times \text{height}) \times (\sqrt{\text{channels}} \times \text{width})$. How are channel dimensions incorporated into height and width dimensions while preserving spatial information in the image representations?

Response: We improved our description about how the 2D sheet is created from the 3D structure. The unfolding procedure reshapes the channel dimension at each spatial location into a $\sqrt{\text{channels}} \times \sqrt{\text{channels}}$ 2D square at that location, thus preserving retinotopy. Hence, the $\text{height} \times \text{width} \times \text{channels}$ 3D layer is unfolded into a $\text{height} * \sqrt{\text{channels}} \times \text{width} * \sqrt{\text{channels}}$ sheet. This unfolding procedure is applied to All-TNNs for calculation of the smoothness loss, and for visualisation of the topographic feature maps of all models. We clarified these aspects in the expanded Methods section.

Actions:

- *p.19 l.568ff:* We have clarified the reshaping procedure in the Methods section on All-TNN architecture.
- *p.21 l.668ff:* We have added a Methods section (*‘LCN and CNN topography visualisation’*) that describes how we use the same reshaping procedure to visualise other models.

6. The authors should provide architectural information about the CNN. They mention that the CNN had a similar number of parameters, but should also include information about filter sizes, channels, etc.

Response: Agreed. We have added a table describing all architectures in detail to the supplementary material (Supp. Table 1). The CNN does not share the same number of parameters as LCNs and All-TNNs, because the weight sharing reduces those significantly. All models have the same unit numbers and kernel sizes, and hyperparameters. Random seeds are also controlled across architectures.

Actions:

- *Supp. Table 1:* We have added a table that describes all architectural details for CNN, LCN and All-TNNs.

7. The authors should consider showing object classification accuracy of each evaluated model on ecoset. Although it is not a motivating aspect of this paper, it could help the audience better understand the applicability of these networks to more general computer vision tasks.

Response: Done. We now report on the classification performance and weight smoothness of our main models (CNN, LCN and All-TNN with varying magnitudes of smoothness loss) in a newly added subsection in the Results. All models all reach similar classification accuracy.

Actions:

- *p.4 l.101ff:* We have added the section “*All-TNNs successfully classify natural images with smooth weight topography*” that reports on the classification accuracy of all models.

Reviewer #3

1 - First of all, a main point: through its supervised nature and being dedicated to categorization, this model stands out from similar models of self-organized learning in topographical maps. However, the emergent properties, particularly of the first layer, are similar to some of these models (such as <http://yann.lecun.com/exdb/publis/pdf/koray-cvpr-09.pdf>, doi:10.1016/S0042-6989(01)00114-6 or doi:10.1371/journal.pcbi.1010270). It would therefore be crucial to differentiate what makes this supervised model different from these unsupervised models of topographical self-organization. Indeed, a crucial difference lies in the definition of a cost function for the retinotopic organization observed in primates. Self-organizing models propose, for example, a metabolic-type optimization (e.g. of wiring length) or in relation to the efficiency of local information coding (e.g. sparseness). Your study could thus propose an alternative with a cost function linked to connection locality and categorization performance, as shown in your perturbation study. This point could be put forwards to discuss the origin and function of cortical topography.

Response: Thank you for the positive evaluation, we agree with your suggestion that our work should be better situated in the literature, particularly the unsupervised models you mention. We implemented several improvements to address this comment.

First, we expanded the discussion relating our approach to other models in the field (citing the relevant papers that you mention). We argue that our model is unique in that it is both explicitly topographic, and capable of performing complex image processing tasks on large datasets. Most other topographic models instead fall in one of two families. The first family, similar to the approaches you mentioned, comprises models that use relatively simple features from which topographies are learnt, usually with some form of unsupervised learning (for example, SOMs, sparsity constraints, etc.). While these models are usually explicitly topographic (in that there is no weight sharing, or a priori enforced structure), they are unable to deal with (complex) visual tasks. This limits them in modelling naturalistic human vision. For example, these models could not easily be tested on our behavioural experiment. The second family encompasses more recent topographic models that use CNN backbones. These models have the advantage of being able to perform complex tasks. However, they ultimately rely on weight sharing, which has several limitations. Weight sharing is not biologically realistic (as it is not the case in the brain that changing one synapse automatically changes synapses at all spatial locations identically). In addition, weight sharing enforces that the same features are extracted across the visual field, making it difficult to model non-homogeneous topography, as well as spatial biases in behaviour. We confirmed this by showing that a recent topographic model based on a CNN (TDANN, both supervised and unsupervised; see Supp. Fig. 20; Margalit et al., 2023/2024) does not develop human-aligned spatial biases in behaviour. Overall, All-TNNs differ from other existing topographic models, in that they are able to learn to perform complex tasks on natural visual input while being mechanistically truly topographic in all layers.

Second, we added All-TNNs trained using the SimCLR self-supervised objective as a way to test for the importance of category training. We find that these self-supervised All-TNNs develop similar topography (Fig. 5A-C; Supp. Fig. 24), and are also able to match object-independent spatial biases in

human behaviour (Fig. 5E), but not object-specific spatial biases (Fig. 5F). Hence, our results should not be interpreted as showing that supervised models are overall more promising than self-supervised approaches. Rather, for almost all results, what matters is the simple constraint of encouraging similar selectivity in neighbouring units. This requirement is sufficient to develop topographies reminiscent of the primate cortex, and human-aligned spatial biases in behaviour to some extent, independent of the specific training objective used. Only object-specific spatial biases are not human-aligned when using SimCLR, a finding that may have several causes, as described in a new discussion paragraph.

Spatial smoothness can be associated with the metabolic constraints discussed in the literature that you mention. For example, we now show that it can be linked to energy efficiency: in a new results section, we show that All-TNNs are more energy efficient than control networks, and that regions that are smoother consume less energy. In addition, our lesion study shows that spatial smoothness can be used as a proxy to determine which units can be removed without strongly harming performance, again tying into efficient coding. In this way, our approach connects to, builds on, and extends previous work. This is now mentioned in the discussion.

Margalit, E. et al. A Unifying Principle for the Functional Organization of Visual Cortex. 2023.05.18.541361 Preprint at <https://doi.org/10.1101/2023.05.18.541361> (2023).

Margalit, E., et al. A unifying framework for functional organisation in early and higher ventral visual cortex. *Neuron* (2024).

Actions:

- *p.17 l.523ff*: We have expanded the discussion to position ourselves better among other models in the literature.
- *Fig. 5; Supp. Fig. 24*: We have analysed an All-TNN trained with SimCLR.
- *Supp. Fig. 20*: We have added two TDANN control models for the behavioural experiment (one instance trained on object categorisation and five using SimCLR).
- *Fig. 3, Supp. Fig. 15*: We have added an analysis that quantifies energy consumption of all models.
- *p.16 l.479ff*: We mention the connection between the smoothness loss and literature on the influence of metabolic constraints on topography in the discussion.

I then have several secondary points:

2.1 - The analysis of the first layer is detailed, but is less precise for the following layers. Do you observe a similar organization? The origin of the variability of the topographies observed remains an enigma (cf doi: 10.1016/j.neuron.2021.09.053 for example) and your study could help to explain it.

Response: We agree that topography in intermediate layers is an intriguing (and challenging) question. To study topography in intermediate layers of All-TNN, we implemented two new analyses. First, we asked whether our observation of the centre-periphery organisation (see Fig. 2), is also

present in intermediate network layers. To test this, we have added an analysis measuring the spatial smoothness of the unit weights in each layer. We find that the centre-periphery distinction is replicated throughout all layers (Supp. Fig. 2). Second, we quantified energy consumption across layers (Fig. 3C; Supp. Fig. 15), and found that energy expenditure is focused in regions with task-relevant information in most layers.

Please note that, while feature selectivity in early and high brain areas along the ventral stream are somewhat well documented, the tuning of intermediate regions is less well understood. Therefore, analogously to the brain, it is difficult to select appropriate stimuli to determine selectivities in intermediate layers of All-TNNs. Future work in this direction could generate stimuli that maximally activate units in a given layer.

Lastly, we note that our current networks cannot easily model the work on the receptive fields of the tree shrew V2 that you shared with us, because our layers are structured retinotopically (except the final layer), i.e. prohibiting the emergence of sinusoidal transformations of the visual field. It could be that an All-TNN that would have the possibility for units to learn their position and connectivity would develop such a variable topography throughout its layers - an interesting question for future work.

Actions:

- *Fig. 1C; Supp. Fig. 2:* We have added an analysis quantifying the weight smoothness of all layers.
- *Fig. 2C:* We now show that the centre-periphery organisation in the first layer of our models is shaped by task-relevant statistics in the dataset.
- *Fig. 3; Supp. Fig. 15:* Quantification of energy consumption throughout layers.

2.2 - Concerning the parallel between the model's response and observed human behavior, the regressions are significant, but the behavioral responses are highly variable. It seems possible that other variables are involved, such as the effect of the intrinsic size of the visual objects in the images, e.g. that a bus is a priori larger than an animal, as is for instance demonstrated in Figure 3 (e.g. airplane, bus, train vs pizza, laptop or scissors).

Response: Thank you for your comment. As part of the revision, we removed the (noisy) regressions you are referring to from the manuscript (old Fig. 4). We made this choice because these regressions were based on the spatial occurrence statistics of our 16 object categories in the COCO dataset, which neither our models nor human subjects are trained on.

To directly test for your suggestion of a possible effect of intrinsic size in our behavioural data, we added a new GLM analysis (new Fig. 4D) that models the human ADMs using several predictors mirroring factors of IT organisation suggested in the past. These include real-world-size, animacy, and “spikiness”. We find that real-world size indeed explains a significant amount of variance (and

animacy plays the most important role, as also evident in a new hierarchical clustering analysis, see Fig. 4D).

Finally, we repeated our behavioural analyses using different stimulus sizes as inputs to our ANN models (see also our response to your comment 2.3). The pattern of our results remained consistent across various sizes, showing that All-TNNs outperforming other networks in matching human spatial biases is robust and not dependent on the chosen presentation size (see Supp. Fig. 23).

Actions:

- We have removed our old figure 4, which, as you mentioned, was particularly noisy, because it was only indirectly linked to our narrative.
- *Supp. Fig. 23:* We have shown that our behavioural results generalise to other stimulus sizes (see comment 2.3).
- *New Fig. 4D:* We have added a hierarchical clustering analysis and GLM analysis to show object-specific patterns in the human behavioural data, and explicitly test the effect of real-world size following your suggestion.

2.3 - On the other hand, have you observed whether the network develops size invariance? This is a property inherent to the dataset used, and should emerge as the network is trained.

Response: Thank you for this suggestion. We explicitly tested size invariance in our networks by testing performance while scaling the size of ecocet test images from 50%-200% of the original size (Supp. Fig. 23E). We find that All-TNNs are similarly size invariant to CNNs, as performance changes similarly for both models when changing the size of test stimuli.

In addition, we now reproduce our core behavioural results (both accuracy map agreement, and Accuracy Dissimilarity Matrix (ADM) agreement) with different object sizes, ranging from 50%-200% of the original size (Supp. Fig. 23A-D). We find that the pattern of results is stable, with All-TNNs outperforming control models in matching human spatial biases in human behaviour overall. This observation also suggests some size tolerance in our networks.

Please note that all our networks are trained with image data augmented with scaling of up to 33%.

Actions:

- *Supp. Fig. 23E:* We tested our networks' performance using different stimulus sizes.
- *Supp. Fig. 23A-D:* We tested how well our networks match spatial biases in human behaviour using different stimulus sizes.

2.4 - In order to make a controlled comparison of the performance of the convolutional network versus the proposed network, it would be useful to quantify the number of free parameters in each of the networks being compared.

Response: We have added a table describing all architectures in detail (including parameter counts) to the supplementary material (Supp. Table 1). This makes it clearer that all models have the same number of units, the same kernel sizes, strides, layer shapes, etc. However, the CNN models do not have the same number of parameters as LCNs and All-TNNs, because weight sharing strongly reduces their parameter count. Note that our focus is not on performance, but rather on topographic features and spatial biases, so the number of parameters is not directly relevant to our claims. However, for full transparency on the relative performance of the networks, we now report the classification accuracy of all networks as part of the main text (Fig. 1B).

Actions:

- *Supp. Table 1:* we have added a table that contains all architectural details for CNNs, LCNs and TNNs.
- *Figure 1B:* we explicitly reported on classification performance for all models.

2.5 - The definition of localization loss seems to derive from a heuristic (as the average of cosine similarity over horizontal and vertical directions) and its use needs to be justified. In particular, is this loss isotropic and does not introduce biases on the cardinals? Will other measures (e.g. the sum of the positive parts of cosine similarities) give similar results?

Response: Thank you for pointing out the need to better motivate the specifics of our smoothness loss function. We now expand on the specific methods, and added a control to verify the validity of these choices.

The smoothness loss is not explicitly isotropic, as we only use the ‘east’ and ‘south’ neighbours. The decision to use only the south and east neighbours stems from the observation that employing a symmetric neighbourhood would result in all cosine distances between units being counted twice (as cosine distance is symmetric). Avoiding these redundant calculations is beneficial in our computationally intense setup.

Our rationale behind using only the cardinal directions and excluding the diagonal neighbours from the loss computation was that the similarity between diagonally adjacent units is achieved indirectly: if the weights of unit A are similar to its eastern neighbour B, which in turn is similar to its southern neighbour C, then A and C are (indirectly) driven to be similar. Hence, we omitted the diagonals for the sake of computational efficiency.

To validate these choices we added a new control network incorporating all 8 neighbours (i.e., an isotropic loss including diagonals). This control network achieves comparable categorization performance and selectivity maps, corroborating the adequacy of our original assumptions, and suggesting that no major bias in the cardinal directions is created by using a non-isotropic loss (Supp. Fig. 25).

Finally, cosine distance was chosen in order to enforce similar directions in weight space, without enforcing identical norms. The suggestion of enforcing only some units to be similar (e.g. by having a

threshold on the cosine distance) is an interesting suggestion that we will consider in future work. Note that there is a large array of possibilities when moving in this direction (other distance measures, choosing which units get to affect their neighbours based on gradient estimates, etc). We therefore decided to stick to the straightforward use of minimising cosine distance between neighbouring unit weight vectors. We mention in the discussion that different ways of enforcing smoothness is an important area of future research.

Actions:

- *p.20 l.606ff*: We better describe how our loss is computed in the Methods section, and justify the choice of only using ‘east’ and ‘south’ neighbours.
- *Supp. Fig. 25*: We have included a new control model trained using an isotropic loss (i.e., all 8 neighbours).
- *p.17 l.544ff*: We discuss different ways of enforcing smoothness as an important area of future research.

2.6 - The development of topographical structure is linked to the development of spontaneous waves of activity in the form of retinal waves, which help to structure topographical relationships in the network. Do you think you could observe similar waves of activity in a continuously activated network?

Response: We agree that retinal waves are an interesting candidate for pre-structuring the topography of All-TNNs before the model is exposed to natural stimuli. We have added a discussion of these ideas to our manuscript. One example of notable work in this area that might be of interest is the following paper which introduces RNNs that develop interesting topographies and dynamics based on retinal waves.

Keller, T. A., & Welling, M. (2023). Neural wave machines: learning spatiotemporally structured representations with locally coupled oscillatory recurrent neural networks. In International Conference on Machine Learning (pp. 16168-16189). PMLR.

Actions:

- *p.16 l.499ff*: We have added a discussion of pre-training with retinal waves.

Minor:

3.1 - Figure 1, the RFs in layer #5 are not to scale,

Response: As the RFs in the first layer would be extremely small, we decided not to make any RFs in this figure to scale but rather decided for “symbolic RFs” that are accompanied by the exact numbers (inspired by a visualisation format customary in ML). Nevertheless, we have reworked the network

illustration in Figure 1 to improve clarity and describe all architectural details in the methods section and Supp. Table 1.

Actions:

- *Fig. 1:* We have added a new and improved version of the figure.
- *Methods:* We have described the model more clearly.
- *Supp. Table 1:* All hyperparameters of the networks are detailed.

3.2 - page 6: when giving the accuracy after low entropy lesion, it is indeed a "relative change in accuracy" (I guessed it from seeing it is over 100%). This is indeed explained later in the paper.

Response: We agree that this was not sufficiently clear. In response, we have decided to report the actual accuracies instead of relative changes in accuracy. The lesion study is now placed in the Supp. Fig. 12.

Actions:

- *Supp. Fig. 12:* We have reported the actual classification accuracy instead of the relative accuracy.

3.3 - I couldn't find a quotation in your paper describing the Mondrian mask synthesis method.

Response: We have added the citation to the relevant paper (Stein et al., 2011) to relevant places in the paper (Results section on behaviour and Methods). We have used code by Martin Hebart to generate the masks, which can be found on his website, which we now cite in the Methods section as well.

Stein T, Hebart MN, Sterzer P. Breaking Continuous Flash Suppression: A New Measure of Unconscious Processing during Interocular Suppression? *Front Hum Neurosci.* 2011 Dec 20;5:167. doi: 10.3389/fnhum.2011.00167. PMID: 22194718; PMCID: PMC3243089.

Actions:

- *p.10 l.290ff; p.25 l.817ff:* We have added the relevant citations to the manuscript.

Dear editor, dear reviewers,

Thank you for the opportunity to provide a new and strengthened version of our manuscript. We would like to express our gratitude for the many constructive and helpful comments. With this revision, we address all concerns raised by the reviewers, as detailed below in our point-by-point reply. The main improvements included with this revision are:

Major improvement 1: A large set of control models trained and tested

The reviewers highlighted several aspects for which additional control models could improve our manuscript. We conducted all of these controls, and more. They include

- (a) All-TNNs with various α values to explore the effect of the smoothness loss magnitude,
- (b) adding LCNs to the behavioural analyses (previously they were only included in the topographical maps section),
- (c) an All-TNN trained using a self-supervised objective (SimCLR) to test for the effects of training without supervision on the emergence of topography and behaviour,
- (d) TDANNs (a CNN modified to study neural topography by Margalit et al., 2023/2024) to test if this alternative approach also matches human spatial biases in behaviour,
- (e) an All-TNN trained using a spatially shifted version of ecoset to show that the topographic maps are driven by the location of task-relevant information in the training dataset, and
- (f) an All-TNN for which the smoothness loss is computed across all 8 neighbouring units to test if our approach of computing the loss for the 'east' and 'south' neighbours only is appropriate.

All of these new controls support and strengthen the paper's main claims, while further allowing us to specify which model aspects are important to mirror primate topography and behaviour.

Major improvement 2: Smoothness, performance, and energy efficiency

The reviewers asked us to expand our analyses of the impact of our smoothness loss on performance. Is there a trade-off between smoothness and performance? Could smoothness be seen as a regulariser that can improve performance? We now report the test performance of all our networks, and find that CNNs, LCN, and All-TNNs with various α values all reach comparable performance (with a slight advantage for CNNs). This raises the question of the computational advantages of topographic organisation in the brain. In previous work, topographic organisation has been linked to reduced energy costs of primate visual processing. In a new results section, we test for this possibility in our networks. Using a proxy of energy consumption in biological networks, we find that All-TNNs are more energy efficient than control models, and that regions with smoother selectivities consume less energy. This suggests that topography may be beneficial to reduce energy consumption, rather than to improve classification performance.

Furthermore, following reviewer and editor comments, we now include a more thorough investigation of the parallels between our models and ventral stream organisation, including a quantification of spatial feature smoothness, radial cluster sizes for low-level orientation selectivity and high-level category selectivity, estimates of spatial variability in feature selectivity with increasing eccentricity, and the developmental trajectory of topographic emergence across model training.

Major improvement 3: Reworked and improved behavioural results section

The reviewers asked for several changes of our section on spatial biases in object recognition behaviour. They pointed to a lack of clarity in certain subsections, raised limitations in the reliability of certain aspects of our results (especially the old Figure 4), and suggested additional control models. In response, we have thoroughly restructured and improved this section. We removed analyses that were not directly linked to our main claims (old Figures 4 & 7), and replaced them with new analyses that strengthen and deepen our central results about the match of All-TNNs to spatial biases in human object recognition. This includes

- (a) New hierarchical clustering and GLM analyses to show structure in our measured human spatial biases in object recognition,
- (b) a direct demonstration that our models can learn object-specific spatial biases,
- (c) the additional control models mentioned earlier (All-TNNs with various alphas, SimCLR-trained All-TNNs, LCNs, and TDANNs), and
- (d) analyses with different stimulus sizes, reproducing our main results.

While the main conclusions from our previous submission (the benefit of All-TNNs over the gold standard of CNNs) remain the same, this substantially reworked section offers a more complete and solidified picture of the alignment of our various models with human behaviour.

Further improvements

Aside from these main changes, to further support our conclusions, we added various quantitative measurements of topography in our networks, and generally improved our analyses, plots, and narrative. The reviewer comments have also helped us to better contextualise our work in the literature, and to provide an expanded Methods section and reworked figures that clarify our approach.

Detailed responses to each reviewer comment are provided below, following this format:

Text items in blue boxes are reviewer comments.

Response: Black text items are our replies.

Actions:

- We provide page and line numbers for the main relevant changes made to the manuscript.

Reviewer #1

1- Questions about the All-TNN model implementation

1A - Alpha hyperparameters

Figure 1 is very helpful, and in general your methods are very clear. To understand the results, I needed more details about the hyperparameter alpha values on the smoothness of the layers, how they were explored, selected, and how that impacts the outcomes. For example, the methods state: “we used a spatial loss of alpha=10 in all layers except the final, for which we used alpha = 10, due to increased smoothness observed in the higher visual cortex”. A couple typos on there, and also the same alpha value, so, hard to follow. *How were these alpha values set and how do they matter?* There is room to do a controlled rearing experiment here, with a suite of models varying in alpha that you could characterize. It also sounds like you needed to increase the alpha of late stages to make it ‘blobbier’? But maybe a supplemental figure just showing the range, from low and salt-and-pepper, to extremely high (e.g. very blobby), would be valuable. As it is now, it is just a little confusing what your commitment is to that alpha value, across layers, and what your search procedure for setting those was, especially as I assume your main topographic results depend on this parameter setting.

Response: Thank you for pointing out the need for more detailed explanations regarding the spatial similarity loss weight (i.e., the α hyperparameter). We did not mean to imply that there is a globally optimal value of α that we commit to strongly. Following your suggestion, we now include various α values ($\alpha \in [1, 10, 100]$) across all of our experiments. We justify the choice of these values in the expanded subsection on All-TNN architecture in the Methods. As detailed in response to your following comments, we find that α can be high without deteriorating classification performance, while impacting most aspects of topography and behaviour. We now discuss these effects throughout the Results and Discussion.

Finally, in the prior version of our work we multiplied the level of regularisation in the final layer by 10, motivated by the fact that higher visual cortex shows more smooth regions. However, we decided to change this to equal regularisation of all layers in the current version for simplicity and clarity.

Actions:

- All our results now include All-TNNs trained with $\alpha \in [1, 10, 100]$, to characterise how topography and behaviour change when α changes. These results are shown in all relevant main Figures, as well as all relevant Supplementary Figures, and are extensively discussed in the main text.
- Our networks are now trained with identical α in each layer.
- *p.19 l.567ff:* We have expanded the *All-Topographic Neural Networks* subsection in the Methods to include better explanation and justifications of our modelling choices.
- The Results and Discussion sections now explicitly address the effect of α .

1B - Spatial similarity loss

The way you implemented tuning spatial similarity loss, as far as I can tell, is that for each unit you compute the kernel weights correlation with the unit next to it to the right, and one below it, then you sum this. Then, you sum this across all units. If I got it correctly, this neighborhood function seems a bit asymmetric and small to me (just the next neighbors, and only two of them rather than, say, all units within some distance on the cortical sheet, including diagonal units). I know this is a huge space and you had to pick something, but I admit this choice of just two adjacent units surprised me a bit. I suspect that variations in the way you operationalize this L_s , e.g. larger neighborhood, would have big effects on the kinds of results you'd get. So, an explanation of why this choice (was it on purpose, and you explored others and this was good for reasons, or was this just a start) would help me as a reader, both in the methods/results and in general discussion. These implementation (along with the Alpha's) really seem like the key decisions you had to make and discussing them, and your commitments to them, would strengthen this work and clarify how future work could build off it.

Response: We agree that the choice of the neighbourhood in which the loss is computed was not sufficiently clearly motivated in the previous version. Our loss is computed by averaging the cosine distance between the kernels of all (south and east) neighbouring units in each layer. Then, the total loss is summed across layers. The choice of only using the south and east neighbours is based on the observation that if we took a symmetrical neighbourhood (for example the von Neumann neighbourhood, i.e., north, east, south and west neighbours), all distances between units would be included twice, since the cosine distance between units A and B is symmetric. This results in costly calculations that have no effect, given that the final loss is averaged across the units. We now motivate our analysis choices in the methods.

This, however, leaves open the possibility that including diagonal units may have an impact. Our reasoning was that the similarity between diagonally adjacent units is achieved indirectly: if unit A is similar to its eastern neighbour B, which is similar to its southern neighbour C, then A and C are (indirectly) driven to be similar. We chose this option because it has the advantage of being computationally more efficient. To check the validity of these choices, following your suggestions, we added a control that includes all 8 neighbours (i.e. Moore neighbourhood, which is isotropic and includes diagonals). This control network achieves comparable categorisation performance, and selectivity maps, confirming the adequacy of our assumptions (Supp. Fig. 25).

Note that we use periodic boundary conditions (i.e., the layer is wrapped in a toroidal fashion), meaning that the unit at the very bottom (resp. right) of the layer has its loss computed with the top-most (resp. top-left) unit of the layer. This is done to ensure that all units have the same number of neighbours, thus avoiding border effects. We clarified these aspects in the Methods, and added a Supp. Fig. 10 to show the boundary conditions.

Actions:

- *p.19 1.578ff*: We have better described how our loss is computed, and justify the choice of only using 'east' and 'south' neighbours.

- Supp. Fig. 25: We have included one model instance trained with a Moore neighbourhood (i.e., all 8 neighbours).
- Supp. Fig. 10: We have added an illustrative figure of how the periodic boundary conditions of the smoothness loss ensure that there are no boundary effects.

1C - Object classification results and smoothness

I found that I really needed a results section, after you describe the model, but before you talk about layer 1 orientation maps, that reports the results of the model training and smoothness objectives! (across all $N=30+$ models you have). What is the average object classification accuracy within a model type (and was there a lot of variance across the 10 seeds?). Next, you can even quantify the emergent smoothness in the LCNs (!!) and ask: what is the L_s of these models? (helpful in the context of the all-tnn models that are optimized for this directly). This just reports that your models trained well and did what you expected (and provides context for the next comment).

Response: Thank you for flagging the need for more in-depth reporting on network training results. As you suggest, we added a new results subsection that includes the classification performance (Fig. 1B), and spatial smoothness (quantified as $1/\text{smoothness loss}$) for all models (Fig. 1C), including LCNs, All-TNNs with various alphas, and CNNs. We find that All-TNNs achieve similar performance to LCNs, while developing spatially smoother weights (for all α values tested). This shows that All-TNNs successfully learn a large-scale image recognition task, while satisfying the smoothness constraint.

Actions:

- *p.4 l.101ff*: We have added a new section reporting on classification performance and smoothness for all models.
- *Fig. 1B & C*: We have added a new figure visualising classification accuracy and emergent smoothness for all models.

1D - Accuracy-smoothness tradeoff

You assert that the spatial similarity loss “forces the network to trade off between i) learning varied feature selectivity... and ii) preserving similar features selectivity between neighboring units.” However this claim is never shown or quantified. You could directly test for accuracy-smoothness trade offs: if you varied alpha, does the level of smoothness (L_s) really trade off with accuracy? (For a richer exploration, you could even let the Alpha’s-by-layer be learnable, and see what values are learned!). These analyses/variations are important, because they get at the theoretically important question of whether topographic tuning smoothness can be a useful inductive bias to help promote classification accuracy in the huge space of possible solutions. Or, alternatively, if in this way of operationalizing the

cortical formation as a joint loss function of categorization and smoothness, these constraints directly trade off.

Response: As covered in above responses, we analysed All-TNNs with varying magnitudes of the α hyperparameter. We find that increasing α does not substantially change classification performance, but does lead to smoother weights (see comment 1C) and smoother selectivity maps (see comment 2A). Hence, we agree that “tradeoff” is not the right term, as it suggests that smoother networks should perform worse. We have removed this term.

We think that the network, having to minimise both losses simultaneously, allocates more computational resources to task-relevant areas of the inputs at the expense of less relevant regions, in line with our analysis showing higher feature variability in the centre of the image (Fig. 2B). In other words, the network achieves a different balance between the two losses in different regions of the layer. We added new results to support this hypothesis, in which we trained an All-TNN on a spatially shifted version of ecoset. This leads to a corresponding shift in the spatial organisation of the network (see Fig. 2C and Supp. Fig. 11).

Since we have no evidence for an “optimal” α where performance is maximised, our smoothness loss cannot be seen as an inductive bias to promote classification performance. This raises the question of what the benefits of smooth topographical organisation are. We added a new section using a proxy of biological energy consumption, showing that, if All-TNNs were to run on biological wetware, they would be more energy efficient than CNN and LCN controls (see Fig. 3). In the discussion, we elaborate on how this may be linked to the aforementioned ability of All-TNNs to allocate computational resources to task-relevant regions.

Concerning the learnable smoothness loss: we agree that this is an interesting option that we intend to explore in the future together with other computational modes to obtain smoothness. In sum, our results suggest that the balancing of our two losses does not yield one optimal α value with optimal performance/smoothness. Therefore, we opted to show results for various α values throughout the manuscript (see our response to your comment 1A).

Actions:

- *p.4 l.101ff:* We have added a new section reporting on classification performance and smoothness for all models.
- *Fig. 1B & C:* We have added a new figure visualising classification accuracy and emergent smoothness for all models.
- *Fig. 2C & Supp. Fig. 11:* We have added a new analysis showing that shifting the input dataset leads to a corresponding shift in topographical organisation.
- *p.8 l.241ff; Fig. 3; Supp. Fig. 15:* We have added a new results section showing that All-TNNs are more energy efficient than controls.
- We have removed the term “trade-off”, and included a more nuanced discussion of the effects of the model balancing the dual loss throughout the manuscript.

I recognize there's a range of possible action items in the comments above. At a minimum, I would suggest reporting the classification accuracy of all the models and their spatial smoothness loss, either by varying alpha directly, or at least by describing your process for hyperparameter search over alphas. And, either test whether these trade off and by how much, or remove that assertion.

Response: Thank you for the comments, which have helped us clarify our approach in the paper, as described above.

1E- "Hypercolumn"

I believe you are using "hypercolumn" incorrectly throughout the manuscript, especially as you introduce this in the results section. A hypercolumn is a vertical column through the laminar structure of the cortex, within a single cortical area. These neurons along the depth axis of the cortex tend to have the same RF and tuning profiles. However, you're using it more like a way to refer to the hierarchical construction of receptive fields, where the increasing receptive field of a unit in layer 2 is linked directly to the connections from layer 1. This is not a hypercolumn structure. This is, maybe, a 'hierarchical spatial receptive field motif' to try to give it a name. Definitely remove "hypercolumn" and use a different term for this.

Response:

We agree that this term can be confusing and decided to remove it from the manuscript. In the previous version, by using the term "hypercolumn", we referred to the classic *ice-cube model* in which units that share a spatial receptive field in V1 are arranged according to their feature selectivity. Hypercolumns (which consist of multiple columns each) thereby have units selective to all orientations at a given location of the visual field. To quote Hubel (1982): a hypercolumn is a 1-mm block of V1 containing "all the machinery necessary to look after everything the visual cortex is responsible for, in a certain small part of the visual world". Analogously, in All-TNNs, each layer contains groups of adjacent units that share the same spatial receptive field location while being selective for varied features. These adjacent units sharing a RF are what we called a "hypercolumn" (note that the connectivity across network layers defines the spatial RF of a given model unit). In theory, each of these All-TNN "hypercolumns" could contain units tuned to all orientations, but the outcome of learning may not always satisfy this constraint, which may be confusing for readers.

Hubel, D. H. (1982). Exploration of the primary visual cortex, 1955–78. *Nature*.

Actions:

- We have removed reference to hypercolumns in the manuscript.

2. Topographic features of the ventral stream

2A - Lovely orientation results.

The main thing that lacked for me is quantification. You talk about orientation ‘smoothness’ but, e.g. didn’t necessarily quantify pinwheels or degree of emergent smoothness, compared to LCNs and CNNs. One idea: what’s the average radius you need to go from a unit to include units with all 8 preferences? This will also allow you to quantify that for All-TNNs the radius will vary as a function of center-to-edge, but won’t for LCN vary in an eccentricity dependent way for LCNs.

Response: We agree that our results can be strengthened by better quantification. To this end, we have implemented several analyses. First, as described in response to your comments 1A and C, we have quantified the emergent smoothness between the weights of neighbouring units. Furthermore, to quantify the smoothness of our orientation- and category-selectivity maps, we apply two methods: the analysis you suggested to determine cluster sizes, as well as a 2D Fast Fourier Transform (FFT). In the analysis quantifying cluster sizes, we find that the average cluster size of the orientation selectivity and category selectivity maps becomes larger with increasing α , while it is smaller in both CNNs and LCNs (Fig. 2A and 2D, Supp. Fig. 5). Plotting this cluster size against eccentricity, we find that, indeed, cluster size increases with eccentricity (Supp. Fig. 6).

In the Fourier analysis, we retrieve the power spectrum distribution of frequencies present in the selectivity map. A smooth map will display more power in low spatial frequencies than a salt-and-pepper map. We find that power in low spatial frequencies increases with increasing values of α , indicating a direct relation between the spatial similarity loss and amount of smoothness (Supp. Fig. 7).

Finally, to show that the All-TNNs are not only smooth, but also have discrete point discontinuities in orientation selectivity (akin to pinwheels), we applied the pinwheel analysis approach by Kaschube et al. (2008) (see Supp. Fig. 4). This analysis shows that, for All-TNNs, the number of pinwheels decreases with α magnitude. LCN ($\alpha = 0$) and CNN controls have an unreasonably high number of pinwheels, due to their salt-and-pepper orientation maps. Note that the common way of retrieving pinwheel density is by normalising pinwheel count by mms of cortex. However, since the exact number is dependent on a contingent decision of matching visual cortex distances to model layer size, we decide to refrain from this. Furthermore, we find that this analysis is very sensitive to noise, causing small clusters of units classified as pinwheels to appear. One way to resolve this is to artificially smoothen the map before performing the analysis; we refrain from doing this, too, and instead select the unit with the highest winding number in the Moore neighbourhood.

Kaschube, M., Schnabel, M., & Wolf, F. (2008). Self-organization and the selection of pinwheel density in visual cortical development. *New journal of physics*, 10(1), 015009.

Actions:

- Fig. 2A and 2D, Supp. Fig. 5: We have implemented the cluster size analysis you suggested.
- Supp. Fig. 6: We have quantified the variation of cluster size of orientation selective clusters per eccentricity.

- *Supp. Fig. 7*: We have quantified smoothness using power spectrum analysis of selectivity maps.
- *Supp. Fig. 4*: We have quantified the pinwheel discontinuities in our networks.
- *p.22 l.721ff*; We have described the methods for these new analyses in the Methods section.

2B - Cortical magnification claims are incorrect.

The way you linked these results to the ‘foveal bias’ confused me—you claim that “cortical magnification emerges”. But, I think there is a deep mismatch here. To expand on this: your first model layer map has units which each see a FIXED size of the visual field. There is no oversampling at the center of the image through a fovea. And, the definition of cortical magnification is that a disproportionately large area of the visual cortex is devoted to processing information from the central region of the visual field (the fovea) compared to the peripheral regions. Thus, your architecture does not have cortical magnification. So, claims to this end need to be removed. What your model DOES show is that, given an architecturally EQUAL sampling of the visual input with even sized receptive fields across the ‘visual field’, that more tuning variation emerges for processing central image statistics than for peripheral image statistics. This outcome implies a more normative possible claim, which is that if one wanted to have similar ‘smoothness’ of orientation tuning across the cortical sheet, one would need to over-represent the fovea and under-represent the periphery, and that would presumably still capture the image statistics well for the task. So, I think it’s fair to claim that your map of orientation gestures at an implication like this, but does not show emergent cortical magnification.

Response: We agree with your characterisation of the effects observed; our vanilla architecture does not have structural cortical magnification, as it has the same number of units per degree of visual angle at all eccentricities. Our claim, and this is now clarified in the text, is that All-TNNs are able to allocate more feature variety to relevant portions of the visual field at the expense of the spatial smoothing loss. We implemented several changes and a new analysis to better support this claim.

First, we now include results showing that shifting the training dataset such that there is more classification-relevant information in the bottom right quadrant, leads to a corresponding shift in the location where feature variability is increased (Fig. 2C, Supp. Fig. 11). This confirms that the increased feature variability is linked to processing task-relevant image statistics.

Second, and perhaps more importantly, given that All-TNNs have more varied feature detectors in regions with more task-relevant information, we tested whether units in regions with redundant selectivities can be lesioned without impacting task performance. Unlike LCNs and CNNs, All-TNNs allow for the smoothness of the feature maps to act as reliable proxy for which units can be lesioned. Once lesioned, All-TNNs, with minimal loss in performance, use more units in locations with task-relevant information (Supp. Fig. 12). The resulting model is reminiscent of cortical magnification in the primate visual cortex, where more neurons per degree of visual angle in regions process foveal information. We clarified the interpretation of this result in the results and discussion.

Actions:

- *Fig. 2C & Supp. Fig. 11:* We have included new results showing that the region with increased feature variability is driven by the location task-relevant information.
- *p.7 l.190ff:* We have clarified the presentation of our lesion study in the results.
- *p.15 l.457ff:* We have clarified and rephrased the interpretation of our results related to cortical magnification in the discussion.

2C - Category-selectivity results.

Unfortunately, when I look at Figure 2, layer 6 visualization, I don't easily see a 'face-selective area' or a 'scene-selective area'. E.g. these do not look like what you get when you make face-selective contrasts and can define face and scene region of OTC. I feel like this should be acknowledged. These maps are in the right direction (e.g. relative to LCNs), but claims of this map forming "category-selective areas" feel to me to be unwarranted by my inspection of the results. Either a case should be made, ideally in a quantified way, that these are 'areas' in some sense, or this claim needs to be significantly weakened. Certainly, for example, the category-tuning clustering is more clumpy than in the LCN, again would be nice if this were quantified.

Response: We agree that a quantification of the category clustering will improve our results, and that our wording hinted at an overly strong interpretation of our findings in terms of brain areas. We now address both of these concerns.

We do not claim that these category-based clusters are directly comparable to the regions found in visual cortex in terms of relative size, shape or distribution, etc. Instead, we claim that both All-TNNs and visual cortex display tuning-specific clustering. We have quantified this clustering in the final layer using the aforementioned methods (power spectrum analysis and cluster size quantification, see Fig. 2D, Supp. Fig. 5B; Supp. Fig. 7B). This improves on our previous analyses by quantitatively demonstrating that category-selective units are more clustered in All-TNN than in control models, and that higher α leads to more clustering.

In the textual revision of the manuscript, we made sure to avoid claims of a direct match to face/tool/scene areas in the brain.

Actions:

- We have verified that we do not refer to direct correspondence with category-selective regions in the primate visual cortex. We only claim that the existence of category-selectivity clusters is reminiscent of the brain.
- *Fig. 2D:* We have quantified category-selective cluster sizes throughout models.
- *Supp. Fig. 5B:* We have quantified the differences of category selective cluster sizes between models.
- *Supp. Fig. 7B:* We have quantified smoothness using power spectrum analysis of category selectivity maps.

3. Behavioral Results

3A - control models.

The LCN models absolutely need to be included in these experiments (Fig. 4 and Fig. 5). They are critical comparisons! All-TNNs should have the capacity to succeed, while LCNs should fail, based on your accounts. If the LCNs succeed, we need to know and understand those results! Similarly, the CCNs also succeed to some degree above chance it seems, at picking up on these spatial bias. Why? How? Do they? That seems worth discussing.

Response:

Before diving into the addition of LCNs and other control models, we would like to point out that we have decided to remove the old Figure 4, which was based on the spatial occurrence statistics of our 16 object categories in COCO. We made this decision for several reasons. First, our models are not trained on COCO, and neither are humans. Therefore, we realised that these COCO based statistics were at best indirectly linked to our results. Second, after implementing the shuffling you suggested in comment 3C, we found that it made little difference. This shows that category-specific differences in COCO spatial occurrence statistics were not relevant for humans and ecoset-trained networks. Please note that this does not invalidate our subsequent results, because they are not based on COCO occurrence statistics. Nevertheless, it makes the old Figure 4 largely irrelevant, and potentially confusing, which motivated us to remove it.

We agree that the LCN models are relevant to the behavioural comparisons, thank you for pointing this out. We have thus included the LCN in all of our main experiments. The LCN results match our hypotheses: they can match human behaviour better than CNNs, but not as well as All-TNNs.

In addition, in response to reviewer 2, we have also included TDANNs (Margalit et al., 2023/2024) as new control models for the behavioural experiments (Supp. Fig. 20), as well as an All-TNN trained with the SimCLR self-supervised learning objective (Fig. 5; Supp. Fig. 24). Together with our exploration of various α values, these new control models offer a more comprehensive picture of the abilities and shortcomings of the different model architectures, hyperparameters, and training objectives.

As you mention, CNNs do have spatial biases to some extent. This is linked to the fact that the last layer of CNNs, despite not having a spatial dimension itself, is free to "listen" to a spatially distinct set of units in the layer below, and hence can represent spatial biases in behaviour, too. Similarly, the fully connected readout can pick up on spatial biases, too. We expect these features to provide the CNNs with limited spatial priors. LCNs and All-TNNs, in contrast, are free to learn different features at different spatial locations throughout the network, which may explain the stronger spatial biases they exhibit. We discuss this in the relevant part of the Results section.

Margalit, E. et al. A Unifying Principle for the Functional Organization of Visual Cortex. 2023.05.18.541361 Preprint at <https://doi.org/10.1101/2023.05.18.541361> (2023).

Margalit, E., et al. A unifying framework for functional organization in early and higher ventral visual cortex. *Neuron* (2024).

Actions:

- *Old figure 4:* we have removed the analyses that involved the positional occurrence statistics of MS COCO.
- *New figure 4:* we have included the LCN in all experiments, including the behavioural experiment.
- *Supp. Fig. 20:* We have added two more control models; namely TDANN trained on object categorisation (one instance) and TDANN trained using SimCLR (5 instances).
- *Fig. 5; Supp. Fig. 24:* We have added a figure that covers all analysis performed on All-TNN trained with SimCLR.
- *p.13 l.383ff:* We now discuss the limited spatial bias in CNNs.

3B - Data reliability. [see edit]

As a first step of the data analysis, I would like to see the noise ceiling of the human data (before you verify these maps correspond to the objects occurrence statistics) For example, in Figure 3b, you show several templates for 16 objects, but I do not have a good sense from the results how much this variation across items is stable or noise. Can you do an analysis that quantifies this? some ideas: reporting the split-half reliability of the pattern? Maybe showing that a pattern is more similar to itself than other items on average across split halves? Maybe a linear mixed effects model statistic where you have to model accuracy with position as a predictor and a random effect of subject, and you can include item as a predictor as well, and item*position, and show that including item*position interactions are significant? Further, situating the results in Figure 4a and 5b with respect to the data reliability would help contextualize the results.

Edit: I just saw in the methods that you do in fact compute the noise ceiling and you are already showing reliability adjusted correlations! Definitely report the noise ceiling first please, and state clearly in the results you're showing noise adjusted correlations. I frankly would prefer to see the raw correlation values with the noise ceiling of the data indicated on the same axes. If the data are noisy, make the case for why the effect is important even if it's noisy, and/or qualify the results and point to important directions for future work. Don't hide the noise.

Response: We performed multiple analyses to estimate the data quality and variation across individuals. First, we now provide two different analysis approaches: once testing the individual model seeds against the average human behaviour per category (Fig. 4E, 4F), and once testing every model seed against every individual separately together with an explicit report of the noise ceiling (as requested; see Supp. Fig. 21). Both analysis approaches confirm the main results, and both are referenced in the main paper.

We perform multiple additional analyses to further test the reliability of our behavioural data. First, we performed a hierarchical clustering analysis and a GLM analysis on the ADM computed from the average human category accuracy maps. We use several predictors inspired by large-scale organisation principles in IT, including real-world-size, animacy, and “spikiness” (main text and Fig. 4D). These show that, indeed, real-world size and animacy explain significant unique variance in the behavioural data, while spikiness does not. Second, we directly test object-based differences in behaviour by statistically comparing the accuracy maps of all pairs of categories, as now reported in the Supplemental Materials. Together, these results indicate that humans do express reliable positional effects on accuracy depending on the object category seen.

Actions:

- *p.25 l.838ff*: We have extended the Methods describing the analysis approach in detail.
- Fig. 4E, 4F: We report on the analysis comparing individual models to average human behaviour.
- Supp. Fig. 21; Supp. Fig. 22B: We have added a figure showing the analysis comparing individual models to individual human behaviour.
- Fig. 4D; Supp. Fig. 18: New hierarchical clustering and GLM analyses showing structure in the human ADM.
- Supp. Fig. 17: We have added a statistical comparison of averaged object accuracy maps to show there are significant object-specific differences in human behaviour.

3C - Shuffle control

For figure 4, in addition to adding the LCNs, it would be useful to add a spatially shuffled prior as a control analysis, to get a sense of where the CNN is falling with respect to the shuffled control.

Response: As described in our response to the previous comment 3A, we have removed the analyses present in Figure 4.

Actions:

- *Old figure 4*: we have removed the analyses that involved the positional occurrence statistics of MS COCO.

4. Discussion/Abstract

Based on the above comments, I recommend: qualifying the claims about the emergent organization of category-selective regions, clarifying the logic of the cortical magnification argument, and removing

the claims about a trade-off between classification and smoothness unless this is directly shown with new analyses.

Response: Thank you for the many great suggestions, all of which we have addressed, as detailed above.

5 Minor/Optional/Potential small edits for increased clarity

5A. Terminology: LCNs: “local” is confusing to me here, because I usually think of local like local WITHIN A MAP/AREA, local connectivity is near-neighbor (in the cortex) connectivity. I think maybe your LCNs could be called ‘pure-hierarchy’ models or some alternative?

Response: We use the term “locally connected network” because it is standard usage in machine learning (for example, our LCNs are implemented using tensorflow’s “LocallyConnected2D” layer). To prevent confusion with lateral connectivity we have included a more detailed description of what differentiates LCNs from CNNs and All-TNNs throughout the paper, for example by pointing out that a LCN has technically the same architecture as a CNN but without weight sharing, and that it is equivalent to an All-TNN but with $\alpha=0$. Furthermore, we have added subsections in the Methods section that describe each model and their differences in more detail.

Actions:

- *p.19 l.567ff:* We have added a section describing each model and their differences more clearly in the Methods.
- We have clarified the differences and similarities between LCN and other models better throughout the paper.

5B. Intro: “locality” desideratum: I’d say ‘units need to have local receptive field *feature weights* that are learned individually and not enforced to be exact duplicates of other RFs’. This is to help clarify the difference between feature tuning (flexibly learned) and spatial receptive fields (fixed by the architecture)

Response: We agree, and have modified this point in the paper following your suggestion. We also rephrased the term “desiderata” to prevent misunderstanding that they provide a complete list of requirements.

Actions:

- *p.1 l.61ff:* We have updated the relevant sentence according to your suggestion.

5C. Methods: Just checking: the cosine difference between the weight kernels, here this would be calculated over say, a flattened 27x27 set of learned weights if that is the size of the kernel? (rather than, say, over the activation profiles of a test set of images, which is more like what is done in the empirical literature noting smoothness, because we don't have access to the connectome weights). Clarifying this in the methods might be valuable.

Response: You are correct, we compute the loss based on the weights rather than on the activity patterns. For each unit, the $kernel_size \times kernel_size$ weight matrix is flattened to a $kernel_size^2$ vector, and cosine distances are computed between such vectors. As you mention, we do not have access to the full connectome of the human brain, but neural network modelling gives us the opportunity to investigate the effect of constraining neural connectivity during learning. We implemented several changes in the methods to clarify how the loss is computed.

Actions:

- *p.19 l.578ff:* We have updated the Methods section to be more clear about the calculation of the smoothness loss.

5D. Figure 2. How are you plotting CNN preference maps, as in 2a. Are you just spatializing the features in a systematic grid? Maybe say this in the caption?

Response: We plot the CNN and LCN maps with the same procedure that is used to create the 2D "cortical sheet" of All-TNNs. This procedure simply involves reshaping the 3D layer to a 2D $\sqrt{channels} * height \times \sqrt{channels} * width$ layer. This means that each channel is reshaped to a 2D $\sqrt{channels} * \sqrt{channels}$ square of units. Upon your feedback, we have clarified this in the methods, and included a new subsection ('LCN and CNN topography visualisation') that explains how we create 2D maps from the 3D layers of CNN and LCN. We now clarify this in relevant parts of the text, such as the caption of figure 2.

Actions:

- *Fig. 2:* We have updated the caption to clarify how CNN and LCN maps are visualised.
- *p.19 l.573ff:* We have expanded the Methods section on the construction of the 2D sheet in All-TNN.
- *p.21 l.668ff:* We have added a new Methods subsection ('LCN and CNN topography visualisation') that clarifies this procedure and where it is used.

5E. Figure 2. You might clarify how you calculated entropy in a sentence in the main text, this term caught me off guard while reading through.

Response: We have clarified this in the text by referencing the type of entropy (Shannon entropy) in the results and extending the relevant Methods subsection.

Actions:

- *p.7 l.176ff:* We have added mention of Shannon entropy and clarified how to interpret it in our context.
- *p.22 l.710:* We have specified the exact function we use in the Methods section.

5F. Figure 4A: it took me a while to parse this diagram. I think it's because the 16 model object maps are arranged vertically and the 16 human object maps horizontally in a matrix, but you're only doing the corresponding correlations (and not making and rdm? Unless I really missed something). Figure 4b, in truth I struggled with this analysis and figure.

Response: We thank you for pointing out that the current arrangement of the first panel of plot 4A is confusing. As mentioned in response to your previous comment 3B, we have removed the old Fig. 4 completely. Previously, we used the same confusing arrangement for the old figure 5A, and have adapted that now to be clearer (new Fig. 4D in the current manuscript).

Actions:

- We have removed the analysis on behaviour-to-object-occurrence-statistics that was previously presented in the old Figure 4.
- The diagram in the new Figure 4D had a similar layout as the one referenced in this comment. We improved it following your comment.

5G. Results order: I got confused by the presentation of behavior-to-object-occurrence statistics rather than behavior-to-model accuracy. Either these could be switched, OR, a little signposting ahead of time might help, because the results header really leans into the behavior-to-model accuracy as the key result.

Response: We have removed the old Figure 4 (i.e., the behaviour-to-object-occurrence-statistics, reasons described in detail in response to your comment 2B). The storyline is hopefully clearer now.

Actions:

- We have removed the analysis of behaviour-to-object-occurrence-statistics, streamlining the flow of our behavioural results.

5H. ADMs results section and stereotypical unit results section. In truth I found these sections a little hard to follow. Maybe separating out logic of the analysis and methods details in different paragraphs could help?

Response: Thank you for pointing out that these sections were not clear. To address this issue, we have firstly decided to remove the section on stereotypical unit activation (old Fig. 7). The idea was that All-TNNs may develop, for example, a “bus subnetwork” during training, and this subnetwork is driven by buses presented at expected locations (dataset statistics). We attempted to locate this subnetwork in the last layer by using the average activation map for buses in the ecoset test set. We thought that, when the bus is presented at unexpected locations, this subnetwork is not engaged, leading to poor performance, thus tying topography and behaviour together. However, when testing LCNs with this analysis we found that they showed the same pattern as All-TNNs, due to their ability to also learn different weights at different locations. Still, All-TNNs better match human visual behaviour. This made us realise that this analysis could not explain the true reason why All-TNNs better match human spatial biases compared to control models, and hence decided to remove this analysis altogether.

Second, removing the old Figures 4 and 7 has allowed us to significantly streamline our storyline, while better motivating, explaining and expanding our core behavioural analyses. Specifically, we added analyses showing that there is object-specific structure in biases in human behaviour (Fig. 4D; Supp. Fig. 17; Supp. Fig. 18), validating the relevance of our behavioural dataset. Then, we proceed in two steps. a) we show that All-TNNs best capture human object independent spatial biases by directly correlating the model and human accuracy maps (Fig. 4E), and b) we show that All-TNNs with $\alpha = 10$ best capture human object dependent spatial biases by correlating Accuracy Dissimilarity Matrices (Fig. 4F). These core results are further strengthened by the inclusion of new control analyses. This includes showing that TDANN control models are outperformed by All-TNNs with $\alpha = 10$ in matching human behaviour (see our response to your comment 3A; Supp. Fig. 20), showing that the All-TNN with $\alpha = 10$ trained using a shifted dataset (see our response to your comment 1D) fails to match human behaviour likely due to misaligned topography (Supp. Fig. 11), and a reproduction of our results using different object sizes (Supp. Fig. 23). Finally, we now conduct our behavioural analyses using SimCLR-trained All-TNNs (Fig. 5).

Actions:

- *p.10 l.282ff*: We have improved the narrative and explanation of the entire behavioural results section.
- We have removed the section on stereotypical unit activation (old Figure 7)
- *Fig. 4, Fig. 5, Supp. Fig. 11, Supp. Fig. 17, Supp. Fig. 18, Supp. Fig. 20, Supp. Fig. 23*: We have added several new analyses and controls to validate our behavioural data and to support our main claim that All-TNNs better match spatial biases in human vision

Reviewer #2

1. The topographic neural networks proposed by Lee et al (2020) [Editors note - the referee has clarified that this refers to Lee, H., Margalit, E., Jozwik, K. M., Cohen, M. A., Kanwisher, N., Yamins, D. L., & DiCarlo, J. J. (2020). Topographic deep artificial neural networks reproduce the hallmarks of the primate inferior temporal cortex face processing network. bioRxiv, 2020-07], which similarly assumed 2D organization of neurons and demonstrates topographic organization, should be included in the evaluating and referenced appropriately as related work. Differences to Lee et al should be pointed out to clarify the novelty of the work.

Response: We agree that the work on TDANN, developed in parallel to ours, is relevant. Following your suggestion, we have included the most recent version of the model (as presented by Margalit et al., 2023/2024, and previously Lee et al., 2020), trained both with supervised and self-supervised objectives, as control models for our behavioural analyses (Supp. Fig. 20). We find that these models mirror human spatial biases in visual behaviour less well than All-TNNs. Please note that given the importance of dataset statistics in our results, the comparison is limited by the fact that the publicly available TDANNs are trained on Imagenet, instead of the ecologically-motivated ecoset that our models are trained on. While All-TNNs outperform TDANN in their published form, we therefore decided to include the comparison as a supplement, and mention this limitation in the discussion.

In addition to the inclusion of TDANNs in the results, we discuss the overall class of topographical models both in the Introduction and Discussion sections. We argue that the central benefit of All-TNNs is that they offer powerful topographic models capable of performing complex tasks, while entirely avoiding weight-sharing. We thereby differentiate our model from TDANN, which, under the hood, is a spatially embedded CNN with weight sharing and thus cannot directly model spatial variance in topography and behaviour. While we do think that CNNs can provide important insights into functional organisation, the assumption of identical features across the whole visual space is biologically unrealistic. As we now mention in the discussion, a large-scale and well-controlled comparison of diverse topographic networks will be important in future work.

Lastly, we reference Lee et al., as you suggested. While we initially cited the most recent publication on TDANN by Margalit et al., the early and influential efforts by Lee et al. in the domain of modelling cortical topography are now also included to properly contextualise our contributions.

Margalit, E. et al. A Unifying Principle for the Functional Organization of Visual Cortex. 2023.05.18.541361 Preprint at <https://doi.org/10.1101/2023.05.18.541361> (2023).

Margalit, E., et al. A unifying framework for functional organization in early and higher ventral visual cortex. *Neuron* (2024).

Lee, H., et al. (2020). Topographic deep artificial neural networks reproduce the hallmarks of the primate inferior temporal cortex face processing network. Preprint at bioRxiv, 2020-07.

Actions:

- *p.10 l.306ff; p.20 l.636ff; Supp. Fig. 20*: we have included TDANN in our evaluation of correspondence to human behaviour.
- *p.1 l.45ff*: We have more extensively discussed the differences to TDANN and other topographic CNNs in the introduction.
- *p.17 l.523ff*: We have contrasted our approach with several other approaches from the literature on topographic models in the discussion.
- We now include Lee et al. (2020) in our references, in addition to Margalit et al. (2023 and 2024).

2. The author's hypothesize that the center-periphery organization (e.g., cortical magnification) of the All-TNNs is likely emergent from the greater presence of object-relevant information in the center of the image. I am not fully convinced of this, especially since similar properties may emerge in LCNs if this were the case. Could this instead be an artifact of the spatial similarity loss? This loss function seems to suggest that central weight filters are implicitly regularized by all neighboring filters (above, below, left, and right), whereas border filters near the edge are not regularized by neighbors in all directions. The authors' claim may be supported better by demonstrating that this center-periphery organization emerges when trained on an augmented dataset that shifts the main image content outside of the image center or by training on a dataset where objects are not commonly centered in the image.

Response: Thank you for pointing out that we were not clear enough in explaining how we rule out the possibility that the centre-periphery organisation may be caused by border effects. As we now clarified in the results and methods, we apply the loss calculation in a toroidal fashion. This means that, a unit at the very bottom (resp. left) of a layer has its loss computed with the unit at the top (resp. right) of the layer. This means that all units are regularised to the same extent, avoiding border effects as an artefact of the smoothness loss. We added a new Supp. Fig. 10 to make this point clear, and show that the toroidal border conditions work as expected.

Further, thank you for the great analysis suggestion. To show that the increased feature variety in the foveal region is caused by unit weights adapting to the spatial statistics of the dataset, we have trained a control All-TNN on a shifted version of ecoset, where all images are displaced to the bottom right (see Fig. 3C, Supp. Fig. 11). In this model, the region with high feature variety is correspondingly shifted to the bottom right as well.

Finally, please note that we only use the "bottom" and "right" neighbours, and not all four direct neighbours. This is done to avoid unnecessary computations, since the loss is symmetric (i.e., the cosine distance between A and B is the same as the cosine distance between B and A). We have added a detailed discussion of this choice in the methods. In addition, we have added a new control where the loss is computed for all eight neighbours to fully rule out any deleterious effect of our neighbourhood choice (Supp. Fig. 25). As expected, this control yields similar results as our main model.

Actions:

- *p.7 l.183ff; p.19 l.581ff*: We now more explicitly mention that the smoothness loss is applied in a toroidal fashion in the Methods section and throughout the relevant sections on the topographical organisation of All-TNNs.
- *Supp. Fig. 10*: We have added a figure visualising that the features of All-TNN are spatially continuous.
- *p.7 l.185ff; Fig. 3C; Supp. Fig. 11*: We have trained an All-TNN on a shifted dataset and show that its feature organisation correspondingly shifts.
- *p.19 l.578ff*: We have extended the methods section where we describe the neighbourhood in which we compute the loss.
- *Supp. Fig. 25*: We have trained a new control model in which the smoothness loss is computed over all 8 neighbours.

3. In the lesion studies (which aim to support the greater importance of foveal processing in All-TNNs), LCNs and CNNs do not have high-entropy and low-entropy regions. Meanwhile, the All-TNN high-entropy lesion region contains the image center, which has more object relevant information, and is therefore more likely to affect classification performance. If the authors were to lesion the same image portions and feed them to the CNN and LCNs, I would imagine more similar observations of post-lesion classification accuracy would be observed across the three models. A more fair comparison may show the accuracy of LCNs and CNNs when the same high-entropy region from the All-TNNs is used as the lesion area.

Response: We agree that the framing of our lesioning study was unclear (please note that we have moved this figure to Supp. Fig. 12). Our claim is that All-TNNs exhibit focused allocation of resources to information-rich regions, including increased feature variety. This is supported by showing that All-TNNs are the only model for which the diversity of feature selectivities can be used as a proxy for which units can be lesioned without strongly deteriorating performance. The same findings hold for the All-TNN trained on a shifted dataset (mentioned in response to your previous comment 2), which develops a corresponding shift in the region with higher feature variability. Again, this shifted feature diversity can be used as a proxy for lesioning. In contrast, CNNs, by definition, fail to express this feature due to weight sharing (we hence removed them from the analysis). LCNs do not develop this characteristic during training, as reported. Together our results support the conclusion that feature variability is driven by dataset statistics in All-TNNs but not in control models, and is a good proxy for which units can be lesioned. Other models may exhibit similar drop in performance if one used the All-TNN maps to guide the lesioning, but the point made is that the All-TNNs themselves can discover those informative regions, whether the control models can or do not.

The view that All-TNNs exhibit focused allocation of resources in information-rich regions is furthermore strengthened by two new results. First, as we now quantify rigorously, All-TNNs have more varied weights in information-rich regions in their layers than control models (Supp. Fig. 2). Second, we added a new section where we quantify energy consumption throughout the network and at different locations within the model layers (Fig. 3; Supp. Fig. 15). We find that All-TNNs

consume more energy in the region with higher feature variability. This focus of energy consumption also shifts for the All-TNN trained with the centre of the images shifted.

Actions:

- *p.7 l.190ff; Supp. Fig. 12:* We have clarified the interpretation of the lesioning results both in the results and in the discussion.
- *Supp. Fig. 12:* We replicated the lesioning results in the All-TNN trained on a shifted version of ecoset.
- *Supp. Fig. 2:* We have included a quantification of weight smoothness throughout layers of all main networks, which also shows a focus on regions with task-relevant information.
- *p.8 l.241ff; Fig. 3; Supp. Fig. 15:* We have added a new analysis of the energy consumption throughout layers of all networks, again showing an increased allocation of resources in regions with relevant information.

4. In the behavioral bias analysis, positional occurrence frequency and uncertainty are computed from COCO images and it is assumed that stereotypical object locations could be computed from these images. Images taken by humans, however, naturally have a bias towards centering an object of interest near the center of the image and may therefore not actually contain representative positional occurrence frequencies as would be observed through the eyes of a human.

Response: We agree with your concern that the statistics of COCO images may not reflect the human visual diet (and our models are trained on ecoset and not COCO). As a result, the COCO based statistics were, at best, indirectly linked to our results. We have hence decided to remove the old Figure 4, which was based on the spatial occurrence statistics for our 16 object categories from COCO.

The rest of our behavioural analyses is not subject to this concern, as it is based on our experimental setup with objects presented on a 5x5 grid for both humans and models. With regard to the experimental human data, we added new hierarchical clustering and GLM analyses to show structured spatial biases in human behaviour (Fig. 4D), in line with previous work, and validating the value of the data for our purposes.

Actions:

- We have removed the analysis linking behaviour to COCO occurrence statistics that was previously presented in Figure 4.
- Fig. 4D: New hierarchical clustering and GLM analyses showing structure in human spatial biases.

5. This may simply be my misunderstanding, but further clarity on how 3D image representations (height, width, and channel) are “unfolded” into 2D neural sheets would be appreciated. The authors state that 3D representations of the form height x width x channel are unfolded into 2D representations of the form $(\sqrt{\text{channels}} \times \text{height}) \times (\sqrt{\text{channels}} \times \text{width})$. How are channel dimensions incorporated into height and width dimensions while preserving spatial information in the image representations?

Response: We improved our description about how the 2D sheet is created from the 3D structure. The unfolding procedure reshapes the channel dimension at each spatial location into a $\sqrt{\text{channels}} \times \sqrt{\text{channels}}$ 2D square at that location, thus preserving retinotopy. Hence, the $\text{height} \times \text{width} \times \text{channels}$ 3D layer is unfolded into a $\text{height} * \sqrt{\text{channels}} \times \text{width} * \sqrt{\text{channels}}$ sheet. This unfolding procedure is applied to All-TNNs for calculation of the smoothness loss, and for visualisation of the topographic feature maps of all models. We clarified these aspects in the expanded Methods section.

Actions:

- *p.19 l.568ff:* We have clarified the reshaping procedure in the Methods section on All-TNN architecture.
- *p.21 l.668ff:* We have added a Methods section (*‘LCN and CNN topography visualisation’*) that describes how we use the same reshaping procedure to visualise other models.

6. The authors should provide architectural information about the CNN. They mention that the CNN had a similar number of parameters, but should also include information about filter sizes, channels, etc.

Response: Agreed. We have added a table describing all architectures in detail to the supplementary material (Supp. Table 1). The CNN does not share the same number of parameters as LCNs and All-TNNs, because the weight sharing reduces those significantly. All models have the same unit numbers and kernel sizes, and hyperparameters. Random seeds are also controlled across architectures.

Actions:

- *Supp. Table 1:* We have added a table that describes all architectural details for CNN, LCN and All-TNNs.

7. The authors should consider showing object classification accuracy of each evaluated model on ecoset. Although it is not a motivating aspect of this paper, it could help the audience better understand the applicability of these networks to more general computer vision tasks.

Response: Done. We now report on the classification performance and weight smoothness of our main models (CNN, LCN and All-TNN with varying magnitudes of smoothness loss) in a newly added subsection in the Results. All models all reach similar classification accuracy.

Actions:

- *p.4 l.101ff:* We have added the section “*All-TNNs successfully classify natural images with smooth weight topography*” that reports on the classification accuracy of all models.

Reviewer #3

1 - First of all, a main point: through its supervised nature and being dedicated to categorization, this model stands out from similar models of self-organized learning in topographical maps. However, the emergent properties, particularly of the first layer, are similar to some of these models (such as <http://yann.lecun.com/exdb/publis/pdf/koray-cvpr-09.pdf>, doi:10.1016/S0042-6989(01)00114-6 or doi:10.1371/journal.pcbi.1010270). It would therefore be crucial to differentiate what makes this supervised model different from these unsupervised models of topographical self-organization. Indeed, a crucial difference lies in the definition of a cost function for the retinotopic organization observed in primates. Self-organizing models propose, for example, a metabolic-type optimization (e.g. of wiring length) or in relation to the efficiency of local information coding (e.g. sparseness). Your study could thus propose an alternative with a cost function linked to connection locality and categorization performance, as shown in your perturbation study. This point could be put forwards to discuss the origin and function of cortical topography.

Response: Thank you for the positive evaluation, we agree with your suggestion that our work should be better situated in the literature, particularly the unsupervised models you mention. We implemented several improvements to address this comment.

First, we expanded the discussion relating our approach to other models in the field (citing the relevant papers that you mention). We argue that our model is unique in that it is both explicitly topographic, and capable of performing complex image processing tasks on large datasets. Most other topographic models instead fall in one of two families. The first family, similar to the approaches you mentioned, comprises models that use relatively simple features from which topographies are learnt, usually with some form of unsupervised learning (for example, SOMs, sparsity constraints, etc.). While these models are usually explicitly topographic (in that there is no weight sharing, or a priori enforced structure), they are unable to deal with (complex) visual tasks. This limits them in modelling naturalistic human vision. For example, these models could not easily be tested on our behavioural experiment. The second family encompasses more recent topographic models that use CNN backbones. These models have the advantage of being able to perform complex tasks. However, they ultimately rely on weight sharing, which has several limitations. Weight sharing is not biologically realistic (as it is not the case in the brain that changing one synapse automatically changes synapses at all spatial locations identically). In addition, weight sharing enforces that the same features are extracted across the visual field, making it difficult to model non-homogeneous topography, as well as spatial biases in behaviour. We confirmed this by showing that a recent topographic model based on a CNN (TDANN, both supervised and unsupervised; see Supp. Fig. 20; Margalit et al., 2023/2024) does not develop human-aligned spatial biases in behaviour. Overall, All-TNNs differ from other existing topographic models, in that they are able to learn to perform complex tasks on natural visual input while being mechanistically truly topographic in all layers.

Second, we added All-TNNs trained using the SimCLR self-supervised objective as a way to test for the importance of category training. We find that these self-supervised All-TNNs develop similar topography (Fig. 5A-C; Supp. Fig. 24), and are also able to match object-independent spatial biases in

human behaviour (Fig. 5E), but not object-specific spatial biases (Fig. 5F). Hence, our results should not be interpreted as showing that supervised models are overall more promising than self-supervised approaches. Rather, for almost all results, what matters is the simple constraint of encouraging similar selectivity in neighbouring units. This requirement is sufficient to develop topographies reminiscent of the primate cortex, and human-aligned spatial biases in behaviour to some extent, independent of the specific training objective used. Only object-specific spatial biases are not human-aligned when using SimCLR, a finding that may have several causes, as described in a new discussion paragraph.

Spatial smoothness can be associated with the metabolic constraints discussed in the literature that you mention. For example, we now show that it can be linked to energy efficiency: in a new results section, we show that All-TNNs are more energy efficient than control networks, and that regions that are smoother consume less energy. In addition, our lesion study shows that spatial smoothness can be used as a proxy to determine which units can be removed without strongly harming performance, again tying into efficient coding. In this way, our approach connects to, builds on, and extends previous work. This is now mentioned in the discussion.

Margalit, E. et al. A Unifying Principle for the Functional Organization of Visual Cortex. 2023.05.18.541361 Preprint at <https://doi.org/10.1101/2023.05.18.541361> (2023).

Margalit, E., et al. A unifying framework for functional organisation in early and higher ventral visual cortex. *Neuron* (2024).

Actions:

- *p.17 l.523ff*: We have expanded the discussion to position ourselves better among other models in the literature.
- *Fig. 5; Supp. Fig. 24*: We have analysed an All-TNN trained with SimCLR.
- *Supp. Fig. 20*: We have added two TDANN control models for the behavioural experiment (one instance trained on object categorisation and five using SimCLR).
- *Fig. 3, Supp. Fig. 15*: We have added an analysis that quantifies energy consumption of all models.
- *p.16 l.479ff*: We mention the connection between the smoothness loss and literature on the influence of metabolic constraints on topography in the discussion.

I then have several secondary points:

2.1 - The analysis of the first layer is detailed, but is less precise for the following layers. Do you observe a similar organization? The origin of the variability of the topographies observed remains an enigma (cf doi: 10.1016/j.neuron.2021.09.053 for example) and your study could help to explain it.

Response: We agree that topography in intermediate layers is an intriguing (and challenging) question. To study topography in intermediate layers of All-TNN, we implemented two new analyses. First, we asked whether our observation of the centre-periphery organisation (see Fig. 2), is also

present in intermediate network layers. To test this, we have added an analysis measuring the spatial smoothness of the unit weights in each layer. We find that the centre-periphery distinction is replicated throughout all layers (Supp. Fig. 2). Second, we quantified energy consumption across layers (Fig. 3C; Supp. Fig. 15), and found that energy expenditure is focused in regions with task-relevant information in most layers.

Please note that, while feature selectivity in early and high brain areas along the ventral stream are somewhat well documented, the tuning of intermediate regions is less well understood. Therefore, analogously to the brain, it is difficult to select appropriate stimuli to determine selectivities in intermediate layers of All-TNNs. Future work in this direction could generate stimuli that maximally activate units in a given layer.

Lastly, we note that our current networks cannot easily model the work on the receptive fields of the tree shrew V2 that you shared with us, because our layers are structured retinotopically (except the final layer), i.e. prohibiting the emergence of sinusoidal transformations of the visual field. It could be that an All-TNN that would have the possibility for units to learn their position and connectivity would develop such a variable topography throughout its layers - an interesting question for future work.

Actions:

- *Fig. 1C; Supp. Fig. 2:* We have added an analysis quantifying the weight smoothness of all layers.
- *Fig. 2C:* We now show that the centre-periphery organisation in the first layer of our models is shaped by task-relevant statistics in the dataset.
- *Fig. 3; Supp. Fig. 15:* Quantification of energy consumption throughout layers.

2.2 - Concerning the parallel between the model's response and observed human behavior, the regressions are significant, but the behavioral responses are highly variable. It seems possible that other variables are involved, such as the effect of the intrinsic size of the visual objects in the images, e.g. that a bus is a priori larger than an animal, as is for instance demonstrated in Figure 3 (e.g. airplane, bus, train vs pizza, laptop or scissors).

Response: Thank you for your comment. As part of the revision, we removed the (noisy) regressions you are referring to from the manuscript (old Fig. 4). We made this choice because these regressions were based on the spatial occurrence statistics of our 16 object categories in the COCO dataset, which neither our models nor human subjects are trained on.

To directly test for your suggestion of a possible effect of intrinsic size in our behavioural data, we added a new GLM analysis (new Fig. 4D) that models the human ADMs using several predictors mirroring factors of IT organisation suggested in the past. These include real-world-size, animacy, and “spikiness”. We find that real-world size indeed explains a significant amount of variance (and

animacy plays the most important role, as also evident in a new hierarchical clustering analysis, see Fig. 4D).

Finally, we repeated our behavioural analyses using different stimulus sizes as inputs to our ANN models (see also our response to your comment 2.3). The pattern of our results remained consistent across various sizes, showing that All-TNNs outperforming other networks in matching human spatial biases is robust and not dependent on the chosen presentation size (see Supp. Fig. 23).

Actions:

- We have removed our old figure 4, which, as you mentioned, was particularly noisy, because it was only indirectly linked to our narrative.
- *Supp. Fig. 23:* We have shown that our behavioural results generalise to other stimulus sizes (see comment 2.3).
- *New Fig. 4D:* We have added a hierarchical clustering analysis and GLM analysis to show object-specific patterns in the human behavioural data, and explicitly test the effect of real-world size following your suggestion.

2.3 - On the other hand, have you observed whether the network develops size invariance? This is a property inherent to the dataset used, and should emerge as the network is trained.

Response: Thank you for this suggestion. We explicitly tested size invariance in our networks by testing performance while scaling the size of ecocet test images from 50%-200% of the original size (Supp. Fig. 23E). We find that All-TNNs are similarly size invariant to CNNs, as performance changes similarly for both models when changing the size of test stimuli.

In addition, we now reproduce our core behavioural results (both accuracy map agreement, and Accuracy Dissimilarity Matrix (ADM) agreement) with different object sizes, ranging from 50%-200% of the original size (Supp. Fig. 23A-D). We find that the pattern of results is stable, with All-TNNs outperforming control models in matching human spatial biases in human behaviour overall. This observation also suggests some size tolerance in our networks.

Please note that all our networks are trained with image data augmented with scaling of up to 33%.

Actions:

- *Supp. Fig. 23E:* We tested our networks' performance using different stimulus sizes.
- *Supp. Fig. 23A-D:* We tested how well our networks match spatial biases in human behaviour using different stimulus sizes.

2.4 - In order to make a controlled comparison of the performance of the convolutional network versus the proposed network, it would be useful to quantify the number of free parameters in each of the networks being compared.

Response: We have added a table describing all architectures in detail (including parameter counts) to the supplementary material (Supp. Table 1). This makes it clearer that all models have the same number of units, the same kernel sizes, strides, layer shapes, etc. However, the CNN models do not have the same number of parameters as LCNs and All-TNNs, because weight sharing strongly reduces their parameter count. Note that our focus is not on performance, but rather on topographic features and spatial biases, so the number of parameters is not directly relevant to our claims. However, for full transparency on the relative performance of the networks, we now report the classification accuracy of all networks as part of the main text (Fig. 1B).

Actions:

- *Supp. Table 1:* we have added a table that contains all architectural details for CNNs, LCNs and TNNs.
- *Figure 1B:* we explicitly reported on classification performance for all models.

2.5 - The definition of localization loss seems to derive from a heuristic (as the average of cosine similarity over horizontal and vertical directions) and its use needs to be justified. In particular, is this loss isotropic and does not introduce biases on the cardinals? Will other measures (e.g. the sum of the positive parts of cosine similarities) give similar results?

Response: Thank you for pointing out the need to better motivate the specifics of our smoothness loss function. We now expand on the specific methods, and added a control to verify the validity of these choices.

The smoothness loss is not explicitly isotropic, as we only use the ‘east’ and ‘south’ neighbours. The decision to use only the south and east neighbours stems from the observation that employing a symmetric neighbourhood would result in all cosine distances between units being counted twice (as cosine distance is symmetric). Avoiding these redundant calculations is beneficial in our computationally intense setup.

Our rationale behind using only the cardinal directions and excluding the diagonal neighbours from the loss computation was that the similarity between diagonally adjacent units is achieved indirectly: if the weights of unit A are similar to its eastern neighbour B, which in turn is similar to its southern neighbour C, then A and C are (indirectly) driven to be similar. Hence, we omitted the diagonals for the sake of computational efficiency.

To validate these choices we added a new control network incorporating all 8 neighbours (i.e., an isotropic loss including diagonals). This control network achieves comparable categorization performance and selectivity maps, corroborating the adequacy of our original assumptions, and suggesting that no major bias in the cardinal directions is created by using a non-isotropic loss (Supp. Fig. 25).

Finally, cosine distance was chosen in order to enforce similar directions in weight space, without enforcing identical norms. The suggestion of enforcing only some units to be similar (e.g. by having a

threshold on the cosine distance) is an interesting suggestion that we will consider in future work. Note that there is a large array of possibilities when moving in this direction (other distance measures, choosing which units get to affect their neighbours based on gradient estimates, etc). We therefore decided to stick to the straightforward use of minimising cosine distance between neighbouring unit weight vectors. We mention in the discussion that different ways of enforcing smoothness is an important area of future research.

Actions:

- *p.20 l.606ff*: We better describe how our loss is computed in the Methods section, and justify the choice of only using ‘east’ and ‘south’ neighbours.
- *Supp. Fig. 25*: We have included a new control model trained using an isotropic loss (i.e., all 8 neighbours).
- *p.17 l.544ff*: We discuss different ways of enforcing smoothness as an important area of future research.

2.6 - The development of topographical structure is linked to the development of spontaneous waves of activity in the form of retinal waves, which help to structure topographical relationships in the network. Do you think you could observe similar waves of activity in a continuously activated network?

Response: We agree that retinal waves are an interesting candidate for pre-structuring the topography of All-TNNs before the model is exposed to natural stimuli. We have added a discussion of these ideas to our manuscript. One example of notable work in this area that might be of interest is the following paper which introduces RNNs that develop interesting topographies and dynamics based on retinal waves.

Keller, T. A., & Welling, M. (2023). Neural wave machines: learning spatiotemporally structured representations with locally coupled oscillatory recurrent neural networks. In International Conference on Machine Learning (pp. 16168-16189). PMLR.

Actions:

- *p.16 l.499ff*: We have added a discussion of pre-training with retinal waves.

Minor:

3.1 - Figure 1, the RFs in layer #5 are not to scale,

Response: As the RFs in the first layer would be extremely small, we decided not to make any RFs in this figure to scale but rather decided for “symbolic RFs” that are accompanied by the exact numbers (inspired by a visualisation format customary in ML). Nevertheless, we have reworked the network

illustration in Figure 1 to improve clarity and describe all architectural details in the methods section and Supp. Table 1.

Actions:

- *Fig. 1:* We have added a new and improved version of the figure.
- *Methods:* We have described the model more clearly.
- *Supp. Table 1:* All hyperparameters of the networks are detailed.

3.2 - page 6: when giving the accuracy after low entropy lesion, it is indeed a "relative change in accuracy" (I guessed it from seeing it is over 100%). This is indeed explained later in the paper.

Response: We agree that this was not sufficiently clear. In response, we have decided to report the actual accuracies instead of relative changes in accuracy. The lesion study is now placed in the Supp. Fig. 12.

Actions:

- *Supp. Fig. 12:* We have reported the actual classification accuracy instead of the relative accuracy.

3.3 - I couldn't find a quotation in your paper describing the Mondrian mask synthesis method.

Response: We have added the citation to the relevant paper (Stein et al., 2011) to relevant places in the paper (Results section on behaviour and Methods). We have used code by Martin Hebart to generate the masks, which can be found on his website, which we now cite in the Methods section as well.

Stein T, Hebart MN, Sterzer P. Breaking Continuous Flash Suppression: A New Measure of Unconscious Processing during Interocular Suppression? *Front Hum Neurosci.* 2011 Dec 20;5:167. doi: 10.3389/fnhum.2011.00167. PMID: 22194718; PMCID: PMC3243089.

Actions:

- *p.10 l.290ff; p.25 l.817ff:* We have added the relevant citations to the manuscript.

Dear editor, dear reviewers,

Thank you for the positive evaluation of our revision. We are happy that only a few points remain to be addressed, all of which are covered in this new version of the manuscript.

Detailed responses to each reviewer comment are provided below, following this format:

Text items in blue boxes are reviewer comments.

Response: Black text items are our replies.

Actions:

We provide page and line numbers for the main relevant changes made to the manuscript.

Reviewer #1

Thanks to the authors for their extensive revisions and thoughtful care to my comments. I re-read the entire manuscript and again find the direction of all-TNNs and addressing the weight-sharing issue to be an important direction for the field. The revised manuscript is much clearer, and I appreciate the alpha sweep inclusion, and the steps taken revising the manuscript, following interpreting those results. The authors are also more careful in many of the claims.

I have only one remaining point. To me one of the strong points of the work is the potential to test for position-dependent categorization effects. The first step of that is understanding how stable those effects are in humans. Supplementary figure 21 shows the lower bound noise ceilings which are around .3 and .4. This is modest but definitely workable. But, if I'm reading these results right, the model correlations with these human data are substantially lower than the reliability in the human data. To the point where, perhaps the first take home claim might be that "none of the models are able capturing much of the stable human effects". And the second claim, along the manipulation of interest is that, looking at relative performance among the models, the all-TNNs slightly edge out the other models.

This matters, because the stated claim at the end of this section is: "In summary, All-TNNs capture category-dependent spatial biases that match those of human behaviour". And, if I'm reading these results right, I just don't think that is an accurate statement! I think All-TNNs have a slight edge compared to other models, when trained with this supervised objective. I certainly wish they nailed it, but at least you can still say that this is a new evaluation test of topographic models, that leave room for other improvements to close the gap.

Also, if you want to put a group-level human noise ceiling on your graphs in Figure 4E/F, you could estimate the group ceiling by split-halving your subjects, iterating over all split halves, and using the spearman-brown prophecy correction formula to account for the half-data problem.

Assuming my interpretations are correct here, I think it is important to acknowledge the fit of the models relative to what they could be giving the reliability of the data. If there's a gap, there's a gap. State it as a future puzzle. To me that doesn't undermine the contributions of this work, even though it's a bummer that it's not higher. But there is value in non weight-sharing approaches, regardless of if they are yet to capture the full rich spatialized human behavioral measures.

Response: Thank you for your encouragement and helpful feedback. We agree that our wording in describing the match between All-TNNs and human behaviour was too strong and altered the manuscript accordingly. We have also added the noise ceiling estimates, following your suggestion, to Figures 4E/F. Finally, we added a part to the discussion of the manuscript to explicitly mention the remaining gap in explainable variance, while suggesting ways in which it could be closed by future research based on All-TNNs.

We would like to emphasise, however, that the amount of variance in human spatial biases that is captured by All-TNNs is far from trivial, as can be seen by the comparison with the CNN gold standard models, which barely capture any variance. More generally, there are two factors to be considered for a proper judgment of model performance. First, our comparisons of accuracy maps entail zero parameters that can be adjusted to improve the fit, contrary to e.g. encoding models. If

parametric flexibility in the mapping were allowed, one could easily achieve higher model correlations, but we think that our more conservative approach is to be preferred here because it constitutes a stronger model test. Secondly, all of our models are trained on ecoset, which consists of static natural images of relevant categories, yet differs from the typical human visual diet. Given this, All-TNNs display a remarkable agreement with behavior and outperform the control models by a large margin. Future work, as now mentioned in this revision, includes the move towards more natural input statistics (including spatial and temporal aspects of vision), as well as explorations of different task objectives, changes in dataset size, model scale, and different base architectures, such as lateral and top-down connectivity.

In line with your comment that All-TNNs constitute an important direction for the field, we now also highlight ways in which All-TNNs may enable the study of further aspects of human visual behavior, such as interactive effects of scene background on object perception, or multi-object perceptual phenomena.

Actions:

- *p.11 l.319; p.13 l.387, l.394ff, l.401ff*: We have revised claims that were too strong.
- *p.11, p.14, p.59, p.60*: We have added the group-level human noise ceilings Figures 4E-F, 5G-H, Supp. Fig. 20A-B, and Supp. Fig. 21A-B.
- *p.39, p.40*: We have added two supplemental tables to report all model fits to human behaviour
- *p.18 l.583ff*: We address the limited correlation of All-TNNs with human behaviour and how this correlation could be further increased by future modelling approaches.
- *p.12 and p.13*: We address the usage of group-level human noise ceiling in the main text.
- *p.27 l.935ff*: We have updated the Methods section according to the changes in the noise ceiling calculation.

Reviewer #3

1. First of all, I would like to thank the authors for the complete revision, the revised paper and the quality of the detailed responses to the reviewers.

I am fully satisfied with the responses to my comments, and the changes made to the paper have greatly improved its quality. I would recommend minor revisions. In particular, I would be very grateful for responses to the following points, and their inclusion in the paper if you consider this would strengthen the paper even further:

Response: Thank you very much for the positive feedback and for the additional suggestions, all of which are now integrated into the manuscript

A key component is the introduction of smoothness on weights in the loss. This is standard practice to work on weights for regularizing models (weight decay, ...) and I understand your choice. However, could you achieve the same result by using a smoothness on activities? This would be more biologically realistic as neurons tend to differentiate activities rather than synaptic weights.

Response: Thank you for drawing attention to our choice of applying the spatial loss to the weight kernels instead of the unit activations, which we now motivate more clearly in the manuscript. We chose to focus on weights, as they determine feature selectivity - the aspect of cortical organisation that we aim to model. Similar weight kernels imply similar activity profiles (now shown in Supp. Fig. 25, which demonstrates correlated activity of spatially nearby units that decreases with increasing distance). However, the reverse is not true - correlated activity does not necessitate similar feature selectivity. For example, it is conceivable that nearby units are selective for different features that co-occur in the input statistics, and thus exhibit correlated activity.

That being said, changing our smoothness loss term to work on unit activity would be entirely possible, but note that enforcing correlated activity is arguably also not biologically realistic (Blauch et al., 2022). Future research will aim to create models that do not need either smoothness constraint, but for which smoothness emerges as part of training (for example by introducing lateral connectivity (Weidler et al., 2021).

Blauch, N. M., Behrmann, M., & Plaut, D. C. (2022). A connectivity-constrained computational account of topographic organization in primate high-level visual cortex. *Proceedings of the National Academy of Sciences*, 119(3).

Weidler, T. et al. Biologically Inspired Semantic Lateral Connectivity for Convolutional Neural Networks. *arXiv* (2021).

Actions:

- *p.67 l.1410ff*: We have added a supplemental figure depicting the positive relation between unit proximity and activity correlation.
- *p.19 l.634ff*: We discuss our motivation for choosing to apply the smoothness loss to the weight kernels instead of to the activations in the Methods section.

2. I appreciated both reviewer #1's comment that there is no retinotopy (cortical magnification) in the input and your detailed response. It seems that your model optimizes efficiency by learning the non-uniform mapping, but that this non-linear transformation could also be introduced a priori using a mapping of the input. Such CNN-based architectures with retinotopic mapping do exist (see doi:10.48550/arXiv.2402.15480), and I would guess that the entropy map computed in Figure 2-B would become more uniform. Following your statements in the conclusion, could you say that an optimal mapping of the input would be one where you include this non-uniform mapping in your model and that the entropy becomes uniform? (Intuitively, this would optimize resource usage)

Response: Thank you for this interesting suggestion. We expect that if a non-uniform mapping would be applied as a transformation to the input data, the entropy of the first layer could end up being more uniform (as the peripheral compression is now taken care of). This result would be in line with experimental data from macaques suggesting that ganglion cell density accounts for the cortical magnification factor (see da Costa et al., 2024; Wässle et al., 2020). Future work might experiment with applying a retinal sampling to the input data to induce a foveal overrepresentation, and testing what effect applying the transformation on the input side has on model topography, learning capacity, and energy consumption.

da Costa, D., Kornemann, L., Goebel, R., & Senden, M. (2024). Convolutional neural networks develop major organizational principles of early visual cortex when enhanced with retinal sampling. *Scientific Reports*, 14(1), 8980.

Wässle, H., Grünert, U., Röhrenbeck, J., & Boycott, B. B. (1990). Retinal ganglion cell density and cortical magnification factor in the primate. *Vision research*, 30(11), 1897-1911.

Actions:

- *p.16 l.481:* We now discuss the potential effect of using foveal and retinotopy-inspired transformations on the entropy profile.

3. It would be useful to provide more details on the SimCLR method, in particular the set of augmentations used. Indeed, Figure 5-H shows the lack of similarity of the SimCLR-trained network, and this may be due to the fact that the augmentations used to guide learning in this scheme are based only on geometric transformations. If an augmentation based on localization and categorization were introduced, the similarity might increase.

Response: We have included more details about the SimCLR training procedure and image augmentations used during training in the Methods section. Please note that the SimCLR training was performed in two steps. During a pretraining phase, larger augmentations were used. Subsequently, a category readout was trained, but with much less augmentation, more closely aligned with the category training of the other networks. This is now detailed in the manuscript together with an acknowledgment that future work may explore augmentation settings that do not change positional information.

Actions:

- *p.20 l.671ff*: We now include a more detailed description of the SimCLR training procedure and used augmentations in the Methods section.
- *p.17 l.537ff*: We discuss the effect of image augmentations on the capacity of the SimCLR-trained network to capture object-specific biases.

4. Other minor points:

- in Figure 1, error bars are too small to be seen or plotted. Indicate them as values in the text or define a larger CI.
- page 7: " performance.To " > " performance. To "
- Figure 4 (and following): you do not define precisely "spikiness" in your text
- page 19: Eq(2) uses wrong indices
- page 20 "computationally efficiency." > "computational efficiency."

Response: Thank you for pointing out these additional points. We have made the appropriate changes to the manuscript to address them.

Actions:

- *p.3 l.98; p.4 l.121*: The mean values and confidence intervals for both classification and smoothness are now reported in the text, and we now refer to them in the figure description.
- *p.7 l.195* and *p.20 l.651*: We have resolved both typos.
- *p.19 l.625*: We have ensured that Equation 2 is now consistent with the formula as presented in Figure 1.
- *p.11 l.332; p.26 l.898*: We now more precisely define spikiness in the figure caption, and have expanded the Methods section by including the corresponding formula.